# Analytic Study of Families of Spurious Minima in Two-Layer ReLU Neural Networks: A Tale of Symmetry II

**Yossi Arjevani**
The Hebrew University
yossi.arjevani@gmail.com

**Michael Field**
UC Santa Barbara
mikefield@gmail.com

## Abstract

We study the optimization problem associated with fitting two-layer ReLU neural networks with respect to the squared loss, where labels are generated by a target network. We make use of the rich symmetry structure to develop a novel set of tools for studying families of spurious minima. In contrast to existing approaches which operate in limiting regimes, our technique directly addresses the nonconvex loss landscape for a finite number of inputs $d$ and neurons $k$, and provides analytic, rather than heuristic, information. In particular, we derive analytic estimates for the loss at different minima, and prove that modulo $O(d^{-1/2})$-terms the Hessian spectrum concentrates near small positive constants, with the exception of $\Theta(d)$ eigenvalues which grow linearly with $d$. We further show that the Hessian spectrum at global and spurious minima coincide to $O(d^{-1/2})$-order, thus challenging our ability to argue about statistical generalization through local curvature. Lastly, our technique provides the exact *fractional* dimensionality at which families of critical points turn from saddles into spurious minima. This makes possible the study of the creation and the annihilation of spurious minima using powerful tools from equivariant bifurcation theory.

One of the outstanding conundrums of deep learning concerns the ability of simple gradient-based methods to successfully train neural networks despite the nonconvexity of the associated optimization problems. Indeed, generic nonconvex optimization landscapes can exhibit wide and flat basins of attraction around poor local minima which may lead to a complete failure of such methods. The nature by which nonconvex problems associated with neural networks deviate from generic ones is currently not well-understood. In particular, much of the dynamics of gradient-based methods follows from the curvature of the loss landscape around local minima. It is therefore vital to study the local geometry of spurious (i.e., non-global local) and global minima in order to understand the mysterious mechanism which drives gradient-based methods towards minima of high quality. However, already establishing the very existence of spurious minima seems to be beyond reach of existing analytic tools; let alone rigorously arguing about their height, curvature and structure—the aim of this work.

In this paper, we focus on two-layer ReLU neural networks of the form

$$\sum_{i=1}^{k} \alpha_i \varphi(\langle \boldsymbol{w}_i, \boldsymbol{x} \rangle), \ W \in M(k,d), \ \boldsymbol{\alpha} \in \mathbb{R}^k, \tag{1}$$

where $\varphi(z) = \max\{0, z\}$ is the ReLU activation function acting entry-wise, $M(k,d)$ denotes the space of $k \times d$ matrices and $\boldsymbol{w}_i$ denotes the $i$th row of $W$. We are primarily interested in characterizing various optimization-related obstructions for local search method, independently of the expressive power of two-layer ReLU networks. Thus, data is understood to be fully realizable. Concretely, we assume that there are $d$ inputs which are drawn from the standard multivariate Gaussian distribution

35th Conference on Neural Information Processing Systems (NeurIPS 2021).

and are labeled by a *planted* target network. We consider directly optimizing the expected squared loss, which results in the following highly nonconvex optimization problem:

$$\mathcal{L}(W, \boldsymbol{\alpha}) := \frac{1}{2} \mathbb{E}_{\boldsymbol{x} \sim \mathcal{N}(\mathbf{0}, I_d)} \left[ \left( \sum_{i=1}^{k} \alpha_i \varphi(\langle \boldsymbol{w}_i, \boldsymbol{x} \rangle) - \sum_{i=1}^{d} \beta_i \varphi(\langle \boldsymbol{v}_i, \boldsymbol{x} \rangle) \right)^2 \right], \qquad (2)$$

where $k$ denotes the number of hidden neurons, $W \in M(k, d)$ and $\alpha_i \in \mathbb{R}^k$ are the optimization variables, and $\boldsymbol{v}_i$ and $\beta_i$ are fixed parameters. This setting, in which the data distribution is regulated rather than being allowed to admit worst-case behavior, has drawn a considerable amount of interest in recent years [66, 17, 19, 40, 61, 11, 24, 54, 4, 1], in part due to the growing number of evidences which indicate that any explanation for the empirical success of deep learning (DL) must take into account the intricate interplay between the network architecture, the input distribution and the label distribution (cf. [11, 9, 59] and references therein for hardness results of optimization and learnability under partial sets of assumptions). Moreover, as demonstrated later in the paper, in spite of its apparent simplicity, nonconvex problem (2) shares a few important characteristics with full-scale neural networks, such as low-dimensional minima and extremely skewed Hessian spectrum.

Learning problem (2) has also been studied in the statistical physics community starting from the 80' [23, 57, 37, 62, 18, 8, 52, 53, 51] under the student-teacher (ST) framework, in which one aims to adjust a *student* network so as to fit the output of a *teacher* network. The ST framework offers a clean venue for analyzing optimization-related aspects of neural network models in the spirit of physical reductionism. The success of DL models in the past decade has reinitiated a surge of interest in this framework, e.g., [7, 28, 42, 47]. However, despite the long tradition in the statistical physics community and the wide effort put nowadays by the machine learning community, the perplexing geometry of problem (2) still seems to be out of reach of existing analytic tools in regimes encountered in practice. In this paper, we present a novel set of symmetry-based tools which allows us, for the first time, to analytically characterize important aspects of the associated highly nonconvex landscape for a finite number of inputs and neurons.

Our contributions, in order of appearance, can be stated as follows:

- We demonstrate that, empirically, and in a well-defined sense, minima in two-layer ReLU neural networks *break* the symmetry of the target weight matrix. Although ReLU networks have been studied for many years, this phenomenon of symmetry breaking seems to have gone largely unnoticed.

- We show that symmetry breaking makes it possible to derive *analytic* expressions for families of spurious minima in the form of *fractional* power series in terms of $d$ and $k$. Crucially, in contrast to existing approaches which employ various limiting processes, e.g., [7, 28, 42, 47, 29, 43, 13, 33, 14], our method operates in the natural regime where $d$ and $k$ are finite.

- We develop a novel technique which yields an analytic characterization of the Hessian spectrum of minima to $O(d^{-1/2})$-order for $d \geq k$, and determine the exact *fractional* value of $d$ at which critical points turn from saddles into spurious minima—a key ingredient in understanding how over-parameterization annihilates spurious minima.

- Based on the unique access to high-dimensional spectral information, we closely examine a number of hypotheses in the machine learning literature pertaining to curvature, optimization and generalization. In particular, we prove that the Hessian spectrum at minima concentrates near small positive constants, with the exception of $\Theta(d)$ eigenvalues which grow linearly with $d$. Although this phenomenon of extremely skewed spectrum has been observed many times [10, 38, 55, 56], to our knowledge, this is the first time it has been established *rigorously* for two-layer ReLU networks. In addition, our analysis shows that the Hessian spectra of spurious and global minima are identical to $O(d^{-1/2})$-terms, and further implies that the inductive bias of stochastic gradient descent (SGD), provably, can not be exclusively explained in terms of local curvature [32, 36, 34, 63, 64, 12, 16].

The results presented in the paper are threefold: identifying a symmetry breaking principle for two-layer ReLU networks, analytically characterizating spurious minima, and computing the Hessian spectrum. The next three sections are organized accordingly, along with a literature survey of related work. The last section is devoted for a high-level description of the novel symmetry-based technique used in this work. Proofs, formalities and lengthy technical details are deferred to the appendix.

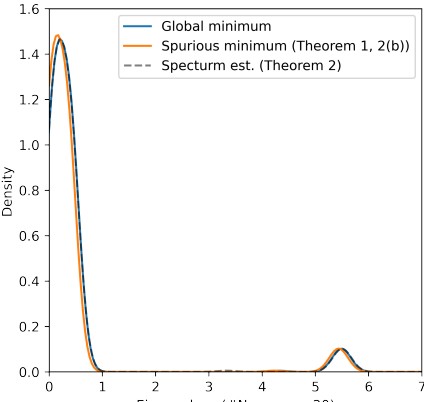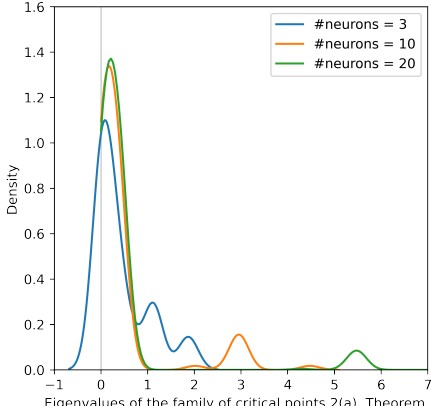

Figure 1: (Left) Our symmetry-based technique yields an *analytic* characterization of the Hessian spectrum (see Theorem 2) to $O(d^{-1/2})$-order (in a dashed line) which provides a good approximation already for small values of inputs and neurons (in solid lines). The analysis further implies that the Hessian spectrum of global and various types of spurious minima agree to within $O(d^{-1/2})$-accuracy. (Right) the Hessian spectrum is extremely skewed and tends to concentrate near small positive constants with the exception of $\Theta(d)$ eigenvalues which grow linearly with $d$. Observe that the family of critical points considered here (see Theorem 1, case 2a) turns from saddles at $d = 3$ into spurious minima when $d \geq 10$. Our analysis shows that the change of stability occurs at $d \approx 5.71$, and more importantly, indicates that the process can in fact be reversed, namely, spurious minima can be turned into saddles by over-parameterizing (i.e., increasing the number of hidden neurons).

## 1 Symmetry breaking in two-layer ReLU neural networks

Optimization problem (2) exhibits a very rich symmetry structure. Indeed, the loss function $\mathcal{L}$ is invariant to left- and right-multiplication of $W$ by permutation matrices (see Section A.1 for a formal proof), i.e., $\mathcal{L}(W, \boldsymbol{\alpha}) = \mathcal{L}(P_\pi W P_\rho^\top, \boldsymbol{\alpha})$ for all $(\pi, \rho) \in S_k \times S_d$, where $S_m$ generally denotes the symmetric group of degree $m$, and

$$(P_\pi)_{ij} = \begin{cases} 1 & i = \pi(j), \\ 0 & \text{o.w.} \end{cases}.$$

It is therefore natural to ask how the critical points of $\mathcal{L}$ reflect this symmetry. For example, the identity matrix $I_d$, one of the global minimizers of $\mathcal{L}$ for the case in which $V = I_d$, $\boldsymbol{\alpha} = \boldsymbol{\beta} = \mathbf{1}_d$ and $k = d$, is invariant under *simultaneous* left- and right-multiplication by any permutation matrix. Indeed, $P_\pi I_d P_\pi^\top = P_\pi P_\pi^\top = I_d$ for any $\pi \in S_d$. (Modulo group conjugation, this holds for any global minimizer of $\mathcal{L}$. See [6, Proposition 4.14.].) This simple observation is conveniently stated using the concept of the *isotropy* group. Given a weight matrix $W \in M(k, d)$, we let

$$\text{Iso}(W) := \{(\pi, \rho) \mid (\pi, \rho) \in S_k \times S_d, \ P_\pi W P_\rho^\top = W\}, \tag{3}$$

Thus, we have $\text{Iso}(I_d) = \Delta S_d$, where $\Delta$ maps any subgroup $H \subseteq S_d$ to its diagonal counterpart $\Delta H := \{(h, h) \mid h \in H\} \subseteq S_d \times S_d$. Empirically, and somewhat miraculously, spurious minima and saddles of $\mathcal{L}$ tend to be highly symmetric in the (formal) sense that their isotropy groups are (conjugated to) large subgroups of $\Delta S_d$, the isotropy of the global minima (see Figure 2). Thus, the principle of symmetry breaking can be concisely phrased as follows:



Spurious minima *break the symmetry* of global minima.



The principle extends to more general target networks; when the isotropy of the target weight matrix $V$ changes, the symmetry of spurious minima belonging to the respective optimization problem. In Section A, we provide a series of experiments which empirically corroborates symmetry breaking for optimization problem (2).

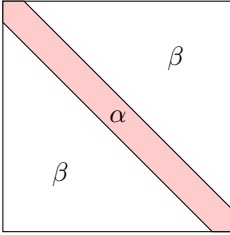 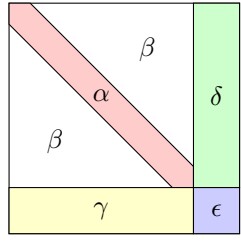 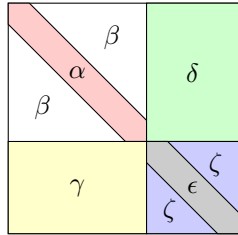

Isotropy $\Delta S_5$        Isotropy $\Delta S_4 \times \Delta S_1$        Isotropy $\Delta S_3 \times \Delta S_2$

Figure 2: A schematic description of $5 \times 5$ matrices with isotropy $\Delta S_5, \Delta S_4 \times \Delta S_1$ and $\Delta S_3 \times S_2$, from left to right (borrowed from [3]). $\alpha, \beta, \gamma, \delta, \epsilon$ and $\zeta$ are assumed to be 'sufficiently' different.

The principle of symmetry breaking for ReLU networks was first studied in [3, 6], and was later extended to various tensor decomposition problems in [2]. Here, we analyze two-layer ReLU networks where *both* layers are trainable, an architecture which has received a considerable amount of attention in recent years, e.g., [66, 17, 19, 40, 61, 11, 24]. We note in passing that, in its broader sense, the principle of symmetry breaking has been observed many times in various scientific fields, e.g., Higgs-Landau theory, equivariant bifurcation theory and replica symmetry-breaking (see, e.g., [46, 30, 21, 15]). Two-layer ReLU networks seems to form a rather unexpected instance of this principle.

One intriguing quality of nonconvex landscapes in which the symmetry breaking principle applies is that local minima lie in fixed low-dimensional spaces (see Section 4). A similar phenomenon of hidden low-dimensional structure has been observed in various learning problems in DL with real datasets [39, 31], and is believed by some to be an important factor of learnability in nonconvex settings. In the context of this work, the hidden low-dimensionality of spurious minima turns out to be a key ingredient to our analytic study, as we now present.

## 2 Power series representation of families of spurious minima

Although the problem of fitting neural networks under the ST framework have been studied for more than 30 years, the symmetry breaking principle exhibited by optimization problem (2) seems to have gone largely unnoticed. Early studies adopted tools (at times, heuristic, e.g., [45, 44]) from statistical physics to analyze phase transitions and generalization errors [23, 57, 37, 62]. Later work focused on the dynamics of SGD and studied the evolution of the generalization error along the optimization process through a set of carefully-derived ODEs [18, 8, 52, 53, 51]. Following the empirical success of DL in the past decade, this line of work has recently drawn a renewed interest, e.g, [7, 28, 47, 29], which puts past analyses on rigorous grounds and addresses a broader class of architectures and activation functions. The methods used in this long line of works operate in the *thermodynamic limit*—where the number of inputs is taken to infinity. Thus, the formal validity of the results to finite width networks is currently limited.

Other common approaches for analyzing problem (2) are based on: mean-field [43], optimal control [13], NTK [33] and compositional kernels [14]. These approaches offer, in essence, convex surrogates which apply in strict parameter regimes (including algorithmic parameters, such as learning rates). Similarly to the thermodynamic limit approach, the convex surrogates are obtained by limiting processes—this time by taking the number of hidden neurons to infinite. A growing number of works has severely limited, if not invalidated, the explanatory power of these approaches for network widths encountered in practice [65, 26].

In sharp contrast to the approaches discussed above, our technique directly addresses the associated highly nonconvex optimization landscape in the natural regime where the number of inputs and neurons is finite, and provides analytic, rather than heuristic, information. This is obtained by exploiting the presence of symmetry breaking phenomena whereby fractional power series representation for families of spurious minima are derived.

Below, we provide analytic expressions for families of minima of different isotropy (see Definition 3 and Figure 2), along with their respective objective value. For brevity, expressions are given to $O(d^{-3/2})$-order. In the appendix, we list additional $O(d^{-5/2})$-order terms which are required for

computing the Hessian spectrum. The invariance properties of optimization problem (2) imply that additional families of minima can be obtained by permuting the rows and the columns of a given family. The number of new distinct minima generated using such transformations (i.e., the minima multiplicity) depends on the very structure of the family of minima under consideration, and is stated in Theorem 1. Lastly, for any $\lambda > 0$ and $i \in [d]$, the value of the ReLU network (1) remains fixed under $(\boldsymbol{w}_i, \alpha_i) \mapsto (\lambda \boldsymbol{w}_i, \alpha_i/\lambda)$. This degree-of-freedom is expressed by using the slack variables $\lambda_i > 0$, $i \in [d]$ below.

**Theorem 1.** *Optimization problem (2) with $k = d$, $V = I_d$ and $\boldsymbol{\beta} = \mathbf{1}_d$ possesses the following families of minima for $d \geq 9$:*

1. *Two families of $\Delta S_d$-minima of multiplicity $d!$ with $W(d) = Diag(\lambda_1, \ldots, \lambda_d) A_d(a_1, a_2)$ and $\alpha(d) = (\lambda_1^{-1}, \ldots, \lambda_d^{-1})$, where $\lambda_i > 0$,*

$$
A_d(a_1, a_2) := \begin{pmatrix} a_1 & a_2 & \ldots & & a_2 \\ a_2 & a_1 & a_2 & \ldots & a_2 \\ & & \vdots & & \vdots \\ a_2 & \ldots & & a_2 & a_1 \end{pmatrix} \in M(d, d),
$$

   $Diag(\cdot)$ *maps a given vector to the diagonal entries of a diagonal matrix, and*

   (a) $a_1 = 1$ *and* $a_2 = 0$ *(a global minimizer), in which case*
   $$
   \mathcal{L}(W(d), \alpha(d)) = 0.
   $$

   (b) $a_1 = -1 + \frac{2}{d} + O\left(d^{\frac{-3}{2}}\right)$, $a_2 = \frac{2}{d} + O\left(d^{\frac{-3}{2}}\right)$, *in which case*
   $$
   \mathcal{L}(W(d), \alpha(d)) = -\frac{1}{\pi} + \frac{1}{2} - \frac{4}{3\pi\sqrt{d}} + \frac{-\frac{1}{2} - \frac{2}{\pi^2} + \frac{3}{\pi}}{d} + O\left(d^{\frac{-3}{2}}\right). \tag{4}
   $$

2. *Two families of $\Delta(S_{d-1} \times S_1)$-minima of multiplicity $d \cdot d!$ with*
   $$
   W(d) = Diag(\lambda_1, \ldots, \lambda_d) \left( \begin{array}{c|c} A_{d-1}(a_1, a_2) & a_3 \mathcal{I}_{d-1,1} \\ \hline a_4 \mathcal{I}_{1,d-1} & a_5 \end{array} \right),
   $$

   *and $\alpha(d) = (\lambda_1^{-1}, \ldots, \lambda_d^{-1})$, where $\lambda_i > 0$ and*

   (a) $a_1 = 1 + O\left(d^{\frac{-3}{2}}\right)$, $a_2 = O\left(d^{\frac{-3}{2}}\right)$, $a_3 = \frac{2}{d} + O\left(d^{\frac{-3}{2}}\right)$, $a_4 = \frac{4}{\pi d} + O\left(d^{\frac{-3}{2}}\right)$ *and* $a_5 = -1 + \frac{\frac{8}{\pi^2} + 2 + \frac{8}{\pi}}{d} + O\left(d^{\frac{-3}{2}}\right)$, *in which case*
   $$
   \mathcal{L}(W(d), \alpha(d)) = \frac{\frac{1}{2} - \frac{2}{\pi^2}}{d} + O\left(d^{\frac{-3}{2}}\right). \tag{5}
   $$

   (b) $a_1 = -1 + \frac{2}{d} + O\left(d^{\frac{-3}{2}}\right)$, $a_2 = \frac{2}{d} + O\left(d^{\frac{-3}{2}}\right)$, $a_3 = O\left(d^{\frac{-3}{2}}\right)$, $a_4 = \frac{2 - \frac{4}{\pi}}{d} + O\left(d^{\frac{-3}{2}}\right)$ *and* $a_5 = 1 + \frac{8(-1+\pi)}{\pi^2 d} + O\left(d^{\frac{-3}{2}}\right)$, *in which case*
   $$
   \mathcal{L}(W(d), \alpha(d)) = -\frac{1}{\pi} + \frac{1}{2} - \frac{4}{3\pi\sqrt{d}} + \frac{-1 - \frac{4}{\pi^2} + \frac{5}{\pi}}{d} + O\left(d^{\frac{-3}{2}}\right). \tag{6}
   $$

3. *Two families of $\Delta(S_{d-2} \times S_2)$-minima of multiplicity $d! \binom{d}{2}$ with*
   $$
   W(d) = Diag(\lambda_1, \ldots, \lambda_d) \left( \begin{array}{c|c} A_{d-2}(a_1, a_2) & a_3 \mathcal{I}_{d-2,2} \\ \hline a_4 \mathcal{I}_{2,d-2} & A_2(a_5, a_6) \end{array} \right),
   $$

   *and $\alpha(d) = (\lambda_1^{-1}, \ldots, \lambda_d^{-1})$, where $\lambda_i > 0$ and*

   (a) $a_1 = 1 + O\left(d^{\frac{-3}{2}}\right)$, $a_2 = 0 + O\left(d^{\frac{-3}{2}}\right)$, $a_3 = \frac{2}{d} + O\left(d^{\frac{-3}{2}}\right)$, $a_4 = \frac{4}{\pi d} + O\left(d^{\frac{-3}{2}}\right)$, $a_5 = -1 + \frac{\frac{8}{\pi^2} + 2 + \frac{8}{\pi}}{d} + O\left(d^{\frac{-3}{2}}\right)$ *and* $a_6 = \frac{2\left(-12\pi + 16 + \pi^3 + 4\pi^2\right)}{\pi^2 d(2+\pi)} + O\left(d^{\frac{-3}{2}}\right)$, *in which case*
   $$
   \mathcal{L}(W(d), \alpha(d)) = \frac{-4 + \pi^2}{\pi^2 d} + O\left(d^{\frac{-3}{2}}\right). \tag{7}
   $$

*(b)* $a_1 = -1 + \frac{2}{d} + O\left(d^{\frac{-3}{2}}\right)$, $a_2 = \frac{2}{d} + O\left(d^{\frac{-3}{2}}\right)$, $a_3 = 0 + O\left(d^{\frac{-3}{2}}\right)$, $a_4 = \frac{2-\frac{4}{\pi}}{d} + O\left(d^{\frac{-3}{2}}\right)$, $a_5 = 1 + \frac{8(-1+\pi)}{\pi^2 d} + O\left(d^{\frac{-3}{2}}\right)$ *and* $a_6 = \frac{4\left(-\pi^2 - 8 + 6\pi\right)}{\pi^2 d(2+\pi)} + O\left(d^{\frac{-3}{2}}\right)$, *in which case*

$$\mathcal{L}(W(d), \alpha(d)) = -\frac{1}{\pi} + \frac{1}{2} - \frac{4}{3\pi\sqrt{d}} + \frac{-\frac{3}{2} - \frac{6}{\pi^2} + \frac{7}{\pi}}{d} + O\left(d^{\frac{-3}{2}}\right). \tag{8}$$

4. *One family of* $\Delta(S_{d-3} \times S_3)$*-minima of multiplicity* $d! \binom{d}{3}$ *with*

$$W(d) = Diag(\lambda_1, \ldots, \lambda_d)\left(\begin{array}{c|c} A_{d-3}(a_1, a_2) & a_3 \mathcal{I}_{d-3,3} \\ \hline a_4 \mathcal{I}_{3,d-3} & A_2(a_5, a_6) \end{array}\right),$$

*and* $\alpha(d) = (\lambda_1^{-1}, \ldots, \lambda_d^{-1})$, *where* $\lambda_i > 0$ *and* $a_1 = 1 + O\left(d^{\frac{-3}{2}}\right)$, $a_2 = O\left(d^{\frac{-3}{2}}\right)$, $a_3 = \frac{2}{d} + O\left(d^{\frac{-3}{2}}\right)$, $a_4 = \frac{4}{\pi d} + O\left(d^{\frac{-3}{2}}\right)$, $a_5 = -1 + \frac{\frac{8}{\pi^2} + 2 + \frac{8}{\pi}}{d} + O\left(d^{\frac{-3}{2}}\right)$ *and* $a_6 = \frac{2\left(-12\pi + 16 + \pi^3 + 4\pi^2\right)}{\pi^2 d(2+\pi)} + O\left(d^{\frac{-3}{2}}\right)$, *in which case*

$$\mathcal{L}(W(d), \alpha(d)) = \frac{\frac{3}{2} - \frac{6}{\pi^2}}{d} + O\left(d^{\frac{-3}{2}}\right). \tag{9}$$

The idea of the proof of Theorem 1 is given in Section 4.1, and the multiplicity computation, a simple application of the orbit-stabilizer theorem, in Section B. We note that the same technique can be used for other choices of target networks, mutatis mutandis. The power series stated in Theorem 1 also represent spurious minima for the under-parameterized $d > k$-case. This is obtained by appropriately padding the entries of the weight matrices by zeros (see [6, section 4.3] for details). The derivation of the respective Hessian spectrum is then an immediate application of [4, section E.1] to the spectral analysis given in Theorem 2 below (for the case where $d = k$). By contrast, studying the over-parameterized regime for which $d < k$ requires qualitatively different tools and is outside the scope of this work.

The analysis above reveals that not all local minima are alike: while the objective value of some minima decays like $\Theta(1/d)$, in other cases the objective value converges to the positive constant $\frac{1}{2} - \frac{1}{\pi}$. In particular, except for the global minima case 1a, all minima described above are spurious. The difference between the two types of behavior seems to lie in the limiting values of the entries of $W(d)$; in the former, the diagonal mainly consists of ones, whereas in the latter mainly minus ones. This is consistent with the fact that all the teacher's diagonal entries are ones.

For reasons which are yet to be understood, empirically, the bias induced by Xavier initialization [27] seems to favor the class of minima for which the objective value decays as $\Theta(1/d)$. This is not to say that under Xavier initialization the expected value upon convergence must decrease with $d$. The objective value decays to zero at different rates (see cases 2a, 3a, 4 above) and depends on the probability to converge to a given type of minima. (A possible proxy to the latter is the minima multiplicity stated in Theorem 1.)

## 3 Analytic study of the Hessian spectrum

Once a power series representation has been obtained, it is possible to analytically characterize various important properties of families of minima. Here, we use this representation to compute yet another fractional power series—this time, of the Hessian spectrum.

**Theorem 2.** *Assuming all the weights of the second layer of (1) are set to one, the (nonnegative) Hessian spectrum of the families of minima considered in Theorem 1 is*

| Eigenvalue | Multiplicity |
|---|---|
| $O(d^{-1/2})$ | $d$ |
| $\frac{1}{4} - \frac{1}{2\pi} + O(d^{-1/2})$ | $\frac{(d-1)(d-2)}{2}$ |
| $\frac{1}{2} - \frac{1}{\pi} + O(d^{-1/2})$ | $d-1$ |
| $\frac{1}{4} + O(d^{-1/2})$ | $d-1$ |
| $\frac{1}{4} + \frac{1}{2\pi} + O(d^{-1/2})$ | $\frac{d(d-3)}{2}$ |
| $\frac{d}{4} + \frac{1}{2} + O(d^{-1/2})$ | $d-1$ |
| $\frac{d}{4} + \frac{-4+\pi+\pi^2}{2\pi(-4+\pi)} + O(d^{-1/2})$ | $1$ |
| $\frac{d}{\pi} + \frac{-10\pi+8+\pi^2}{2\pi(-4+\pi)} + O(d^{-1/2})$ | $1.$ |

The method we developed for computing the Hessian spectrum builds on [4], but differs in three crucial aspects. First, in our setting the second layer is trainable. The associated so-called *isotypic decomposition* (a set of 'simple' subspaces compatible with the action of row- and column-permutations formally introduced in Section D) must therefore cover the space of $d \times d$ weight matrices, as well as the $d$ weights of the second layer. Secondly, instead of expressing the Hessian entries in terms of $d, W(d)$ and $\boldsymbol{\alpha}(d)$ and then extracting fractional power series for eigenvalues, we directly express the eigenvalues in these terms. This considerably simplifies computations and facilitates the computation of eigenvalues to potentially any order. Thirdly, and perhaps most importantly, computing the Hessian to high accuracy reveals that for small values of $d$ the families of minima presented in Theorem 1 are in fact saddles. Regarding $d$ as a real number, we pinpoint the exact *fractional* dimension at which a given family of critical points turns from saddles into spurious minima (see also Figure 1). One is now led into the dual question: is there a mechanism by which spurious minima may be annihilated (that is, turn into saddles)? We have found that the process of the creation of minima can be reversed in the over-parameterized regime. This provides a strong evidence for the benefits of increasing the number of student neurons from an optimization point of view—adding more neurons turns spurious minima into saddles, thus encouraging gradient-based methods to converge to minima of better quality. A rigorous study of this process requires tools from equivariant bifurcation theory, and is deferred to future work (cf. [5]).

Quite remarkably, although families of minima presented in Theorem 1 differ significantly, their Hessian spectra coincide to $O(d^{-1/2})$-order. Another noticeable implication of Theorem 2 is that the Hessian spectrum tends to be extremely skewed. Both phenomena, as well as their consequences for optimization-related aspects, are discussed in detail below .

**Positively-skewed Hessian spectral density.** Although first reported nearly 30 years ago [10], to our knowledge, this is the first time that this phenomenon of extremely skewed spectral density has been established *rigorously* for two trainable ReLU layers of arbitrarily large dimensionality. Early empirical studies of the Hessian spectrum [10] revealed that local minima tend to be extremely ill-conditioned. This intriguing observation was corroborated and further refined in a series of works [38, 55, 56] which studied how the spectrum evolves along the training process. It was noticed that, upon convergence, the spectral density decomposes into two parts: a bulk of eigenvalues concentrated near small positive constants, and a set of positive outliers located away from zero.

Due to the high computational cost of an exact computation of the Hessian spectrum (cubic in the problem parameters $k$ and $d$), this phenomenon of extremely skewed spectral densities has only been confirmed for small-scale networks. Other methods for extracting second-order information in large-scale problems roughly fall into two general categories. The first class of methods approximate the Hessian spectral density by employing various numerical estimation techniques, most notably stochastic Lanczos method (e.g., [25, 48]). These methods have provided various numerical evidences which indicate that a similar skewed spectrum phenomenon also occurs in full-scale modern neural networks. The second class of techniques builds on tools from random matrix theory. The latter approach yields an exact computation of the limiting spectral distribution (where the number of neurons is taken to infinity), assuming the inputs, as well as the model weights are drawn at random [49, 50, 41, 35]. In contrast, our method gives an analytic description of the spectral density for

any number of neurons (granted $d \geq 9$), and at critical points rather than randomly drawn weight matrices.

**The flat minima conjecture and implicit bias.** It has long been debated whether some notion of local curvature can be used to explain the remarkable generalization capabilities of modern neural networks [32, 36, 34, 63, 64, 12, 16]. One intriguing hypothesis argues that minima with wider basins of attraction tend to generalize better. The suggested intuitive explanation is that flatness promotes statistical and numerical stability; together with low empirical loss, these ingredients are widely used to achieve good generalization, cf. [58]. However, the analysis presented in Theorem 2 shows that the spectra of global minima and spurious minima agree to $O(d^{-1/2})$-order. Consequently, in our setting, local second-order curvature *cannot* be used to separate global from spurious minima. This rules out any notion of flatness which exclusively relies on the Hessian spectrum. Of course, other metrics of a 'wideness of basins' may well apply.

# 4 A symmetry-based analytic framework

In the sequel, we present the main ingredients of the symmetry-based technique developed in this work. The technique builds on, and significantly extends various components from [4, 3, 6]. To ease exposition, we illustrate with reference to the case where $k = d$ and the weights of the second layer are set to one. We let the resulting nonconvex function be denoted by $f(W) := \mathcal{L}(W, \mathbf{1})$.

## 4.1 Power series representation of minima

The first step in deriving a power series representation is to restrict the nonconvex function under consideration to *fixed point subspaces* (see [20] for a more complete account); spaces that consists of weight matrices which remain fixed under row- and column-permutations in a certain subgroup of $S_d \times S_d$. For concreteness, consider the space of all matrices which are invariant to permutations in $\Delta S_d$, i.e.,

$$\mathcal{W}_d := \{ W \in M(d,d) \mid W = P_\pi W P_\pi^\top \text{ for all } (\pi, \pi) \in \Delta S_d \}. \tag{10}$$

It is easy to verify that $\mathcal{W}_d$ is in fact a 2-dimensional subspace of the form $\mathcal{W}_d = \{(a_1, a_2) \mid a_1 I_d + a_2 (\mathbf{1}_d \mathbf{1}_d^\top - I)\}$, and that if $W \in \mathcal{W}_d$, then also $\nabla f(W) \in \mathcal{W}_d$ (see [3, Proposition 3]). Thus, one may regard $f$ as a function from $\mathbb{R}^2$ to $\mathbb{R}^2$. Next, we make the dependence of $f$ on $d$ explicit by defining $F : \mathbb{R}^3 \to \mathbb{R}^2 : (a_1, a_2, d) \mapsto \nabla f(a_1 I_d + a_2 (\mathbf{1}_d \mathbf{1}_d^\top - I_d))$. Note that although $d$ was initially taken to be a natural number, in the following explicit expressions of $F(a_1, a_2, d)$, we regard $d$ as a real variable

$$\left( \begin{array}{c} \frac{a_1 d}{2} - \frac{a_1 d \sin\left(\beta_{(1)}^{(1)}\right)}{2\pi \nu_{(1)}} + \frac{a_1\left(d^2-d\right)\sin\left(\alpha_{(1)}^{(2)}\right)}{2\pi} - \frac{a_1\left(d^2-d\right)\sin\left(\beta_{(1)}^{(2)}\right)}{2\pi \nu_{(1)}} - \frac{a_2 \alpha_{(1)}^{(2)}\left(d^2-d\right)}{2\pi} + \frac{a_2\left(d^2-d\right)}{2} + \frac{\beta_{(1)}^{(1)} d}{2\pi} - \frac{d}{2} \\ -\frac{a_1 \alpha_{(1)}^{(2)}\left(d^2-d\right)}{2\pi} + \frac{a_1\left(d^2-d\right)}{2} - \frac{a_2 \alpha_{(1)}^{(2)}\left(d^3-3d^2+2d\right)}{2\pi} + \frac{a_2\left(d^3-2d^2+2d\right)\sin\left(\alpha_{(1)}^{(2)}\right)}{2\pi} + \frac{a_2\left(d^3-2d^2+2d\right)}{2} - \frac{a_2\left(d^2-d\right)\sin\left(\beta_{(1)}^{(1)}\right)}{2\pi \nu_{(1)}} - \frac{a_2\left(d^3-2d^2+d\right)\sin\left(\beta_{(1)}^{(2)}\right)}{2\pi \nu_{(1)}} + \frac{\beta_{(1)}^{(2)}\left(d^2-d\right)}{2\pi} - \frac{d^2}{2} + \frac{d}{2} \end{array} \right)$$

where, here and below, $\alpha_{(i)}^{(j)}$ (resp. $\beta_{(i)}^{(j)}$) denotes the angles between the $i$th row of $W$ and the $j$th of $W$ (resp. $V$). Using the implicit function theorem one can establish the existence as well as the uniqueness of a path of critical points $(a_1(d), a_2(d))$. This gives a formal meaning for examining a given family of minima at a *fractional* dimensionality. Formalities are covered in length in [6].

We are left with the following two issues. First, the implicit function theorem does not yield an explicit form of $a_1(d)$ and $a_2(d)$. This issue is addressed by using a *real analytic* version of the implicit function theorem, and then computing the coefficients of the power series of $a_1(d)$ and $a_2(d)$ to a desired order. Secondly, the implicit function theorem requires initial values for $a_1$ and $a_2$ which solve $F(a_1, a_2, d) = 0$. To handle this issue, we form power series in $1/d$ and, loosely speaking, develop them at $d = \infty$. The advantage of this nonstandard manipulation is that the limiting entries of $W(d)$ have a very simple form, which are then used as the initial values required for invoking the implicit function theorem. In fact, due to the dependence of $\nabla f$ on the angels between rows of $W$ and $V$, the power series of $a_1(d)$ and $a_2(d)$ are expressed in terms of $1/\sqrt{d}$. Theorem 1 is established by following the recipe described here for various families of critical points of different isotropy. The gradient expressions involved in the process are lengthy and are therefore relegated to Section E.

## 4.2 Computing the Hessian spectrum

Our next goal is to derive an analytic characterization of the Hessian spectrum. The invariance properties of $\mathcal{L}$ imply that although the dimension of the Hessian, $k(d+1) \times k(d+1)$, depends on $k$ and $d$, the number of distinct eigenvalues remains fixed. This is an immediate consequence of the respective isotypic decomposition (see Section D). A formal introduction of the isotypic decomposition requires some familiarity with group actions, and is therefore deferred to Section D.2. Here, we shall focus on one particularly simple case.

The isotypic decomposition associated with $\Delta S_d$, the isotropy of $I_d$, implies that for any $d$,

$$
\overline{\mathfrak{Y}_d} := \mathrm{vec}\left(\begin{bmatrix}
0 & d-3 & 3-d & \cdots & 0 & 0 & 0 \\
d-3 & 0 & 0 & \cdots & -1 & -1 & -1 \\
3-d & 0 & 0 & \cdots & 1 & 1 & 1 \\
0 & -1 & 1 & \cdots & 0 & 0 & 0 \\
\cdots & \cdots & \cdots & \cdots & \cdots & \cdots \\
0 & -1 & 1 & \cdots & 0 & 0 & 0 \\
0 & -1 & 1 & \cdots & 0 & 0 & 0
\end{bmatrix}\right),
$$

is an eigenvector of $\nabla^2 f(I_d)$, where $\mathrm{vec}$ denotes the linear transformation which stacks the rows of a given matrix on top of one another as a column vector. Thus, the eigenvalue corresponding to $\overline{\mathfrak{Y}_d}$ equals $(\nabla^2 f(I_d)\overline{\mathfrak{Y}_d})_2/(d-3)$, which takes the following explicit form

$$
\frac{a_1^2}{2\pi \nu_{(1)}^2 \nu_{(1)}^{(2)}} + \frac{a_1^2 \sin\left(\alpha_{(1)}^{(2)}\right)}{2\pi \nu_{(1)}^2 \left(\nu_{(1)}^{(2)}\right)^2} - \frac{a_1 a_2}{\pi \nu_{(1)}^2 \nu_{(1)}^{(2)}} - \frac{a_1 a_2 \sin\left(\alpha_{(1)}^{(2)}\right)}{\pi \nu_{(1)}^2 \left(\nu_{(1)}^{(2)}\right)^2} + \frac{a_2^2}{2\pi \nu_{(1)}^2 \nu_{(1)}^{(2)}} + \frac{a_2^2 \sin\left(\alpha_{(1)}^{(2)}\right)}{2\pi \nu_{(1)}^2 \left(\nu_{(1)}^{(2)}\right)^2} + \frac{\alpha_{(1)}^{(2)}}{2\pi} + \frac{(d-1)\sin\left(\alpha_{(1)}^{(2)}\right)}{2\pi} - \frac{(d-1)\sin\left(\beta_{(1)}^{(2)}\right)}{2\pi \nu_{(1)}} - \frac{\sin\left(\beta_{(1)}^{(1)}\right)}{2\pi \nu_{(1)}} - \frac{\sin\left(\beta_{(1)}^{(2)}\right)}{2\pi \left(\mu_{(1)}^{(2)}\right)^2 \nu_{(1)}},
$$

where $\nu_{(i)}$ denotes the norm of the $i$th row of $W$, and $\nu_{(i)}^{(j)}$ (resp. $\mu_{(i)}^{(j)}$) denotes $\sin\arccos(\alpha_{(i)}^{(j)})$

(resp. $\sin\arccos(\beta_{(i)}^{(j)})$). To complete the derivation of the eigenvalue $\overline{\mathfrak{Y}_d}$ for families of isotropy $\Delta S_d$, e.g., cases 1(a) and 1(b) in Theorem 1, one plugs-in the power series representation of $a_1(d)$ and $a_2(d)$ into the expression above, and forms a new power series to get an estimate for $\overline{\mathfrak{Y}_d}$. The same procedure is used for all families of minima presented in Theorem 1, and is described in detail in Section 4.2.

## 5 Conclusion

When present, the principle of symmetry breaking forms a powerful tool of reducing the complexity of nonconvex problems in a nonlinear way. This yields unprecedented analytic information for ReLU networks with a finite number of inputs and neurons (rather than limiting models of varying qualities). At this stage of the theory, the extent to which the symmetry breaking principle applies is not well-understood. However, it does seem to hold in a broader class of fundamental nonconvex problems, e.g., [2].

An intriguing finding allowed by the principle of symmetry breaking is that various families of spurious and global minima are locally identical to $O(d^{-1/2})$-order. Since the multiplicity of local minima for $k = d$ is essentially exponential in $d$, it is rather unlikely for SGD to be able to escape spurious minima when $d$ grows (see [54, Table 1]). Why is it then that SGD is generally able to find good minima for optimization problems associated with fitting ReLU neural networks? Preliminary experiments we conducted indicate that *continuously* increasing $k$ and leaving $d$ fixed turns families of spurious minima turn into saddles. (The objective value seems to remain constant.) We believe that this phenomenon is instrumental for explaining the effectiveness of gradient-based methods, SGD in particular. A thorough study of this requires tools from equivariant bifurcation theory and is postponed to future work.

## Acknowledgements

Part of this work was completed while YA was a postdoctoral researcher at NYU. We thank Joan Bruna, Ohad Shamir, Shimon Shpiz and Daniel Soudry for valuable discussions.

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
