# A The principle of symmetry breaking

## A.1 $S_k \times S_d$-invariance properties of (2)

We show that $\mathcal{L}$ is $S_k \times S_d$-invariant in its first parameter. The proof is a straightforward adaption of [3, Section 4.1]. First, we make the dependence of $\mathcal{L}$ on the target weight $d \times d$-matrix $V$ explicit:

$$\overline{\mathcal{L}}(W, \boldsymbol{\alpha}; V, \boldsymbol{\beta}) := \frac{1}{2} \mathbb{E}_{\boldsymbol{x} \sim \mathcal{N}(\mathbf{0}, I_d)} \left[ \left( \sum_{i=1}^{k} \alpha_i \varphi(\langle \boldsymbol{w}_i, \boldsymbol{x} \rangle) - \sum_{i=1}^{d} \beta_i \varphi(\langle \boldsymbol{v}_i, \boldsymbol{x} \rangle) \right)^2 \right], \qquad (11)$$

Next, we observe that for any $\pi \in S_k, \rho \in S_d$ and $U \in \mathrm{O}(d)$, the group of all $d \times d$-orthogonal matrices, we have

$$\overline{\mathcal{L}}(W, \boldsymbol{\alpha}; V, \boldsymbol{\beta}) = \overline{\mathcal{L}}(P_\pi W, \boldsymbol{\alpha}; V, \boldsymbol{\beta}) = \overline{\mathcal{L}}(W, \boldsymbol{\alpha}; P_\rho V, \boldsymbol{\beta}), \qquad (12)$$

$$\overline{\mathcal{L}}(W, \boldsymbol{\alpha}; V, \boldsymbol{\beta}) = \overline{\mathcal{L}}(WU, \boldsymbol{\alpha}; VU, \boldsymbol{\beta}), \qquad (13)$$

where the last equality follows by the $\mathrm{O}(d)$-invariance of the standard multivariate Gaussian distribution. Therefore, for any $\rho \in S_d$ and $U \in \mathrm{O}(d)$ such that $V = P_\rho V U^\top$ and any $\pi \in S_k$, we have

$$\overline{\mathcal{L}}(W, \boldsymbol{\alpha}, V, \boldsymbol{\beta}) = \overline{\mathcal{L}}(W, \boldsymbol{\alpha}, P_\rho V U^\top, \boldsymbol{\beta}) \stackrel{(12)}{=} \overline{\mathcal{L}}(W, \boldsymbol{\alpha}, V U^\top, \boldsymbol{\beta}) \stackrel{(13)}{=} \overline{\mathcal{L}}(WU, \boldsymbol{\alpha}, V U^\top U, \boldsymbol{\beta})$$

$$= \overline{\mathcal{L}}(WU, \boldsymbol{\alpha}, V, \boldsymbol{\beta}) \stackrel{(12)}{=} \overline{\mathcal{L}}(P_\pi WU, \boldsymbol{\alpha}, V, \boldsymbol{\beta}).$$

In particular, for $V = I_d$, we have $V = P_\pi V P_\pi^\top$ for any $\pi \in S_d$, thus $\mathcal{L}(W, \boldsymbol{\alpha}) = \overline{\mathcal{L}}(W, \boldsymbol{\alpha}, I_d, \boldsymbol{\beta})$ is $S_k \times S_d$-invariant w.r.t. $W$. Note that here we do not exploit the rotational invariance of the standard Gaussian distribution, but rather its invariance to permutations. Hence, the same $S_k \times S_d$-invariance holds for any product distribution if $V = I_d$. Indeed, critical points admit maximal isotropy types also when the input distribution is $\mathcal{D} = \mathcal{U}([-1, 1]^d)$ (but not when $\mathcal{D} = \mathcal{U}([0, 2]^d)$).

## A.2 Examples of minima for Problem (2)

We display several examples for optimization problem (2) with $k = d = 10$ obtained by running SGD until the gradient norm is driven below $1e-8$.

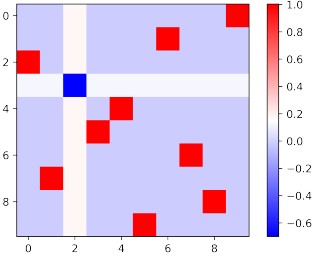

Figure 3: A spurious minimum of isotropy (conjugated to) $\Delta(S_{d-1} \times S_1)$ of (2) with $k = d = 10$. The objective value is $\approx 0.018$.

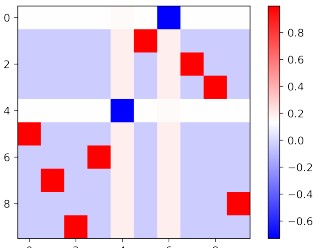

Figure 4: A spurious minimum of isotropy (conjugated to) $\Delta(S_{d-2} \times S_2)$ of (2) with $k = d = 10$. The objective value is $\approx 0.035$.

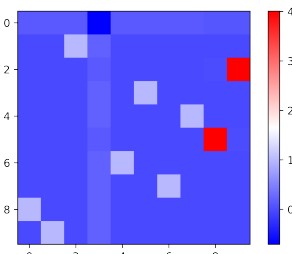

Figure 5: A spurious minimum of (2) with $k = d = 10$, where $V = \mathrm{Diag}(1, \ldots, 1, 2, 2)$ and $\beta = (1, \ldots, 1, 2, 2)$. The symmetry of the minimum adapt to that of the global minimizer $V$.

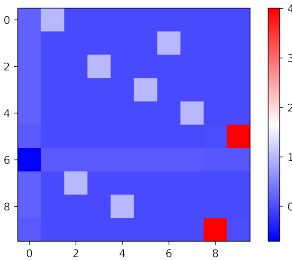

Figure 6: A spurious minimum of (2) with $k = d = 10$, where the $V = \mathrm{Diag}(1, \ldots, 1, 2, 2)$ and $\beta = (1, \ldots, 1, 2, 2)$. The symmetry of the minimum adapt to that of the global minimizer $V$.

## B    Counting multiplicity of families of minima

The computation of the multiplicity of minima is based on the orbit-stabilizer theorem. Instantiating this theorem to the natural action of $S_k \times S_d$ (i.e., row- and column- permutation. See Section D.1 below for a formal introduction of group action) yields

$$\mathrm{Multiplicity}(W) = \frac{|S_k \times S_d|}{|\mathrm{Iso}(W)|}.$$

Observing that $|S_k \times S_d| = d!\, k!$, $|\Delta S_d| = d!$, $|\Delta(S_{d-1} \times S_1)| = (d-1)!$, $|\Delta(S_{d-2} \times S_2)| = (d-2)!\, 2!$ and $|\Delta(S_{d-3} \times S_3)| = (d-3)!\, 3!$ gives the multiplicities stated in Theorem 1.

# C  Gradient expressions

In the sequel, we provide explicit expressions for $\nabla\mathcal{L}$ (defined in (2)) restricted the fixed point spaces $\mathcal{W}_{d-1}, \mathcal{W}_{d-2}, \mathcal{W}_{d-3}$, which naturally extend Definition 10 as follows

$$\mathcal{W}_p := \{W \in M(d,d) \mid W = P_\pi W P_\pi^\top \text{ for all } (\pi,\pi) \in \Delta(S_{d-p} \times S_p)\}.$$

The gradient expressions corresponding to $\mathcal{W}_d$ are given in the body of the paper in Section 4.1.

In all expressions below:

- $\alpha_{(i)}^{(j)}$ (resp. $\beta_{(i)}^{(j)}$) denotes the angles between the $i$th row of $W$ and the $j$th of $W$ (resp. $V$, the target weight matrix).
- $\nu_{(i)}$ (resp. $\mu_{(i)}$) denotes the norm of the $i$th row of $W$ (resp. $V$).
- $\nu_{(i)}^{(j)}$ (resp. $\mu_{(i)}^{(j)}$) denotes $\sin\arccos(\alpha_{(i)}^{(j)})$ (resp. $\sin\arccos(\beta_{(i)}^{(j)})$).

In Section F, we list the coefficients of the families of minima considered in Theorem 1 to $O(d^{-5/2})$-order.

## C.1  Gradient expressions for $\mathcal{W}_1$

The space $\mathcal{W}_1$ is five-dimensional. A weight matrix for $d = 8$ can be parameterized as follows

$$
\begin{bmatrix}
a_1 & a_2 & a_2 & a_2 & a_2 & a_2 & a_2 & a_3 \\
a_2 & a_1 & a_2 & a_2 & a_2 & a_2 & a_2 & a_3 \\
a_2 & a_2 & a_1 & a_2 & a_2 & a_2 & a_2 & a_3 \\
a_2 & a_2 & a_2 & a_1 & a_2 & a_2 & a_2 & a_3 \\
a_2 & a_2 & a_2 & a_2 & a_1 & a_2 & a_2 & a_3 \\
a_2 & a_2 & a_2 & a_2 & a_2 & a_1 & a_2 & a_3 \\
a_2 & a_2 & a_2 & a_2 & a_2 & a_2 & a_1 & a_3 \\
a_4 & a_4 & a_4 & a_4 & a_4 & a_4 & a_4 & a_5
\end{bmatrix}.
$$

The gradient entries, denoted by $g_1, g_2, g_3, g_4$ and $g_5$, are:

$$
g_1 = \frac{a_1(d-1)}{2} + \frac{a_1(d^2 - 3d + 2)\sin\left(\alpha_{(1)}^{(2)}\right)}{2\pi} + \frac{a_1\nu_{(d)}(d-1)\sin\left(\alpha_{(1)}^{(d)}\right)}{2\pi\nu_{(1)}}
$$

$$
- \frac{a_1(d-1)\sin\left(\beta_{(1)}^{(1)}\right)}{2\pi\nu_{(1)}} - \frac{a_1(d-1)\sin\left(\beta_{(1)}^{(d)}\right)}{2\pi\nu_{(1)}}
$$

$$
- \frac{a_1(d^2 - 3d + 2)\sin\left(\beta_{(1)}^{(2)}\right)}{2\pi\nu_{(1)}} - \frac{a_2\alpha_{(1)}^{(2)}(d^2 - 3d + 2)}{2\pi}
$$

$$
+ \frac{a_2(d^2 - 3d + 2)}{2} - \frac{a_4\alpha_{(1)}^{(d)}(d-1)}{2\pi} + \frac{a_4(d-1)}{2} + \frac{\beta_{(1)}^{(1)}(d-1)}{2\pi} - \frac{d}{2} + \frac{1}{2},
$$

$$
g_2 = -\frac{a_1\alpha_{(1)}^{(2)}(d^2 - 3d + 2)}{2\pi} + \frac{a_1(d^2 - 3d + 2)}{2} - \frac{a_2\alpha_{(1)}^{(2)}(d^3 - 6d^2 + 11d - 6)}{2\pi}
$$

$$
+ \frac{a_2(d^3 - 5d^2 + 8d - 4)\sin\left(\alpha_{(1)}^{(2)}\right)}{2\pi} + \frac{a_2(d^3 - 5d^2 + 8d - 4)}{2}
$$

$$
+ \frac{a_2\nu_{(d)}(d^2 - 3d + 2)\sin\left(\alpha_{(1)}^{(d)}\right)}{2\pi\nu_{(1)}} - \frac{a_2(d^2 - 3d + 2)\sin\left(\beta_{(1)}^{(1)}\right)}{2\pi\nu_{(1)}}
$$

$$
- \frac{a_2(d^2 - 3d + 2)\sin\left(\beta_{(1)}^{(d)}\right)}{2\pi\nu_{(1)}} - \frac{a_2(d^3 - 5d^2 + 8d - 4)\sin\left(\beta_{(1)}^{(2)}\right)}{2\pi\nu_{(1)}}
$$

$$
- \frac{a_4\alpha_{(1)}^{(d)}(d^2 - 3d + 2)}{2\pi} + \frac{a_4(d^2 - 3d + 2)}{2} + \frac{\beta_{(1)}^{(2)}(d^2 - 3d + 2)}{2\pi} - \frac{d^2}{2} + \frac{3d}{2} - 1,
$$

$$g_3 = -\frac{a_3\alpha_{(1)}^{(2)}\left(d^2-3d+2\right)}{2\pi} + \frac{a_3\left(d^2-3d+2\right)\sin\left(\alpha_{(1)}^{(2)}\right)}{2\pi} + \frac{a_3\left(d^2-2d+1\right)}{2}$$

$$+\frac{a_3\nu_{(d)}\left(d-1\right)\sin\left(\alpha_{(1)}^{(d)}\right)}{2\pi\nu_{(1)}} - \frac{a_3\left(d-1\right)\sin\left(\beta_{(1)}^{(1)}\right)}{2\pi\nu_{(1)}} - \frac{a_3\left(d-1\right)\sin\left(\beta_{(1)}^{(d)}\right)}{2\pi\nu_{(1)}}$$

$$-\frac{a_3\left(d^2-3d+2\right)\sin\left(\beta_{(1)}^{(2)}\right)}{2\pi\nu_{(1)}} - \frac{a_5\alpha_{(1)}^{(d)}\left(d-1\right)}{2\pi} + \frac{a_5\left(d-1\right)}{2} + \frac{\beta_{(1)}^{(d)}\left(d-1\right)}{2\pi} - \frac{d}{2} + \frac{1}{2},$$

$$g_4 = -\frac{a_1\alpha_{(1)}^{(d)}\left(d-1\right)}{2\pi} + \frac{a_1\left(d-1\right)}{2} - \frac{a_2\alpha_{(1)}^{(d)}\left(d^2-3d+2\right)}{2\pi}$$

$$+\frac{a_2\left(d^2-3d+2\right)}{2} + \frac{a_4\nu_{(1)}\left(d^2-2d+1\right)\sin\left(\alpha_{(1)}^{(d)}\right)}{2\pi\nu_{(d)}} + \frac{a_4\left(d-1\right)}{2}$$

$$-\frac{a_4\left(d-1\right)\sin\left(\beta_{(d)}^{(d)}\right)}{2\pi\nu_{(d)}} - \frac{a_4\left(d^2-2d+1\right)\sin\left(\beta_{(d)}^{(1)}\right)}{2\pi\nu_{(d)}} + \frac{\beta_{(d)}^{(1)}\left(d-1\right)}{2\pi} - \frac{d}{2} + \frac{1}{2},$$

$$g_5 = -\frac{a_3\alpha_{(1)}^{(d)}\left(d-1\right)}{2\pi} + \frac{a_3\left(d-1\right)}{2} + \frac{a_5\nu_{(1)}\left(d-1\right)\sin\left(\alpha_{(1)}^{(d)}\right)}{2\pi\nu_{(d)}}$$

$$+\frac{a_5}{2} - \frac{a_5\left(d-1\right)\sin\left(\beta_{(d)}^{(1)}\right)}{2\pi\nu_{(d)}} - \frac{a_5\sin\left(\beta_{(d)}^{(d)}\right)}{2\pi\nu_{(d)}} + \frac{\beta_{(d)}^{(d)}}{2\pi} - \frac{1}{2}.$$

## C.2 Gradient expressions for $\mathcal{W}_2$

The space $\mathcal{W}_2$ is six-dimensional. A weight matrix for $d = 8$ can be parameterized as follows

$$\begin{bmatrix} a_1 & a_2 & a_2 & a_2 & a_2 & a_2 & a_3 & a_3 \\ a_2 & a_1 & a_2 & a_2 & a_2 & a_2 & a_3 & a_3 \\ a_2 & a_2 & a_1 & a_2 & a_2 & a_2 & a_3 & a_3 \\ a_2 & a_2 & a_2 & a_1 & a_2 & a_2 & a_3 & a_3 \\ a_2 & a_2 & a_2 & a_2 & a_1 & a_2 & a_3 & a_3 \\ a_2 & a_2 & a_2 & a_2 & a_2 & a_1 & a_3 & a_3 \\ a_4 & a_4 & a_4 & a_4 & a_4 & a_4 & a_5 & a_6 \\ a_4 & a_4 & a_4 & a_4 & a_4 & a_4 & a_6 & a_5 \end{bmatrix}.$$

The gradient entries, denoted by $g_1, g_2, g_3, g_4, g_5$ and $g_6$, are:

$$g_1 = \frac{a_1\left(d-2\right)}{2} + \frac{a_1\left(d^2-5d+6\right)\sin\left(\alpha_{(1)}^{(2)}\right)}{2\pi} + \frac{a_1\nu_{(d-1)}\left(d-2\right)\sin\left(\alpha_{(1)}^{(d-1)}\right)}{\pi\nu_{(1)}}$$

$$-\frac{a_1\left(d-2\right)\sin\left(\beta_{(1)}^{(1)}\right)}{2\pi\nu_{(1)}} - \frac{a_1\left(d-2\right)\sin\left(\beta_{(1)}^{(d-1)}\right)}{\pi\nu_{(1)}}$$

$$-\frac{a_1\left(d^2-5d+6\right)\sin\left(\beta_{(1)}^{(2)}\right)}{2\pi\nu_{(1)}} - \frac{a_2\alpha_{(1)}^{(2)}\left(d^2-5d+6\right)}{2\pi}$$

$$+\frac{a_2\left(d^2-5d+6\right)}{2} - \frac{a_4\alpha_{(1)}^{(d-1)}\left(d-2\right)}{\pi} + a_4\left(d-2\right) + \frac{\beta_{(1)}^{(1)}\left(d-2\right)}{2\pi} - \frac{d}{2} + 1,$$

$$g_2 = -\frac{a_1\alpha_{(1)}^{(2)}\left(d^2-5d+6\right)}{2\pi} + \frac{a_1\left(d^2-5d+6\right)}{2} - \frac{a_2\alpha_{(1)}^{(2)}\left(d^3-9d^2+26d-24\right)}{2\pi}$$

$$+\frac{a_2\left(d^3-8d^2+21d-18\right)\sin\left(\alpha_{(1)}^{(2)}\right)}{2\pi} + \frac{a_2\left(d^3-8d^2+21d-18\right)}{2}$$

$$+\frac{a_2\nu_{(d-1)}\left(d^2-5d+6\right)\sin\left(\alpha_{(1)}^{(d-1)}\right)}{\pi\nu_{(1)}} - \frac{a_2\left(d^2-5d+6\right)\sin\left(\beta_{(1)}^{(1)}\right)}{2\pi\nu_{(1)}}$$

$$-\frac{a_2\left(d^2-5d+6\right)\sin\left(\beta_{(1)}^{(d-1)}\right)}{\pi\nu_{(1)}} - \frac{a_2\left(d^3-8d^2+21d-18\right)\sin\left(\beta_{(1)}^{(2)}\right)}{2\pi\nu_{(1)}}$$

$$-\frac{a_4\alpha_{(1)}^{(d-1)}\left(d^2-5d+6\right)}{\pi} + a_4\left(d^2-5d+6\right) + \frac{\beta_{(1)}^{(2)}\left(d^2-5d+6\right)}{2\pi} - \frac{d^2}{2} + \frac{5d}{2} - 3,$$

$$g_3 = -\frac{a_3\alpha_{(1)}^{(2)}\left(d^2-5d+6\right)}{\pi} + \frac{a_3\left(d^2-5d+6\right)\sin\left(\alpha_{(1)}^{(2)}\right)}{\pi}$$

$$+a_3\left(d^2-4d+4\right) + \frac{2a_3\nu_{(d-1)}\left(d-2\right)\sin\left(\alpha_{(1)}^{(d-1)}\right)}{\pi\nu_{(1)}} - \frac{a_3\left(d-2\right)\sin\left(\beta_{(1)}^{(1)}\right)}{\pi\nu_{(1)}}$$

$$-\frac{2a_3\left(d-2\right)\sin\left(\beta_{(1)}^{(d-1)}\right)}{\pi\nu_{(1)}} - \frac{a_3\left(d^2-5d+6\right)\sin\left(\beta_{(1)}^{(2)}\right)}{\pi\nu_{(1)}} - \frac{a_5\alpha_{(1)}^{(d-1)}\left(d-2\right)}{\pi}$$

$$+a_5\left(d-2\right) - \frac{a_6\alpha_{(1)}^{(d-1)}\left(d-2\right)}{\pi} + a_6\left(d-2\right) + \frac{\beta_{(1)}^{(d-1)}\left(d-2\right)}{\pi} - d + 2,$$

$$g_4 = -\frac{a_1\alpha_{(1)}^{(d-1)}\left(d-2\right)}{\pi} + a_1\left(d-2\right) - \frac{a_2\alpha_{(1)}^{(d-1)}\left(d^2-5d+6\right)}{\pi}$$

$$+a_2\left(d^2-5d+6\right) - \frac{a_4\alpha_{(d-1)}^{(d)}\left(d-2\right)}{\pi} + \frac{a_4\nu_{(1)}\left(d^2-4d+4\right)\sin\left(\alpha_{(1)}^{(d-1)}\right)}{\pi\nu_{(d-1)}}$$

$$+\frac{a_4\left(d-2\right)\sin\left(\alpha_{(d-1)}^{(d)}\right)}{\pi} + 2a_4\left(d-2\right) - \frac{a_4\left(d-2\right)\sin\left(\beta_{(d-1)}^{(d)}\right)}{\pi\nu_{(d-1)}}$$

$$-\frac{a_4\left(d-2\right)\sin\left(\beta_{(d-1)}^{(d-1)}\right)}{\pi\nu_{(d-1)}} - \frac{a_4\left(d^2-4d+4\right)\sin\left(\beta_{(d-1)}^{(1)}\right)}{\pi\nu_{(d-1)}} + \frac{\beta_{(d-1)}^{(1)}\left(d-2\right)}{\pi} - d + 2,$$

$$g_5 = -\frac{a_3\alpha_{(1)}^{(d-1)}\left(d-2\right)}{\pi} + a_3\left(d-2\right) + \frac{a_5\nu_{(1)}\left(d-2\right)\sin\left(\alpha_{(1)}^{(d-1)}\right)}{\pi\nu_{(d-1)}} + \frac{a_5\sin\left(\alpha_{(d-1)}^{(d)}\right)}{\pi} + a_5$$

$$-\frac{a_5\left(d-2\right)\sin\left(\beta_{(d-1)}^{(1)}\right)}{\pi\nu_{(d-1)}} - \frac{a_5\sin\left(\beta_{(d-1)}^{(d)}\right)}{\pi\nu_{(d-1)}} - \frac{a_5\sin\left(\beta_{(d-1)}^{(d-1)}\right)}{\pi\nu_{(d-1)}} - \frac{a_6\alpha_{(d-1)}^{(d)}}{\pi} + a_6 + \frac{\beta_{(d-1)}^{(d-1)}}{\pi} - 1$$

$$g_6 = -\frac{a_3\alpha_{(1)}^{(d-1)}\left(d-2\right)}{\pi} + a_3\left(d-2\right) - \frac{a_5\alpha_{(d-1)}^{(d)}}{\pi} + a_5$$

$$+\frac{a_6\nu_{(1)}\left(d-2\right)\sin\left(\alpha_{(1)}^{(d-1)}\right)}{\pi\nu_{(d-1)}} + \frac{a_6\sin\left(\alpha_{(d-1)}^{(d)}\right)}{\pi} + a_6$$

$$-\frac{a_6\left(d-2\right)\sin\left(\beta_{(d-1)}^{(1)}\right)}{\pi\nu_{(d-1)}} - \frac{a_6\sin\left(\beta_{(d-1)}^{(d)}\right)}{\pi\nu_{(d-1)}} - \frac{a_6\sin\left(\beta_{(d-1)}^{(d-1)}\right)}{\pi\nu_{(d-1)}} + \frac{\beta_{(d-1)}^{(d)}}{\pi} - 1.$$

## C.3  Gradient expressions for $\mathcal{W}_3$

The space $\mathcal{W}_3$ is six-dimensional. A weight matrix for $d = 8$ can be parameterized as follows

$$
\begin{bmatrix}
a_1 & a_2 & a_2 & a_2 & a_2 & a_3 & a_3 & a_3 \\
a_2 & a_1 & a_2 & a_2 & a_2 & a_3 & a_3 & a_3 \\
a_2 & a_2 & a_1 & a_2 & a_2 & a_3 & a_3 & a_3 \\
a_2 & a_2 & a_2 & a_1 & a_2 & a_3 & a_3 & a_3 \\
a_2 & a_2 & a_2 & a_2 & a_1 & a_3 & a_3 & a_3 \\
a_4 & a_4 & a_4 & a_4 & a_4 & a_5 & a_6 & a_6 \\
a_4 & a_4 & a_4 & a_4 & a_4 & a_6 & a_5 & a_6 \\
a_4 & a_4 & a_4 & a_4 & a_4 & a_6 & a_6 & a_5
\end{bmatrix}
$$

The gradient entries, denoted by $g_1, g_2, g_3, g_4, g_5$ and $g_6$, are:

$$
\begin{aligned}
g_1 =& \frac{a_1\,(d-3)}{2} + \frac{a_1\,(d^2 - 7d + 12)\sin\left(\alpha^{(2)}_{(1)}\right)}{2\pi} + \frac{3a_1\nu_{(d-2)}\,(d-3)\sin\left(\alpha^{(d-2)}_{(1)}\right)}{2\pi\nu_{(1)}} \\[2mm]
& - \frac{a_1\,(d-3)\sin\left(\beta^{(1)}_{(1)}\right)}{2\pi\nu_{(1)}} - \frac{3a_1\,(d-3)\sin\left(\beta^{(d-2)}_{(1)}\right)}{2\pi\nu_{(1)}} \\[2mm]
& - \frac{a_1\,(d^2 - 7d + 12)\sin\left(\beta^{(2)}_{(1)}\right)}{2\pi\nu_{(1)}} - \frac{a_2\alpha^{(2)}_{(1)}\,(d^2 - 7d + 12)}{2\pi} + \frac{a_2\,(d^2 - 7d + 12)}{2} \\[2mm]
& - \frac{3a_4\alpha^{(d-2)}_{(1)}\,(d-3)}{2\pi} + \frac{3a_4\,(d-3)}{2} + \frac{\beta^{(1)}_{(1)}\,(d-3)}{2\pi} - \frac{d}{2} + \frac{3}{2}, \\[3mm]
g_2 =& - \frac{a_1\alpha^{(2)}_{(1)}\,(d^2 - 7d + 12)}{2\pi} + \frac{a_1\,(d^2 - 7d + 12)}{2} - \frac{a_2\alpha^{(2)}_{(1)}\,(d^3 - 12d^2 + 47d - 60)}{2\pi} \\[2mm]
& + \frac{a_2\,(d^3 - 11d^2 + 40d - 48)\sin\left(\alpha^{(2)}_{(1)}\right)}{2\pi} + \frac{a_2\,(d^3 - 11d^2 + 40d - 48)}{2} \\[2mm]
& + \frac{3a_2\nu_{(d-2)}\,(d^2 - 7d + 12)\sin\left(\alpha^{(d-2)}_{(1)}\right)}{2\pi\nu_{(1)}} - \frac{a_2\,(d^2 - 7d + 12)\sin\left(\beta^{(1)}_{(1)}\right)}{2\pi\nu_{(1)}} \\[2mm]
& - \frac{3a_2\,(d^2 - 7d + 12)\sin\left(\beta^{(d-2)}_{(1)}\right)}{2\pi\nu_{(1)}} - \frac{a_2\,(d^3 - 11d^2 + 40d - 48)\sin\left(\beta^{(2)}_{(1)}\right)}{2\pi\nu_{(1)}} \\[2mm]
& - \frac{3a_4\alpha^{(d-2)}_{(1)}\,(d^2 - 7d + 12)}{2\pi} + \frac{3a_4\,(d^2 - 7d + 12)}{2} + \frac{\beta^{(2)}_{(1)}\,(d^2 - 7d + 12)}{2\pi} - \frac{d^2}{2} + \frac{7d}{2} - 6, \\[3mm]
g_3 =& - \frac{3a_3\alpha^{(2)}_{(1)}\,(d^2 - 7d + 12)}{2\pi} + \frac{3a_3\,(d^2 - 7d + 12)\sin\left(\alpha^{(2)}_{(1)}\right)}{2\pi} + \frac{3a_3\,(d^2 - 6d + 9)}{2} \\[2mm]
& + \frac{9a_3\nu_{(d-2)}\,(d-3)\sin\left(\alpha^{(d-2)}_{(1)}\right)}{2\pi\nu_{(1)}} - \frac{3a_3\,(d-3)\sin\left(\beta^{(1)}_{(1)}\right)}{2\pi\nu_{(1)}} - \frac{9a_3\,(d-3)\sin\left(\beta^{(d-2)}_{(1)}\right)}{2\pi\nu_{(1)}} \\[2mm]
& - \frac{3a_3\,(d^2 - 7d + 12)\sin\left(\beta^{(2)}_{(1)}\right)}{2\pi\nu_{(1)}} - \frac{3a_5\alpha^{(d-2)}_{(1)}\,(d-3)}{2\pi} + \frac{3a_5\,(d-3)}{2} - \frac{3a_6\alpha^{(d-2)}_{(1)}\,(d-3)}{\pi} \\[2mm]
& + 3a_6\,(d-3) + \frac{3\beta^{(d-2)}_{(1)}\,(d-3)}{2\pi} - \frac{3d}{2} + \frac{9}{2},
\end{aligned}
$$

$$g_4 = -\frac{3a_1\alpha_{(1)}^{(d-2)}\,(d-3)}{2\pi} + \frac{3a_1\,(d-3)}{2} - \frac{3a_2\alpha_{(1)}^{(d-2)}\,(d^2-7d+12)}{2\pi}$$

$$+ \frac{3a_2\,(d^2-7d+12)}{2} - \frac{3a_4\alpha_{(d-2)}^{(d-1)}\,(d-3)}{\pi} + \frac{3a_4\nu_{(1)}\,(d^2-6d+9)\sin\left(\alpha_{(1)}^{(d-2)}\right)}{2\pi\nu_{(d-2)}}$$

$$+ \frac{3a_4\,(d-3)\sin\left(\alpha_{(d-2)}^{(d-1)}\right)}{\pi} + \frac{9a_4\,(d-3)}{2} - \frac{3a_4\,(d-3)\sin\left(\beta_{(d-2)}^{(d-1)}\right)}{\pi\nu_{(d-2)}}$$

$$- \frac{3a_4\,(d-3)\sin\left(\beta_{(d-2)}^{(d-2)}\right)}{2\pi\nu_{(d-2)}} - \frac{3a_4\,(d^2-6d+9)\sin\left(\beta_{(d-2)}^{(1)}\right)}{2\pi\nu_{(d-2)}} + \frac{3\beta_{(d-2)}^{(1)}\,(d-3)}{2\pi} - \frac{3d}{2} + \frac{9}{2},$$

$$g_5 = -\frac{3a_3\alpha_{(1)}^{(d-2)}\,(d-3)}{2\pi} + \frac{3a_3\,(d-3)}{2} + \frac{3a_5\nu_{(1)}\,(d-3)\sin\left(\alpha_{(1)}^{(d-2)}\right)}{2\pi\nu_{(d-2)}}$$

$$+ \frac{3a_5\sin\left(\alpha_{(d-2)}^{(d-1)}\right)}{\pi} + \frac{3a_5}{2} - \frac{3a_5\,(d-3)\sin\left(\beta_{(d-2)}^{(1)}\right)}{2\pi\nu_{(d-2)}} - \frac{3a_5\sin\left(\beta_{(d-2)}^{(d-1)}\right)}{\pi\nu_{(d-2)}}$$

$$- \frac{3a_5\sin\left(\beta_{(d-2)}^{(d-2)}\right)}{2\pi\nu_{(d-2)}} - \frac{3a_6\alpha_{(d-2)}^{(d-1)}}{\pi} + 3a_6 + \frac{3\beta_{(d-2)}^{(d-2)}}{2\pi} - \frac{3}{2},$$

$$g_6 = -\frac{3a_3\alpha_{(1)}^{(d-2)}\,(d-3)}{\pi} + 3a_3\,(d-3) - \frac{3a_5\alpha_{(d-2)}^{(d-1)}}{\pi} + 3a_5 - \frac{3a_6\alpha_{(d-2)}^{(d-1)}}{\pi}$$

$$+ \frac{3a_6\nu_{(1)}\,(d-3)\sin\left(\alpha_{(1)}^{(d-2)}\right)}{\pi\nu_{(d-2)}} + \frac{6a_6\sin\left(\alpha_{(d-2)}^{(d-1)}\right)}{\pi} + 6a_6$$

$$- \frac{3a_6\,(d-3)\sin\left(\beta_{(d-2)}^{(1)}\right)}{\pi\nu_{(d-2)}} - \frac{6a_6\sin\left(\beta_{(d-2)}^{(d-1)}\right)}{\pi\nu_{(d-2)}} - \frac{3a_6\sin\left(\beta_{(d-2)}^{(d-2)}\right)}{\pi\nu_{(d-2)}} + \frac{3\beta_{(d-2)}^{(d-1)}}{\pi} - 3.$$

# D   Hessian spectrum

Below, we describe the technique we use to derive an analytic description of the Hessian spectrum. Some parts follow [4] verbatim. In order to avoid a long preliminaries section, key ideas and concepts are introduced and organized so as to illuminate our strategy for analyzing the Hessian. We illustrate with reference to the global minimum $W = V$ where $d = k$, the second layer is all ones, and the target weight matrix $V$ is the identity $I_d$. In Section E, we provide the eigenvalue expressions for $\Delta(S_{d-1} \times S_1)$, organized by their isotypic component.

## D.1   Studying invariance properties via group action

We first review background material on group actions and fix notations (see [20, Chapters 1, 2] for a more complete account). Elementary concepts from group theory are assumed known. We start with two examples that are used later.

**Examples 1.** (1) The *symmetric group* $S_d$, $d \in \mathbb{N}$, is the group of permutations of $[d] \doteq \{1, \ldots, d\}$. (2) Let $\mathrm{GL}(d, \mathbb{R})$ denote the space of invertible linear maps on $\mathbb{R}^d$. Under composition, $\mathrm{GL}(d, \mathbb{R})$ has the structure of a group. The *orthogonal group* $\mathrm{O}(d)$ is the subgroup of $\mathrm{GL}(d, \mathbb{R})$ defined by $\mathrm{O}(d) = \{A \in \mathrm{GL}(d, \mathbb{R}) \mid \|Ax\| = \|x\|, \text{ for all } x \in \mathbb{R}^d\}$. Both $\mathrm{GL}(d, \mathbb{R})$ and $\mathrm{O}(d)$ can be viewed as groups of invertible $d \times d$ matrices.

Characteristically, these groups consist of *transformations* of a set and so we are led to the notion of a *G-space* $X$ where we have an *action* of a group $G$ on a set $X$. Formally, this is a group homomorphism from $G$ to the group of bijections of $X$. For example, $S_d$ naturally acts on $[d]$ as permutations and both $\mathrm{GL}(d, \mathbb{R})$ and $\mathrm{O}(d)$ act on $\mathbb{R}^d$ as linear transformations (or matrix multiplication).

An example, which we use extensively in studying the invariance properties of $\mathcal{L}$, is given by the action of the group $S_k \times S_d \subset S_{k \times d}$, $k, d \in \mathbb{N}$, on $[k] \times [d]$ defined by

$$(\pi, \rho)(i, j) = (\pi^{-1}(i), \rho^{-1}(j)), \ \pi \in S_k, \rho \in S_d, \ (i, j) \in [k] \times [d]. \tag{14}$$

This action induces an action on the space $M(k, d)$ of $k \times d$-matrices $A = [A_{ij}]$ by $(\pi, \rho)[A_{ij}] = [A_{\pi^{-1}(i), \rho^{-1}(j)}]$. The action can be defined in terms of permutation matrices but is easier to describe in terms of rows and columns: $(\pi, \rho)A$ permutes rows (resp. columns) of $A$ according to $\pi$ (resp. $\rho$). As mentioned in the introduction, for our choice of $V = I_d$, $\mathcal{L}$ is $S_k \times S_d$-invariant. Note that $\Delta S_d \approx S_d$. When we restrict the $S_d \times S_d$-action on $M(k, k)$ to $\Delta S_d$, we refer to the diagonal $S_d$-action, or just the $S_d$-action on $M(d, d)$. This action of $S_d$ on $M(d, d)$ maps diagonal matrices to diagonal matrices and should not be confused with the actions of $S_d$ on $M(d, d)$ defined by either permuting rows or columns.

**Example 2.** Take $p, q \in \mathbb{N}$, $p + q = d$, and consider the diagonal action of $S_p \times S_q \subset S_d$ on $M(d, d)$. Write $A \in M(d, d)$ in block matrix form as $A = \begin{bmatrix} A_{p,p} & A_{p,q} \\ A_{q,p} & A_{q,q} \end{bmatrix}$. If $(g, h) \in S_p \times S_q \subset S_d$, then

$(g, h)A = \begin{bmatrix} gA_{p,p} & (g, h)A_{p,q} \\ (h, g)A_{q,p} & hA_{q,q} \end{bmatrix}$ where $gA_{p,p}$ (resp. $hA_{q,q}$) are defined via the diagonal action of $S_p$ (resp. $S_q$) on $A_{p,p}$ (resp. $A_{q,q}$), and $(g, h)A_{p,q}$ and $(h, g)A_{q,p}$ are defined through the natural action of $S_p \times S_q$ on rows and columns. Thus, for $(g, h)A_{p,q}$ (resp. $(h, g)A_{q,p}$) we permute rows (resp. columns) according to $g$ and columns (resp. rows) according to $h$. In the case when $p = d - 1$, $q = 1$, $S_{d-1}$ will act diagonally on $A_{d-1,d-1}$, fix $a_{dd}$, and act by permuting the first $(d - 1)$ entries of the last row and column.

As mentioned in body of the paper, given $W \in M(d, d)$, the largest subgroup of $S_d \times S_d$ fixing $W$ is called the *isotropy* subgroup of $W$ and is used as means of measuring the symmetry of $W$. The isotropy subgroup of $V \in M(d, d)$ is the diagonal subgroup $\Delta S_d$. Our focus will be on critical points $W$ whose isotropy groups are subgroups of the target matrix $V = I_d$, that is, $\Delta S_d$ and $\Delta(S_{d-1} \times S_1)$ (see Figure 2—we use the notation $\Delta S_d$ as the isotropy is a *subgroup* of $S_d \times S_d$). In the next section, we show how the symmetry of local minima greatly simplifies the analysis of their Hessian.

## D.2 The spectrum of equivariant linear isomorphisms

If $G$ is a subgroup of $O(d)$, the action on $\mathbb{R}^d$ is called an *orthogonal* representation of $G$ (we often drop the qualifier orthogonal). Denote by $(\mathbb{R}^d, G)$ as necessary. The *degree* of a representation $(V, G)$ is the dimension of $V$ ($V$ will always be a linear subspace of some $\mathbb{R}^n$ with the induced Euclidean inner product). The action of $S_k \times S_d \subset S_{k \times d}$ on $M(k, d)$ is orthogonal with respect to the standard Euclidean inner product on $M(k, d) \approx \mathbb{R}^{k \times d}$ since the action permutes the coordinates of $\mathbb{R}^{k \times d}$ (equivalently, components of $k \times d$ matrices). Given two representations $(V, G)$ and $(W, G)$, a map $A : V \to W$ is called $G$-equivariant if $A(gv) = gA(v)$, for all $g \in G, v \in V$. If $A$ is linear and equivariant, we say $A$ is a $G$-*map*. Invariant functions naturally provide examples of equivariant maps. Thus the gradient $\nabla \mathcal{L}$ is a $S_k \times S_d$-equivariant self map of $M(k, d)$ and if $W$ is a critical point of $\nabla \mathcal{L}$ with isotropy $G \subset S_k \times S_d$, then $\nabla^2 \mathcal{L}(W) : M(k, d) \to M(k, d)$ is a $G$-map (see [6]). The equivariance of the Hessian is the key ingredient that allows us to study the spectral density at *symmetric* local minima.

A representation $(\mathbb{R}^n, G)$ is *irreducible* if the only linear subspaces of $\mathbb{R}^n$ that are preserved (invariant) by the $G$-action are $\mathbb{R}^n$ and $\{0\}$. Two orthogonal representations $(V, G)$, $(W, G)$ are *isomorphic* (and have the same *isomorphism class*) if there exists a $G$-map $A : V \to W$ which is a linear isomorphism. If $(V, G)$, $(W, G)$ are irreducible but not isomorphic then every $G$-map $A : V \to W$ is zero (as the kernel and the image of a $G$-map are $G$-invariant). If $(V, G)$ is irreducible, then the space $\mathrm{Hom}_G(V, V)$ of $G$-maps (endomorphisms) of $V$ is a real associative division algebra and is isomorphic by a theorem of Frobenius to either $\mathbb{R}, \mathbb{C}$ or $\mathbb{H}$ (the quaternions). The *only* case that will concern us here is when $\mathrm{Hom}_G(V, V) \approx \mathbb{R}$ when we say the representation is *real*.

**Example 3.** Let $n > 1$. Take the natural (orthogonal) action of $S_n$ on $\mathbb{R}^n$ defined by permuting coordinates. The representation is not irreducible since the subspace $T = \{(x, x, \cdots, x) \in \mathbb{R}^n \mid x \in \mathbb{R}\}$ is invariant by the action of $S_n$, as is the hyperplane $H_{n-1} = T^\perp = \{(x_1, \cdots, x_n) \mid \sum_{i \in [n]} x_i = 0\}$. It is easy to check that $(T, S_n)$, also called the *trivial* representation of $S_n$, and $(H_{n-1}, S_n)$, the *standard* representation, are irreducible, real, and not isomorphic.

Every representation $(\mathbb{R}^n, G)$ can be written uniquely, up to order, as an orthogonal direct sum $\oplus_{i \in [m]} V_i$, where each $(V_i, G)$ is an orthogonal direct sum of isomorphic irreducible representations $(V_{ij}, G)$, $j \in [p_i]$, and $(V_{ij}, G)$ is isomorphic to $(V_{i'j'}, G)$ if and only if $i' = i$. The subspaces $V_{ij}$ are *not* uniquely determined if $p_i > 1$. If there are $m$ distinct isomorphism classes $\mathfrak{v}_1, \cdots, \mathfrak{v}_m$ of irreducible representations, then $(\mathbb{R}^n, G)$ may be represented by the sum $p_1 \mathfrak{v}_1 + \cdots + p_m \mathfrak{v}_m$, where $p_i \geq 1$ counts the number of representations with isomorphism class $\mathfrak{v}_i$. Up to order, this sum (that is, the $\mathfrak{v}_i$ and their multiplicities) is uniquely determined by $(\mathbb{R}^n, G)$. This is the *isotypic decomposition* of $(\mathbb{R}^n, G)$ (see [60]). The isotypic decomposition is a powerful tool for extracting information about the spectrum of $G$-maps.

If $G = S_d$, then every irreducible representation of $S_d$ is real [22, Thm. 4.3]. Suppose, as above, that $(\mathbb{R}^n, S_d) = \oplus_{i \in [m]} V_i$ and $A : \mathbb{R}^n \to \mathbb{R}^n$ is an $S_d$-map. Since the induced maps $A_{ii'} : V_i \to V_{i'}$ must be zero if $i \neq i'$, $A$ is uniquely determined by the $S_d$-maps $A_{ii} : V_i \to V_i$, $i \in [m]$. Fix $i$ and choose an $S_d$-representation $(W, S_d)$ in the isomorphism class $\mathfrak{v}_i$. Choose $S_d$-isomorphisms $W \to V_{ij}$, $j \in [p_i]$. Then $A_{ii}$ induces $\overline{A}_{ii} : W^{p_i} \to W^{p_i}$ and so determines a (real) matrix $M_i \in M(p_i, p_i)$ since $\text{Hom}_{S_d}(W, W) \approx \mathbb{R}$. Different choices of $V_{ij}$, or isomorphism $W \to V_{ij}$, yield a matrix similar to $M_i$. Each eigenvalue of $M_i$ of multiplicity $r$ gives an eigenvalue of $A_{ii}$, and so of $A$, of multiplicity $r \, \text{degree}(\mathfrak{v}_i)$.

**Fact 1.** (Notations and assumptions as above.) If $A$ is the Hessian, all eigenvalues are real and each eigenvalue of $M_i$ of multiplicity $r$ will be an eigenvalue of $A$ with multiplicity $r \, \text{degree}(\mathfrak{v}_i)$. In particular, $A$ has most $\sum_{i \in [m]} p_i$ distinct real eigenvalues—regardless of the dimension of the underlying space.

Our strategy can be now summarized as follows. Given a local minima $W$, we compute the isotropy group $G \subset S_k \times S_d$ of $W$. Since the Hessian of $\mathcal{F}$ at $W$ is a $G$-map, may use the isotypic decomposition of the action of $G$ on $M(k, d)$ to extract the spectral properties of the Hessian. In our setting, local minima have large isotropy groups, typically, as large as $\Delta(S_p \times S_{d-p})$, $0 \leq p < d/2$. Studying the Hessian at these minima requires the isotopic decomposition corresponding to $\Delta(S_p \times S_{d-p})$, $0 \leq p < d/2$, which we detail in Theorem D.3 below.

### D.3 The isotypic decomposition of $(M(d, d), S_d)$ and the spectrum at $W = V$

Regard $M(d, d)$ as an $S_d$-space (diagonal action). The trivial representation, denoted by $\mathfrak{t}_d$, and the standard representation, denoted by $\mathfrak{s}_d$, introduced in Example 3 are examples of the many irreducible representations of $S_d$. In the general theory, each irreducible representation of $S_d$ is associated to a partition of the set $[d]$. The description of the isotypic decomposition of $(M(d, d), S_d)$ is relatively simple and uses just 4 irreducible representations of $S_d$ for $d \geq 4$.

- The trivial representation $\mathfrak{t}_d$ of degree 1.

- The standard representation $\mathfrak{s}_d$ of $S_d$ of degree $k - 1$.

- The exterior square representation $\mathfrak{r}_d = \wedge^2 \mathfrak{s}_d$ of degree $\frac{(d-1)(d-2)}{2}$.

- A representation $\mathfrak{y}_d$ of degree $\frac{d(d-3)}{2}$. We describe $\mathfrak{y}_d$ explicitly later in terms of symmetric matrices (formally, it is the representation associated to the partition $(d - 2, 2)$).

We omit the subscript $d$ when clear from the context. Assume that $d \geq 4$. We begin with a well-known result about the representation $\mathfrak{s} \otimes \mathfrak{s}$ (see, e.g., [22]). If $\mathfrak{s} \odot \mathfrak{s}$ denotes the symmetric tensor product of $\mathfrak{s}$, then

$$\mathfrak{s} \otimes \mathfrak{s} = \mathfrak{s} \odot \mathfrak{s} + \mathfrak{r} = \mathfrak{t} + \mathfrak{s} + \mathfrak{y} + \mathfrak{r}. \tag{15}$$

Since all the irreducible $S_d$-representations are real, they are isomorphic to their dual representations and so we have the isotypic decomposition

$$M(k, k) \quad \approx \quad \mathbb{R}^k \otimes \mathbb{R}^k \approx (\mathfrak{s} + \mathfrak{t}) \otimes (\mathfrak{s} + \mathfrak{t}) = 2\mathfrak{t} + 3\mathfrak{s} + \mathfrak{r} + \mathfrak{y}, \tag{16}$$

since $\mathfrak{t} \otimes \mathfrak{s} = \mathfrak{s}$ and $\mathfrak{t} \otimes \mathfrak{t} = \mathfrak{t}$.

Using Fact 1, information can immediately be deduced from Equation (16). For example, if $W$ is a critical point of isotropy $\Delta S_d$ (a fixed point of the $S_d$-action on $M(d, d)$), then the spectrum of the Hessian contains at most $2 + 3 + 1 + 1 = 7$ distinct eigenvalues which distribute as follows: $\mathfrak{t}$ contributes 2 eigenvalues of multiplicity 1, $\mathfrak{s}$ contributes 2 eigenvalues of multiplicity $d - 1$, $\mathfrak{r}$

contributes one eigenvalue of multiplicity $\frac{(d-1)(d-2)}{2}$, and $\mathfrak{y}$ contributes one eigenvalue of multiplicity $\frac{d(d-3)}{2}$. This applies to the global minimum $W = V$ and the spurious minimum of type A.

Next, we would like to compute the actual eigenvalues. We demonstrate the method for the single $\mathfrak{x}$-eigenvalue (the example given in the body of the paper in section Section 4.2 refers to the single $\mathfrak{y}$-eigenvalue). Pick a non-zero vector from the $\mathfrak{x}$-representation. For example,

$$
\mathfrak{X}^d = \begin{bmatrix}
0 & 1 & \dots & 1 & -(d-2) \\
-1 & 0 & \dots & 0 & 1 \\
\dots & \dots & \dots & \dots & \dots \\
-1 & 0 & \dots & 0 & 1 \\
(d-2) & -1 & \dots & -1 & 0
\end{bmatrix},
$$

where rows and columns sum to zero and the only non-zero entries are in rows and columns 1 and $d$. Let $\overline{\mathfrak{X}^d} \in \mathbb{R}^{d \times d}$ be defined by concatenating the rows of $\mathfrak{X}^d$. Since $\mathfrak{x}$ only occurs once in the isotopic decomposition and $\nabla^2 \mathcal{L}(V)$ is $S_d$-equivariant, $\overline{\mathfrak{X}^d}$ must be an eigenvector. In particular, $(\nabla^2 \mathcal{L}(V) \overline{\mathfrak{X}^d})_i = \lambda_{\mathfrak{x}} \overline{\mathfrak{X}^d_i}$, all $i \in [d^2]$. Choose $i$ so that $\overline{\mathfrak{X}^d_i} \neq 0$. For example, $\overline{\mathfrak{X}^d_2} = 1$. Matrix multiplication, yields $\lambda_{\mathfrak{x}} = 1/4 - 1/2\pi$. A similar analysis holds for the eigenvalue associated to $\mathfrak{y}$. The multiple factors $2\mathfrak{t}$ and $3\mathfrak{s}$ are handled by making judicious choices of orthogonal invariant subspaces and representative vectors in $M(k, k)$.

Having described the general strategy for analyzing the Hessian spectrum for global minima, we now examine the spectrum at various types of spurious minima. We need two additional ingredients: a specification of the entries of a given family of spurious minima and the respective isotypic decomposition; we begin with the latter.

As discussed in the body of the paper, the symmetry-based analysis of the Hessian relies on the fact that isotropy groups of spurious minima tend to be (and some provably are) maximal subgroups of the target matrix isotropy. For $V = I$, the relevant maximal isotropy groups are of the form $\Delta(S_p \times S_q)$, $p + q = d$. Below, we provide the corresponding isotypic decomposition. Assume $d = k$ and regard $M(d, d)$ as an $S_p \times S_q$-space, where $S_p \times S_q \subset S_d$ and the (diagonal) action of $S_d$ is restricted to the subgroup $S_p \times S_q$.

**Theorem** ([4, Theorem 4]). *The isotypic decomposition of* $(M(d, d), S_p \times S_q)$ *is given by:*

1. *If* $p = d - 1$, $q = 1$, *and* $k \geq 5$,

$$
M(d, d) = 5\mathfrak{t} + 5\mathfrak{s}_{d-1} + \mathfrak{x}_{d-1} + \mathfrak{y}_{d-1}.
$$

2. *If* $q \geq 2$, $d - 1 > p > p/2$ *and* $d \geq 4 + q$, *then*

$$
M(d, d) = 6\mathfrak{t} + 5\mathfrak{s}_p + a\mathfrak{s}_q + \mathfrak{x}_p + \mathfrak{y}_p + b\mathfrak{x}_q + c\mathfrak{y}_q + 2\mathfrak{s}_p \boxtimes \mathfrak{s}_q,
$$

*where if* $q = 2$, *then* $a = 4, b = c = 0$; *if* $q = 3$, *then* $a = 5, b = 1, c = 0$; *and if* $q \geq 4$, *then* $a = 5, b = c = 1$.

(There is a minor error in [4, Theorem 4]. The correct value for $a$ in the second case is as stated here.)

In contrast to the setting considered in [4], in our case the second layer is trainable. To compute the isotypic decomposition corresponding to this case (i.e., $M(d, d) \times \mathbb{R}^d$), we simply treat the weights of the second layer as an additional row of the weight matrix of the first layer. The additional row is split into $p$ entries and $d - p$ entries. This adds $\mathfrak{t} + \mathfrak{s}_p$ to the isotypic decomposition if $q = 0$, $2\mathfrak{t} + \mathfrak{s}_p$ to the isotypic decomposition if $q = 1$, and $2\mathfrak{t} + \mathfrak{s}_p + \mathfrak{s}_q$ otherwise.

**Theorem 4.** *The isotypic decomposition of* $(M(d, d), S_p \times S_q) \otimes (\mathbb{R}^d, S_p \times S_q)$ *is given by:*

1. *If* $p = d$, $q = 0$, *and* $k \geq 5$,

$$
M(d, d) = 3\mathfrak{t} + 4\mathfrak{s}_d + \mathfrak{x}_d + \mathfrak{y}_d.
$$

2. *If* $p = d - 1$, $q = 1$, *and* $k \geq 5$,

$$
M(d, d) = 7\mathfrak{t} + 6\mathfrak{s}_{d-1} + \mathfrak{x}_{d-1} + \mathfrak{y}_{d-1}.
$$

3. *If $q \geq 2$, $d - 1 > p > p/2$ and $d \geq 4 + q$, then*

$$M(d, d) = 8\mathfrak{t} + 6\mathfrak{s}_p + a\mathfrak{s}_q + \mathfrak{x}_p + \mathfrak{y}_p + b\mathfrak{x}_q + c\mathfrak{y}_q + 2\mathfrak{s}_p \boxtimes \mathfrak{s}_q,$$

*where if $q = 2$, then $a = 5, b = c = 0$; if $q = 3$, then $a = 6, b = 1, c = 0$; and if $q \geq 4$, then $a = 6, b = c = 1$.*

Theorem 4 implies that the Hessian spectrum of local minima (or critical points) with isotropy $\Delta(S_p \times S_q)$ has at most 9 distinct eigenvalues if (1) applies, at most 15 distinct eigenvalues if (2) applies, and if (3) holds, at most 24 distinct eigenvalues if $q = 2$, at most 25 distinct eigenvalues if $q = 3$, and at most 26 distinct eigenvalues if $q \geq 4$. We omit some less interesting cases when $k$ is small.

Following the same lines of argument described in Section D.3, the next step is to pick a set of non-zero representative vectors for each irreducible representation that will allow us to compute the spectrum. We adopt the same of choice of representative vectors from [4]. We demonstrate the final step with respect to the two trivial factors of $S_d$ in (16). Let $\mathfrak{D}_1$ and $\mathfrak{D}_2$ be the two representatives and let $\nabla^2 \mathcal{L}(\mathfrak{D}_i) = \alpha_{i1}\mathfrak{D}_1 + \alpha_{i2}\mathfrak{D}_2$, $i = 1, 2$. The eigenvalues of $\nabla^2 \mathcal{L}|2\mathfrak{t}$ are then the eigenvalues of the $2 \times 2$ transition matrix $A = [\alpha_{ij}]$. To compute the eigenvalues of $A$ to, say, $O(d^{-1/2})$-order, one solves the equation

$$\det(A - (x_1 d + x_2 d^{1/2} + x^3)) = 0,$$

for $x_1, x_2, x_3$, where sufficiently many coefficients of the power series of the entries of $A$ are assumed known. The same recipe is used for computing the rest of the eigenvalues.

# E  Eigenvalues transition matrices

Below, we provide the explicit form of the eigenvalue transition matrices for the natural representation of $S_d$.

**$\mathfrak{x}$-rep.**  Let the associated $1 \times 1$ transition matrix be denoted by $T^{\mathfrak{x}}$. Then,

$$
\begin{aligned}
T_{1,1}^{\mathfrak{x}} = &-\frac{a_1^2}{2\pi\nu_{(1)}^2\nu_{(1)}^{(2)}} + \frac{a_1^2 \sin\left(\alpha_{(1)}^{(2)}\right)}{2\pi\nu_{(1)}^2 \left(\nu_{(1)}^{(2)}\right)^2} + \frac{a_1 a_2}{\pi\nu_{(1)}^2\nu_{(1)}^{(2)}} - \frac{a_1 a_2 \sin\left(\alpha_{(1)}^{(2)}\right)}{\pi\nu_{(1)}^2 \left(\nu_{(1)}^{(2)}\right)^2} \\
&- \frac{a_2^2}{2\pi\nu_{(1)}^2\nu_{(1)}^{(2)}} + \frac{a_2^2 \sin\left(\alpha_{(1)}^{(2)}\right)}{2\pi\nu_{(1)}^2 \left(\nu_{(1)}^{(2)}\right)^2} + \frac{\alpha_{(1)}^{(2)}}{2\pi} + \frac{(d-1)\sin\left(\alpha_{(1)}^{(2)}\right)}{2\pi} \\
&- \frac{(d-1)\sin\left(\beta_{(1)}^{(2)}\right)}{2\pi\nu_{(1)}} - \frac{\sin\left(\beta_{(1)}^{(1)}\right)}{2\pi\nu_{(1)}} - \frac{\sin\left(\beta_{(1)}^{(2)}\right)}{2\pi \left(\mu_{(1)}^{(2)}\right)^2 \nu_{(1)}}.
\end{aligned}
$$

**$\mathfrak{y}$-rep.**  Let the associated $1 \times 1$ transition matrix be denoted by $T^{\mathfrak{y}}$. Then,

$$
\begin{aligned}
T_{1,1}^{\mathfrak{y}} = &\frac{a_1^2}{2\pi\nu_{(1)}^2\nu_{(1)}^{(2)}} + \frac{a_1^2 \sin\left(\alpha_{(1)}^{(2)}\right)}{2\pi\nu_{(1)}^2 \left(\nu_{(1)}^{(2)}\right)^2} - \frac{a_1 a_2}{\pi\nu_{(1)}^2\nu_{(1)}^{(2)}} - \frac{a_1 a_2 \sin\left(\alpha_{(1)}^{(2)}\right)}{\pi\nu_{(1)}^2 \left(\nu_{(1)}^{(2)}\right)^2} + \frac{a_2^2}{2\pi\nu_{(1)}^2\nu_{(1)}^{(2)}} + \frac{a_2^2 \sin\left(\alpha_{(1)}^{(2)}\right)}{2\pi\nu_{(1)}^2 \left(\nu_{(1)}^{(2)}\right)^2} \\
&+ \frac{\alpha_{(1)}^{(2)}}{2\pi} + \frac{(d-1)\sin\left(\alpha_{(1)}^{(2)}\right)}{2\pi} - \frac{(d-1)\sin\left(\beta_{(1)}^{(2)}\right)}{2\pi\nu_{(1)}} - \frac{\sin\left(\beta_{(1)}^{(1)}\right)}{2\pi\nu_{(1)}} - \frac{\sin\left(\beta_{(1)}^{(2)}\right)}{2\pi \left(\mu_{(1)}^{(2)}\right)^2 \nu_{(1)}}.
\end{aligned}
$$

**s-rep.** Let the associated $4 \times 4$ transition matrix be denoted by $T^{\mathfrak{s}}$. Then,

$$T^{\mathfrak{s}}_{1,1} = -\frac{a_1^2\,(d-1)\sin\left(\alpha^{(2)}_{(1)}\right)}{2\pi\nu^2_{(1)}} - \frac{a_1^2}{2\pi\nu^2_{(1)}\nu^{(2)}_{(1)}} + \frac{a_1^2\,(d-1)\sin\left(\alpha^{(2)}_{(1)}\right)\cos^2\left(\alpha^{(2)}_{(1)}\right)}{2\pi\nu^2_{(1)}\left(\nu^{(2)}_{(1)}\right)^2}$$

$$+ \frac{a_1^2\,(d-1)\sin\left(\beta^{(2)}_{(1)}\right)}{2\pi\nu^3_{(1)}} + \frac{a_1^2\sin\left(\beta^{(1)}_{(1)}\right)}{2\pi\nu^3_{(1)}} - \frac{a_1^2\,(d-1)\sin\left(\beta^{(2)}_{(1)}\right)\cos^2\left(\beta^{(2)}_{(1)}\right)}{2\pi\left(\mu^{(2)}_{(1)}\right)^2\nu^3_{(1)}}$$

$$- \frac{a_1^2\sin\left(\beta^{(1)}_{(1)}\right)\cos^2\left(\beta^{(1)}_{(1)}\right)}{2\pi\left(\mu^{(1)}_{(1)}\right)^2\nu^3_{(1)}} + \frac{a_1 a_2\cos\left(\alpha^{(2)}_{(1)}\right)}{\pi\nu^2_{(1)}\nu^{(2)}_{(1)}} - \frac{a_1 a_2\,(d-1)\sin\left(\alpha^{(2)}_{(1)}\right)\cos\left(\alpha^{(2)}_{(1)}\right)}{\pi\nu^2_{(1)}\left(\nu^{(2)}_{(1)}\right)^2}$$

$$+ \frac{a_1\sin\left(\beta^{(1)}_{(1)}\right)\cos\left(\beta^{(1)}_{(1)}\right)}{\pi\left(\mu^{(1)}_{(1)}\right)^2\nu^2_{(1)}} - \frac{a_2^2}{2\pi\nu^2_{(1)}\nu^{(2)}_{(1)}} + \frac{a_2^2\,(d-1)\sin\left(\alpha^{(2)}_{(1)}\right)}{2\pi\nu^2_{(1)}\left(\nu^{(2)}_{(1)}\right)^2}$$

$$+ \frac{(d-1)\sin\left(\alpha^{(2)}_{(1)}\right)}{2\pi} + \frac{1}{2} - \frac{(d-1)\sin\left(\beta^{(2)}_{(1)}\right)}{2\pi\nu_{(1)}} - \frac{\sin\left(\beta^{(1)}_{(1)}\right)}{2\pi\nu_{(1)}} - \frac{\sin\left(\beta^{(1)}_{(1)}\right)}{2\pi\left(\mu^{(1)}_{(1)}\right)^2\nu_{(1)}},$$

$$T^{\mathfrak{s}}_{1,2} = \frac{a_1^2 d\cos\left(\alpha^{(2)}_{(1)}\right)}{2\pi\nu^2_{(1)}\nu^{(2)}_{(1)}} - \frac{a_1^2 d\sin\left(\alpha^{(2)}_{(1)}\right)\cos\left(\alpha^{(2)}_{(1)}\right)}{2\pi\nu^2_{(1)}\left(\nu^{(2)}_{(1)}\right)^2} - \frac{a_1 a_2 d}{\pi\nu^2_{(1)}\nu^{(2)}_{(1)}} + \frac{a_1 a_2 d\sin\left(\alpha^{(2)}_{(1)}\right)}{2\pi\nu^2_{(1)}\left(\nu^{(2)}_{(1)}\right)^2}$$

$$+ \frac{a_1 a_2 d\sin\left(\beta^{(1)}_{(1)}\right)}{2\pi\nu^3_{(1)}} - \frac{a_1 a_2 d\sin\left(\beta^{(1)}_{(1)}\right)\cos^2\left(\beta^{(1)}_{(1)}\right)}{2\pi\left(\mu^{(1)}_{(1)}\right)^2\nu^3_{(1)}} - \frac{a_1 a_2\,(d^2-d)\sin\left(\alpha^{(2)}_{(1)}\right)}{2\pi\nu^2_{(1)}}$$

$$- \frac{a_1 a_2\,(d^2-2d)\sin\left(\alpha^{(2)}_{(1)}\right)\cos\left(\alpha^{(2)}_{(1)}\right)}{2\pi\nu^2_{(1)}\left(\nu^{(2)}_{(1)}\right)^2} + \frac{a_1 a_2\,(d^2-d)\sin\left(\alpha^{(2)}_{(1)}\right)\cos^2\left(\alpha^{(2)}_{(1)}\right)}{2\pi\nu^2_{(1)}\left(\nu^{(2)}_{(1)}\right)^2}$$

$$+ \frac{a_1 a_2\,(d^2-d)\sin\left(\beta^{(2)}_{(1)}\right)}{2\pi\nu^3_{(1)}} - \frac{a_1 a_2\,(d^2-d)\sin\left(\beta^{(2)}_{(1)}\right)\cos^2\left(\beta^{(2)}_{(1)}\right)}{2\pi\left(\mu^{(2)}_{(1)}\right)^2\nu^3_{(1)}}$$

$$+ \frac{a_1 d\sin\left(\beta^{(2)}_{(1)}\right)\cos\left(\beta^{(2)}_{(1)}\right)}{2\pi\left(\mu^{(2)}_{(1)}\right)^2\nu^2_{(1)}} + \frac{a_2^2 d\cos\left(\alpha^{(2)}_{(1)}\right)}{2\pi\nu^2_{(1)}\nu^{(2)}_{(1)}} + \frac{a_2^2\,(d^2-2d)\sin\left(\alpha^{(2)}_{(1)}\right)}{2\pi\nu^2_{(1)}\left(\nu^{(2)}_{(1)}\right)^2}$$

$$- \frac{a_2^2\,(d^2-d)\sin\left(\alpha^{(2)}_{(1)}\right)\cos\left(\alpha^{(2)}_{(1)}\right)}{2\pi\nu^2_{(1)}\left(\nu^{(2)}_{(1)}\right)^2} + \frac{a_2 d\sin\left(\beta^{(1)}_{(1)}\right)\cos\left(\beta^{(1)}_{(1)}\right)}{2\pi\left(\mu^{(1)}_{(1)}\right)^2\nu^2_{(1)}} + \frac{\alpha^{(2)}_{(1)} d}{2\pi} - \frac{d}{2},$$

$$T_{1,3}^{\mathfrak{s}} = -\frac{a_1^2\,(d-2)\cos\left(\alpha_{(1)}^{(2)}\right)}{2\pi\nu_{(1)}^2\,\nu_{(1)}^{(2)}} - \frac{a_1^2\,(d-2)\sin\left(\alpha_{(1)}^{(2)}\right)\cos\left(\alpha_{(1)}^{(2)}\right)}{2\pi\nu_{(1)}^2\left(\nu_{(1)}^{(2)}\right)^2}$$

$$-\frac{a_1 a_2\,\left(d^2-3d+2\right)\sin\left(\alpha_{(1)}^{(2)}\right)}{2\pi\nu_{(1)}^2} + \frac{a_1 a_2\,(d-2)\cos\left(\alpha_{(1)}^{(2)}\right)}{\pi\nu_{(1)}^2\,\nu_{(1)}^{(2)}} + \frac{a_1 a_2\,(d-2)\sin\left(\alpha_{(1)}^{(2)}\right)}{2\pi\nu_{(1)}^2\left(\nu_{(1)}^{(2)}\right)^2}$$

$$-\frac{a_1 a_2\,\left(d^2-4d+4\right)\sin\left(\alpha_{(1)}^{(2)}\right)\cos\left(\alpha_{(1)}^{(2)}\right)}{2\pi\nu_{(1)}^2\left(\nu_{(1)}^{(2)}\right)^2} + \frac{a_1 a_2\,\left(d^2-3d+2\right)\sin\left(\alpha_{(1)}^{(2)}\right)\cos^2\left(\alpha_{(1)}^{(2)}\right)}{2\pi\nu_{(1)}^2\left(\nu_{(1)}^{(2)}\right)^2}$$

$$+\frac{a_1 a_2\,(d-2)\sin\left(\beta_{(1)}^{(1)}\right)}{2\pi\nu_{(1)}^3} + \frac{a_1 a_2\,\left(d^2-3d+2\right)\sin\left(\beta_{(1)}^{(2)}\right)}{2\pi\nu_{(1)}^3}$$

$$-\frac{a_1 a_2\,\left(d^2-3d+2\right)\sin\left(\beta_{(1)}^{(2)}\right)\cos^2\left(\beta_{(1)}^{(2)}\right)}{2\pi\left(\mu_{(1)}^{(2)}\right)^2\nu_{(1)}^3} - \frac{a_1 a_2\,(d-2)\sin\left(\beta_{(1)}^{(1)}\right)\cos^2\left(\beta_{(1)}^{(1)}\right)}{2\pi\left(\mu_{(1)}^{(1)}\right)^2\nu_{(1)}^3}$$

$$+\frac{a_1\,(d-2)\sin\left(\beta_{(1)}^{(2)}\right)\cos\left(\beta_{(1)}^{(2)}\right)}{2\pi\left(\mu_{(1)}^{(2)}\right)^2\nu_{(1)}^2} + \frac{a_2^2\,(d-2)\cos\left(\alpha_{(1)}^{(2)}\right)}{2\pi\nu_{(1)}^2\,\nu_{(1)}^{(2)}} - \frac{a_2^2\,(d-2)}{\pi\nu_{(1)}^2\,\nu_{(1)}^{(2)}}$$

$$+\frac{a_2^2\,\left(d^2-4d+4\right)\sin\left(\alpha_{(1)}^{(2)}\right)}{2\pi\nu_{(1)}^2\left(\nu_{(1)}^{(2)}\right)^2} - \frac{a_2^2\,\left(d^2-3d+2\right)\sin\left(\alpha_{(1)}^{(2)}\right)\cos\left(\alpha_{(1)}^{(2)}\right)}{2\pi\nu_{(1)}^2\left(\nu_{(1)}^{(2)}\right)^2}$$

$$+\frac{a_2\,(d-2)\sin\left(\beta_{(1)}^{(1)}\right)\cos\left(\beta_{(1)}^{(1)}\right)}{2\pi\left(\mu_{(1)}^{(1)}\right)^2\nu_{(1)}^2} - \frac{\alpha_{(1)}^{(2)}\,(d-2)}{2\pi} + \frac{d}{2} - 1,$$

$$T_{1,4}^{\mathfrak{s}} = \frac{a_1\,(d-2)\sin\left(\alpha_{(1)}^{(2)}\right)}{2\pi} + a_1 - \frac{a_1\,(d-1)\sin\left(\beta_{(1)}^{(2)}\right)}{2\pi\nu_{(1)}}$$

$$-\frac{a_1\sin\left(\beta_{(1)}^{(1)}\right)}{2\pi\nu_{(1)}} - \frac{a_2\alpha_{(1)}^{(2)}\,(d-2)}{2\pi} + \frac{a_2\,(d-2)}{2} + \frac{\beta_{(1)}^{(1)}}{2\pi} - \frac{1}{2},$$

$$T_{2,1}^{\mathfrak{s}} = \frac{a_1^2 \cos\left(\alpha_{(1)}^{(2)}\right)}{4\pi\nu_{(1)}^2 \nu_{(1)}^{(2)}} - \frac{a_1^2 \sin\left(\alpha_{(1)}^{(2)}\right)\cos\left(\alpha_{(1)}^{(2)}\right)}{4\pi\nu_{(1)}^2 \left(\nu_{(1)}^{(2)}\right)^2} - \frac{a_1 a_2 \left(d-1\right)\sin\left(\alpha_{(1)}^{(2)}\right)}{4\pi\nu_{(1)}^2} - \frac{a_1 a_2}{2\pi\nu_{(1)}^2 \nu_{(1)}^{(2)}}$$

$$- \frac{a_1 a_2 \left(d-2\right)\sin\left(\alpha_{(1)}^{(2)}\right)\cos\left(\alpha_{(1)}^{(2)}\right)}{4\pi\nu_{(1)}^2 \left(\nu_{(1)}^{(2)}\right)^2} + \frac{a_1 a_2 \left(d-1\right)\sin\left(\alpha_{(1)}^{(2)}\right)\cos^2\left(\alpha_{(1)}^{(2)}\right)}{4\pi\nu_{(1)}^2 \left(\nu_{(1)}^{(2)}\right)^2}$$

$$+ \frac{a_1 a_2 \sin\left(\alpha_{(1)}^{(2)}\right)}{4\pi\nu_{(1)}^2 \left(\nu_{(1)}^{(2)}\right)^2} + \frac{a_1 a_2 \left(d-1\right)\sin\left(\beta_{(1)}^{(2)}\right)}{4\pi\nu_{(1)}^3} + \frac{a_1 a_2 \sin\left(\beta_{(1)}^{(1)}\right)}{4\pi\nu_{(1)}^3}$$

$$- \frac{a_1 a_2 \left(d-1\right)\sin\left(\beta_{(1)}^{(2)}\right)\cos^2\left(\beta_{(1)}^{(2)}\right)}{4\pi\left(\mu_{(1)}^{(2)}\right)^2 \nu_{(1)}^3} - \frac{a_1 a_2 \sin\left(\beta_{(1)}^{(1)}\right)\cos^2\left(\beta_{(1)}^{(1)}\right)}{4\pi\left(\mu_{(1)}^{(1)}\right)^2 \nu_{(1)}^3}$$

$$+ \frac{a_1 \sin\left(\beta_{(1)}^{(2)}\right)\cos\left(\beta_{(1)}^{(2)}\right)}{4\pi\left(\mu_{(1)}^{(2)}\right)^2 \nu_{(1)}^2} + \frac{a_2^2 \cos\left(\alpha_{(1)}^{(2)}\right)}{4\pi\nu_{(1)}^2 \nu_{(1)}^{(2)}} + \frac{a_2^2 \left(d-2\right)\sin\left(\alpha_{(1)}^{(2)}\right)}{4\pi\nu_{(1)}^2 \left(\nu_{(1)}^{(2)}\right)^2}$$

$$- \frac{a_2^2 \left(d-1\right)\sin\left(\alpha_{(1)}^{(2)}\right)\cos\left(\alpha_{(1)}^{(2)}\right)}{4\pi\nu_{(1)}^2 \left(\nu_{(1)}^{(2)}\right)^2} + \frac{a_2 \sin\left(\beta_{(1)}^{(1)}\right)\cos\left(\beta_{(1)}^{(1)}\right)}{4\pi\left(\mu_{(1)}^{(1)}\right)^2 \nu_{(1)}^2} + \frac{\alpha_{(1)}^{(2)}}{4\pi} - \frac{1}{4},$$

$$T_{2,2}^{\mathfrak{s}} = -\frac{a_1^2}{2\pi\nu_{(1)}^2 \nu_{(1)}^{(2)}} + \frac{a_1^2 \sin\left(\alpha_{(1)}^{(2)}\right)}{2\pi\nu_{(1)}^2 \left(\nu_{(1)}^{(2)}\right)^2} + \frac{a_1 a_2 d \cos\left(\alpha_{(1)}^{(2)}\right)}{2\pi\nu_{(1)}^2 \nu_{(1)}^{(2)}} - \frac{a_1 a_2 d \sin\left(\alpha_{(1)}^{(2)}\right)\cos\left(\alpha_{(1)}^{(2)}\right)}{2\pi\nu_{(1)}^2 \left(\nu_{(1)}^{(2)}\right)^2}$$

$$- \frac{a_1 a_2 \left(d-2\right)}{2\pi\nu_{(1)}^2 \nu_{(1)}^{(2)}} + \frac{a_1 a_2 \left(d-2\right)\sin\left(\alpha_{(1)}^{(2)}\right)}{2\pi\nu_{(1)}^2 \left(\nu_{(1)}^{(2)}\right)^2} + \frac{a_2^2 d \sin\left(\beta_{(1)}^{(1)}\right)}{4\pi\nu_{(1)}^3}$$

$$- \frac{a_2^2 d \sin\left(\beta_{(1)}^{(1)}\right)\cos^2\left(\beta_{(1)}^{(1)}\right)}{4\pi\left(\mu_{(1)}^{(1)}\right)^2 \nu_{(1)}^3} - \frac{a_2^2 \left(d^2-d\right)\sin\left(\alpha_{(1)}^{(2)}\right)}{4\pi\nu_{(1)}^2} - \frac{a_2^2}{2\pi\nu_{(1)}^2 \nu_{(1)}^{(2)}}$$

$$- \frac{a_2^2 \left(d^2-2d\right)\sin\left(\alpha_{(1)}^{(2)}\right)\cos\left(\alpha_{(1)}^{(2)}\right)}{2\pi\nu_{(1)}^2 \left(\nu_{(1)}^{(2)}\right)^2} + \frac{a_2^2 \left(d^2-d\right)\sin\left(\alpha_{(1)}^{(2)}\right)\cos^2\left(\alpha_{(1)}^{(2)}\right)}{4\pi\nu_{(1)}^2 \left(\nu_{(1)}^{(2)}\right)^2}$$

$$+ \frac{a_2^2 \left(d^2-3d+2\right)\sin\left(\alpha_{(1)}^{(2)}\right)}{4\pi\nu_{(1)}^2 \left(\nu_{(1)}^{(2)}\right)^2} + \frac{a_2^2 \left(d^2-d\right)\sin\left(\beta_{(1)}^{(2)}\right)}{4\pi\nu_{(1)}^3}$$

$$- \frac{a_2^2 \left(d^2-d\right)\sin\left(\beta_{(1)}^{(2)}\right)\cos^2\left(\beta_{(1)}^{(2)}\right)}{4\pi\left(\mu_{(1)}^{(2)}\right)^2 \nu_{(1)}^3} + \frac{a_2 d \sin\left(\beta_{(1)}^{(2)}\right)\cos\left(\beta_{(1)}^{(2)}\right)}{2\pi\left(\mu_{(1)}^{(2)}\right)^2 \nu_{(1)}^2} - \frac{\alpha_{(1)}^{(2)} \left(d-2\right)}{4\pi}$$

$$+ \frac{d}{4} + \frac{\left(d-1\right)\sin\left(\alpha_{(1)}^{(2)}\right)}{2\pi} - \frac{\left(d-1\right)\sin\left(\beta_{(1)}^{(2)}\right)}{2\pi\nu_{(1)}} - \frac{\sin\left(\beta_{(1)}^{(1)}\right)}{2\pi\nu_{(1)}} - \frac{\sin\left(\beta_{(1)}^{(2)}\right)}{2\pi\left(\mu_{(1)}^{(2)}\right)^2 \nu_{(1)}},$$

$$T_{2,3}^{\mathfrak{s}} = -\frac{a_1 a_2 (d-2) \sin\left(\alpha_{(1)}^{(2)}\right) \cos\left(\alpha_{(1)}^{(2)}\right)}{2\pi \nu_{(1)}^2 \left(\nu_{(1)}^{(2)}\right)^2} + \frac{a_1 a_2 (d-2) \sin\left(\alpha_{(1)}^{(2)}\right)}{2\pi \nu_{(1)}^2 \left(\nu_{(1)}^{(2)}\right)^2}$$

$$-\frac{a_2^2 \left(d^2 - 3d + 2\right) \sin\left(\alpha_{(1)}^{(2)}\right)}{4\pi \nu_{(1)}^2} + \frac{a_2^2 (d-2) \cos\left(\alpha_{(1)}^{(2)}\right)}{2\pi \nu_{(1)}^2 \nu_{(1)}^{(2)}} - \frac{a_2^2 (d-2)}{2\pi \nu_{(1)}^2 \nu_{(1)}^{(2)}}$$

$$+\frac{a_2^2 \left(d^2 - 5d + 6\right) \sin\left(\alpha_{(1)}^{(2)}\right)}{4\pi \nu_{(1)}^2 \left(\nu_{(1)}^{(2)}\right)^2} - \frac{a_2^2 \left(d^2 - 4d + 4\right) \sin\left(\alpha_{(1)}^{(2)}\right) \cos\left(\alpha_{(1)}^{(2)}\right)}{2\pi \nu_{(1)}^2 \left(\nu_{(1)}^{(2)}\right)^2}$$

$$+\frac{a_2^2 \left(d^2 - 3d + 2\right) \sin\left(\alpha_{(1)}^{(2)}\right) \cos^2\left(\alpha_{(1)}^{(2)}\right)}{4\pi \nu_{(1)}^2 \left(\nu_{(1)}^{(2)}\right)^2}$$

$$+\frac{a_2^2 (d-2) \sin\left(\beta_{(1)}^{(1)}\right)}{4\pi \nu_{(1)}^3} + \frac{a_2^2 \left(d^2 - 3d + 2\right) \sin\left(\beta_{(1)}^{(2)}\right)}{4\pi \nu_{(1)}^3}$$

$$-\frac{a_2^2 \left(d^2 - 3d + 2\right) \sin\left(\beta_{(1)}^{(2)}\right) \cos^2\left(\beta_{(1)}^{(2)}\right)}{4\pi \left(\mu_{(1)}^{(2)}\right)^2 \nu_{(1)}^3} - \frac{a_2^2 (d-2) \sin\left(\beta_{(1)}^{(1)}\right) \cos^2\left(\beta_{(1)}^{(1)}\right)}{4\pi \left(\mu_{(1)}^{(1)}\right)^2 \nu_{(1)}^3}$$

$$+\frac{a_2 (d-2) \sin\left(\beta_{(1)}^{(2)}\right) \cos\left(\beta_{(1)}^{(2)}\right)}{2\pi \left(\mu_{(1)}^{(2)}\right)^2 \nu_{(1)}^2} + \frac{\alpha_{(1)}^{(2)} (d-2)}{4\pi} - \frac{d}{4} + \frac{1}{2},$$

$$T_{2,4}^{\mathfrak{s}} = -\frac{a_2 \alpha_{(1)}^{(2)} (d-2)}{4\pi} + \frac{a_2 d}{4} + \frac{a_2 (d-2) \sin\left(\alpha_{(1)}^{(2)}\right)}{4\pi}$$

$$-\frac{a_2 (d-1) \sin\left(\beta_{(1)}^{(2)}\right)}{4\pi \nu_{(1)}} - \frac{a_2 \sin\left(\beta_{(1)}^{(1)}\right)}{4\pi \nu_{(1)}} + \frac{\beta_{(1)}^{(2)}}{4\pi} - \frac{1}{4},$$

$$T_{3,1}^{\mathfrak{s}} = -\frac{a_1^2 \cos\left(\alpha_{(1)}^{(2)}\right)}{4\pi \nu_{(1)}^2 \nu_{(1)}^{(2)}} - \frac{a_1^2 \sin\left(\alpha_{(1)}^{(2)}\right) \cos\left(\alpha_{(1)}^{(2)}\right)}{4\pi \nu_{(1)}^2 \left(\nu_{(1)}^{(2)}\right)^2} - \frac{a_1 a_2 (d-1) \sin\left(\alpha_{(1)}^{(2)}\right)}{4\pi \nu_{(1)}^2} + \frac{a_1 a_2 \cos\left(\alpha_{(1)}^{(2)}\right)}{2\pi \nu_{(1)}^2 \nu_{(1)}^{(2)}}$$

$$-\frac{a_1 a_2 (d-2) \sin\left(\alpha_{(1)}^{(2)}\right) \cos\left(\alpha_{(1)}^{(2)}\right)}{4\pi \nu_{(1)}^2 \left(\nu_{(1)}^{(2)}\right)^2} + \frac{a_1 a_2 (d-1) \sin\left(\alpha_{(1)}^{(2)}\right) \cos^2\left(\alpha_{(1)}^{(2)}\right)}{4\pi \nu_{(1)}^2 \left(\nu_{(1)}^{(2)}\right)^2}$$

$$+\frac{a_1 a_2 \sin\left(\alpha_{(1)}^{(2)}\right)}{4\pi \nu_{(1)}^2 \left(\nu_{(1)}^{(2)}\right)^2} + \frac{a_1 a_2 (d-1) \sin\left(\beta_{(1)}^{(2)}\right)}{4\pi \nu_{(1)}^3} + \frac{a_1 a_2 \sin\left(\beta_{(1)}^{(1)}\right)}{4\pi \nu_{(1)}^3}$$

$$-\frac{a_1 a_2 (d-1) \sin\left(\beta_{(1)}^{(2)}\right) \cos^2\left(\beta_{(1)}^{(2)}\right)}{4\pi \left(\mu_{(1)}^{(2)}\right)^2 \nu_{(1)}^3} - \frac{a_1 a_2 \sin\left(\beta_{(1)}^{(1)}\right) \cos^2\left(\beta_{(1)}^{(1)}\right)}{4\pi \left(\mu_{(1)}^{(1)}\right)^2 \nu_{(1)}^3}$$

$$+\frac{a_1 \sin\left(\beta_{(1)}^{(2)}\right) \cos\left(\beta_{(1)}^{(2)}\right)}{4\pi \left(\mu_{(1)}^{(2)}\right)^2 \nu_{(1)}^2} + \frac{a_2^2 \cos\left(\alpha_{(1)}^{(2)}\right)}{4\pi \nu_{(1)}^2 \nu_{(1)}^{(2)}} - \frac{a_2^2}{2\pi \nu_{(1)}^2 \nu_{(1)}^{(2)}} + \frac{a_2^2 (d-2) \sin\left(\alpha_{(1)}^{(2)}\right)}{4\pi \nu_{(1)}^2 \left(\nu_{(1)}^{(2)}\right)^2}$$

$$-\frac{a_2^2 (d-1) \sin\left(\alpha_{(1)}^{(2)}\right) \cos\left(\alpha_{(1)}^{(2)}\right)}{4\pi \nu_{(1)}^2 \left(\nu_{(1)}^{(2)}\right)^2} + \frac{a_2 \sin\left(\beta_{(1)}^{(1)}\right) \cos\left(\beta_{(1)}^{(1)}\right)}{4\pi \left(\mu_{(1)}^{(1)}\right)^2 \nu_{(1)}^2} - \frac{\alpha_{(1)}^{(2)}}{4\pi} + \frac{1}{4},$$

$$T_{3,2}^{\mathfrak{s}} = -\frac{a_1 a_2 d \sin\left(\alpha_{(1)}^{(2)}\right) \cos\left(\alpha_{(1)}^{(2)}\right)}{2\pi \nu_{(1)}^2 \left(\nu_{(1)}^{(2)}\right)^2} + \frac{a_1 a_2 d \sin\left(\alpha_{(1)}^{(2)}\right)}{2\pi \nu_{(1)}^2 \left(\nu_{(1)}^{(2)}\right)^2} + \frac{a_2^2 d \cos\left(\alpha_{(1)}^{(2)}\right)}{2\pi \nu_{(1)}^2 \nu_{(1)}^{(2)}} - \frac{a_2^2 d}{2\pi \nu_{(1)}^2 \nu_{(1)}^{(2)}}$$

$$+ \frac{a_2^2 d \sin\left(\beta_{(1)}^{(1)}\right)}{4\pi \nu_{(1)}^3} - \frac{a_2^2 d \sin\left(\beta_{(1)}^{(1)}\right) \cos^2\left(\beta_{(1)}^{(1)}\right)}{4\pi \left(\mu_{(1)}^{(1)}\right)^2 \nu_{(1)}^3} - \frac{a_2^2 \left(d^2 - d\right) \sin\left(\alpha_{(1)}^{(2)}\right)}{4\pi \nu_{(1)}^2}$$

$$+ \frac{a_2^2 \left(d^2 - 3d\right) \sin\left(\alpha_{(1)}^{(2)}\right)}{4\pi \nu_{(1)}^2 \left(\nu_{(1)}^{(2)}\right)^2} - \frac{a_2^2 \left(d^2 - 2d\right) \sin\left(\alpha_{(1)}^{(2)}\right) \cos\left(\alpha_{(1)}^{(2)}\right)}{2\pi \nu_{(1)}^2 \left(\nu_{(1)}^{(2)}\right)^2}$$

$$+ \frac{a_2^2 \left(d^2 - d\right) \sin\left(\alpha_{(1)}^{(2)}\right) \cos^2\left(\alpha_{(1)}^{(2)}\right)}{4\pi \nu_{(1)}^2 \left(\nu_{(1)}^{(2)}\right)^2} + \frac{a_2^2 \left(d^2 - d\right) \sin\left(\beta_{(1)}^{(2)}\right)}{4\pi \nu_{(1)}^3}$$

$$- \frac{a_2^2 \left(d^2 - d\right) \sin\left(\beta_{(1)}^{(2)}\right) \cos^2\left(\beta_{(1)}^{(2)}\right)}{4\pi \left(\mu_{(1)}^{(2)}\right)^2 \nu_{(1)}^3} + \frac{a_2 d \sin\left(\beta_{(1)}^{(2)}\right) \cos\left(\beta_{(1)}^{(2)}\right)}{2\pi \left(\mu_{(1)}^{(2)}\right)^2 \nu_{(1)}^2} + \frac{\alpha_{(1)}^{(2)} d}{4\pi} - \frac{d}{4},$$

$$T_{3,3}^{\mathfrak{s}} = \frac{a_1^2}{2\pi \nu_{(1)}^2 \nu_{(1)}^{(2)}} + \frac{a_1^2 \sin\left(\alpha_{(1)}^{(2)}\right)}{2\pi \nu_{(1)}^2 \left(\nu_{(1)}^{(2)}\right)^2} + \frac{a_1 a_2 \left(d - 4\right)}{2\pi \nu_{(1)}^2 \nu_{(1)}^{(2)}} - \frac{a_1 a_2 \left(d - 2\right) \cos\left(\alpha_{(1)}^{(2)}\right)}{2\pi \nu_{(1)}^2 \nu_{(1)}^{(2)}}$$

$$+ \frac{a_1 a_2 \left(d - 4\right) \sin\left(\alpha_{(1)}^{(2)}\right)}{2\pi \nu_{(1)}^2 \left(\nu_{(1)}^{(2)}\right)^2} - \frac{a_1 a_2 \left(d - 2\right) \sin\left(\alpha_{(1)}^{(2)}\right) \cos\left(\alpha_{(1)}^{(2)}\right)}{2\pi \nu_{(1)}^2 \left(\nu_{(1)}^{(2)}\right)^2}$$

$$- \frac{a_2^2 \left(d^2 - 3d + 2\right) \sin\left(\alpha_{(1)}^{(2)}\right)}{4\pi \nu_{(1)}^2} - \frac{a_2^2 \left(d - \frac{5}{2}\right)}{\pi \nu_{(1)}^2 \nu_{(1)}^{(2)}} + \frac{a_2^2 \left(d - 2\right) \cos\left(\alpha_{(1)}^{(2)}\right)}{\pi \nu_{(1)}^2 \nu_{(1)}^{(2)}}$$

$$+ \frac{a_2^2 \left(d^2 - 5d + 8\right) \sin\left(\alpha_{(1)}^{(2)}\right)}{4\pi \nu_{(1)}^2 \left(\nu_{(1)}^{(2)}\right)^2} - \frac{a_2^2 \left(d^2 - 4d + 4\right) \sin\left(\alpha_{(1)}^{(2)}\right) \cos\left(\alpha_{(1)}^{(2)}\right)}{2\pi \nu_{(1)}^2 \left(\nu_{(1)}^{(2)}\right)^2}$$

$$+ \frac{a_2^2 \left(d^2 - 3d + 2\right) \sin\left(\alpha_{(1)}^{(2)}\right) \cos^2\left(\alpha_{(1)}^{(2)}\right)}{4\pi \nu_{(1)}^2 \left(\nu_{(1)}^{(2)}\right)^2} + \frac{a_2^2 \left(d - 2\right) \sin\left(\beta_{(1)}^{(1)}\right)}{4\pi \nu_{(1)}^3}$$

$$+ \frac{a_2^2 \left(d^2 - 3d + 2\right) \sin\left(\beta_{(1)}^{(2)}\right)}{4\pi \nu_{(1)}^3} - \frac{a_2^2 \left(d^2 - 3d + 2\right) \sin\left(\beta_{(1)}^{(2)}\right) \cos^2\left(\beta_{(1)}^{(2)}\right)}{4\pi \left(\mu_{(1)}^{(2)}\right)^2 \nu_{(1)}^3}$$

$$- \frac{a_2^2 \left(d - 2\right) \sin\left(\beta_{(1)}^{(1)}\right) \cos^2\left(\beta_{(1)}^{(1)}\right)}{4\pi \left(\mu_{(1)}^{(1)}\right)^2 \nu_{(1)}^3} + \frac{a_2 \left(d - 2\right) \sin\left(\beta_{(1)}^{(2)}\right) \cos\left(\beta_{(1)}^{(2)}\right)}{2\pi \left(\mu_{(1)}^{(2)}\right)^2 \nu_{(1)}^2} - \frac{\alpha_{(1)}^{(2)} \left(d - 4\right)}{4\pi}$$

$$+ \frac{d}{4} + \frac{\left(d - 1\right) \sin\left(\alpha_{(1)}^{(2)}\right)}{2\pi} - \frac{1}{2} - \frac{\left(d - 1\right) \sin\left(\beta_{(1)}^{(2)}\right)}{2\pi \nu_{(1)}} - \frac{\sin\left(\beta_{(1)}^{(1)}\right)}{2\pi \nu_{(1)}} - \frac{\sin\left(\beta_{(1)}^{(2)}\right)}{2\pi \left(\mu_{(1)}^{(2)}\right)^2 \nu_{(1)}},$$

$$T_{3,4}^{\mathfrak{s}} = -\frac{a_1 \alpha_{(1)}^{(2)}}{2\pi} + \frac{a_1}{2} - \frac{a_2 \alpha_{(1)}^{(2)} \left(d - 4\right)}{4\pi} + \frac{a_2 \left(d - 2\right) \sin\left(\alpha_{(1)}^{(2)}\right)}{4\pi}$$

$$+ \frac{a_2 \left(d - 2\right)}{4} - \frac{a_2 \left(d - 1\right) \sin\left(\beta_{(1)}^{(2)}\right)}{4\pi \nu_{(1)}} - \frac{a_2 \sin\left(\beta_{(1)}^{(1)}\right)}{4\pi \nu_{(1)}} + \frac{\beta_{(1)}^{(2)}}{4\pi} - \frac{1}{4},$$

$$T_{4,1}^{\mathsf{s}} = \frac{a_1\,(d-2)\sin\left(\alpha_{(1)}^{(2)}\right)}{2\pi} + a_1 - \frac{a_1\,(d-1)\sin\left(\beta_{(1)}^{(2)}\right)}{2\pi\nu_{(1)}}$$

$$- \frac{a_1\sin\left(\beta_{(1)}^{(1)}\right)}{2\pi\nu_{(1)}} - \frac{a_2\alpha_{(1)}^{(2)}\,(d-2)}{2\pi} + \frac{a_2\,(d-2)}{2} + \frac{\beta_{(1)}^{(1)}}{2\pi} - \frac{1}{2},$$

$$T_{4,2}^{\mathsf{s}} = -\frac{a_2\alpha_{(1)}^{(2)}\left(d^2-2d\right)}{2\pi} + \frac{a_2 d^2}{2} - \frac{a_2 d\sin\left(\beta_{(1)}^{(1)}\right)}{2\pi\nu_{(1)}}$$

$$+ \frac{a_2\left(d^2-2d\right)\sin\left(\alpha_{(1)}^{(2)}\right)}{2\pi} - \frac{a_2\left(d^2-d\right)\sin\left(\beta_{(1)}^{(2)}\right)}{2\pi\nu_{(1)}} + \frac{\beta_{(1)}^{(2)} d}{2\pi} - \frac{d}{2},$$

$$T_{4,3}^{\mathsf{s}} = -\frac{a_1\alpha_{(1)}^{(2)}\,(d-2)}{\pi} + a_1\,(d-2) - \frac{a_2\alpha_{(1)}^{(2)}\left(d^2-6d+8\right)}{2\pi}$$

$$+ \frac{a_2\left(d^2-4d+4\right)\sin\left(\alpha_{(1)}^{(2)}\right)}{2\pi} + \frac{a_2\left(d^2-4d+4\right)}{2} - \frac{a_2\,(d-2)\sin\left(\beta_{(1)}^{(1)}\right)}{2\pi\nu_{(1)}}$$

$$- \frac{a_2\left(d^2-3d+2\right)\sin\left(\beta_{(1)}^{(2)}\right)}{2\pi\nu_{(1)}} + \frac{\beta_{(1)}^{(2)}\,(d-2)}{2\pi} - \frac{d}{2} + 1,$$

$$T_{4,4}^{\mathsf{s}} = \frac{\alpha_{(1)}^{(2)}\nu_{(1)}^2\cos\left(\alpha_{(1)}^{(2)}\right)}{2\pi} - \frac{\nu_{(1)}^2\sin\left(\alpha_{(1)}^{(2)}\right)}{2\pi} - \frac{\nu_{(1)}^2\cos\left(\alpha_{(1)}^{(2)}\right)}{2} + \frac{\nu_{(1)}^2}{2}.$$

**t-rep.** Let the associated $3\times 3$ transition matrix be denoted by $T^{\mathsf{t}}$. Then,

$$T_{1,1}^{\mathsf{t}} = -\frac{a_1^2\,(d-1)\sin\left(\alpha_{(1)}^{(2)}\right)}{2\pi\nu_{(1)}^2} + \frac{a_1^2\,(d-1)}{2\pi\nu_{(1)}^2\nu_{(1)}^{(2)}} + \frac{a_1^2\,(d-1)\sin\left(\alpha_{(1)}^{(2)}\right)\cos^2\left(\alpha_{(1)}^{(2)}\right)}{2\pi\nu_{(1)}^2\left(\nu_{(1)}^{(2)}\right)^2}$$

$$+ \frac{a_1^2\,(d-1)\sin\left(\beta_{(1)}^{(2)}\right)}{2\pi\nu_{(1)}^3} + \frac{a_1^2\sin\left(\beta_{(1)}^{(1)}\right)}{2\pi\nu_{(1)}^3} - \frac{a_1^2\,(d-1)\sin\left(\beta_{(1)}^{(2)}\right)\cos^2\left(\beta_{(1)}^{(2)}\right)}{2\pi\left(\mu_{(1)}^{(2)}\right)^2\nu_{(1)}^3}$$

$$- \frac{a_1^2\sin\left(\beta_{(1)}^{(1)}\right)\cos^2\left(\beta_{(1)}^{(1)}\right)}{2\pi\left(\mu_{(1)}^{(1)}\right)^2\nu_{(1)}^3} - \frac{a_1 a_2\,(d-1)\cos\left(\alpha_{(1)}^{(2)}\right)}{\pi\nu_{(1)}^2\nu_{(1)}^{(2)}}$$

$$- \frac{a_1 a_2\,(d-1)\sin\left(\alpha_{(1)}^{(2)}\right)\cos\left(\alpha_{(1)}^{(2)}\right)}{\pi\nu_{(1)}^2\left(\nu_{(1)}^{(2)}\right)^2} + \frac{a_1\sin\left(\beta_{(1)}^{(1)}\right)\cos\left(\beta_{(1)}^{(1)}\right)}{\pi\left(\mu_{(1)}^{(1)}\right)^2\nu_{(1)}^2}$$

$$+ \frac{a_2^2\,(d-1)}{2\pi\nu_{(1)}^2\nu_{(1)}^{(2)}} + \frac{a_2^2\,(d-1)\sin\left(\alpha_{(1)}^{(2)}\right)}{2\pi\nu_{(1)}^2\left(\nu_{(1)}^{(2)}\right)^2} + \frac{(d-1)\sin\left(\alpha_{(1)}^{(2)}\right)}{2\pi}$$

$$+ \frac{1}{2} - \frac{(d-1)\sin\left(\beta_{(1)}^{(2)}\right)}{2\pi\nu_{(1)}} - \frac{\sin\left(\beta_{(1)}^{(1)}\right)}{2\pi\nu_{(1)}} - \frac{\sin\left(\beta_{(1)}^{(1)}\right)}{2\pi\left(\mu_{(1)}^{(1)}\right)^2\nu_{(1)}},$$

$$T_{1,2}^{\mathrm{t}} = -\frac{a_1^2\,(d-1)\cos\left(\alpha_{(1)}^{(2)}\right)}{2\pi\nu_{(1)}^2\nu_{(1)}^{(2)}} - \frac{a_1^2\,(d-1)\sin\left(\alpha_{(1)}^{(2)}\right)\cos\left(\alpha_{(1)}^{(2)}\right)}{2\pi\nu_{(1)}^2\left(\nu_{(1)}^{(2)}\right)^2}$$

$$-\frac{a_1 a_2\,(d^2-2d+1)\sin\left(\alpha_{(1)}^{(2)}\right)}{2\pi\nu_{(1)}^2} + \frac{a_1 a_2\,(d^2-d)}{2\pi\nu_{(1)}^2\nu_{(1)}^{(2)}} - \frac{a_1 a_2\,(d^2-3d+2)\cos\left(\alpha_{(1)}^{(2)}\right)}{2\pi\nu_{(1)}^2\nu_{(1)}^{(2)}}$$

$$+\frac{a_1 a_2\,(d-1)\sin\left(\alpha_{(1)}^{(2)}\right)}{2\pi\nu_{(1)}^2\left(\nu_{(1)}^{(2)}\right)^2} - \frac{a_1 a_2\,(d^2-3d+2)\sin\left(\alpha_{(1)}^{(2)}\right)\cos\left(\alpha_{(1)}^{(2)}\right)}{2\pi\nu_{(1)}^2\left(\nu_{(1)}^{(2)}\right)^2}$$

$$+\frac{a_1 a_2\,(d^2-2d+1)\sin\left(\alpha_{(1)}^{(2)}\right)\cos^2\left(\alpha_{(1)}^{(2)}\right)}{2\pi\nu_{(1)}^2\left(\nu_{(1)}^{(2)}\right)^2} + \frac{a_1 a_2\,(d-1)\sin\left(\beta_{(1)}^{(1)}\right)}{2\pi\nu_{(1)}^3}$$

$$+\frac{a_1 a_2\,(d^2-2d+1)\sin\left(\beta_{(1)}^{(2)}\right)}{2\pi\nu_{(1)}^3} - \frac{a_1 a_2\,(d^2-2d+1)\sin\left(\beta_{(1)}^{(2)}\right)\cos^2\left(\beta_{(1)}^{(2)}\right)}{2\pi\left(\mu_{(1)}^{(2)}\right)^2\nu_{(1)}^3}$$

$$-\frac{a_1 a_2\,(d-1)\sin\left(\beta_{(1)}^{(1)}\right)\cos^2\left(\beta_{(1)}^{(1)}\right)}{2\pi\left(\mu_{(1)}^{(1)}\right)^2\nu_{(1)}^3} + \frac{a_1\,(d-1)\sin\left(\beta_{(1)}^{(2)}\right)\cos\left(\beta_{(1)}^{(2)}\right)}{2\pi\left(\mu_{(1)}^{(2)}\right)^2\nu_{(1)}^2}$$

$$+\frac{a_2^2\,(d^2-3d+2)}{2\pi\nu_{(1)}^2\nu_{(1)}^{(2)}} - \frac{a_2^2\,(d^2-2d+1)\cos\left(\alpha_{(1)}^{(2)}\right)}{2\pi\nu_{(1)}^2\nu_{(1)}^{(2)}}$$

$$+\frac{a_2^2\,(d^2-3d+2)\sin\left(\alpha_{(1)}^{(2)}\right)}{2\pi\nu_{(1)}^2\left(\nu_{(1)}^{(2)}\right)^2} - \frac{a_2^2\,(d^2-2d+1)\sin\left(\alpha_{(1)}^{(2)}\right)\cos\left(\alpha_{(1)}^{(2)}\right)}{2\pi\nu_{(1)}^2\left(\nu_{(1)}^{(2)}\right)^2}$$

$$+\frac{a_2\,(d-1)\sin\left(\beta_{(1)}^{(1)}\right)\cos\left(\beta_{(1)}^{(1)}\right)}{2\pi\left(\mu_{(1)}^{(1)}\right)^2\nu_{(1)}^2} - \frac{\alpha_{(1)}^{(2)}\,(d-1)}{2\pi} + \frac{d}{2} - \frac{1}{2},$$

$$T_{1,3}^{\mathrm{t}} = \frac{a_1\,(d-1)\sin\left(\alpha_{(1)}^{(2)}\right)}{\pi} + a_1 - \frac{a_1\,(d-1)\sin\left(\beta_{(1)}^{(2)}\right)}{2\pi\nu_{(1)}}$$

$$-\frac{a_1\sin\left(\beta_{(1)}^{(1)}\right)}{2\pi\nu_{(1)}} - \frac{a_2\alpha_{(1)}^{(2)}\,(d-1)}{\pi} + a_2\,(d-1) + \frac{\beta_{(1)}^{(1)}}{2\pi} - \frac{1}{2},$$

$$T_{2,1}^{\mathsf{t}} = -\frac{a_1^2 \cos\left(\alpha_{(1)}^{(2)}\right)}{2\pi\nu_{(1)}^2 \nu_{(1)}^{(2)}} - \frac{a_1^2 \sin\left(\alpha_{(1)}^{(2)}\right)\cos\left(\alpha_{(1)}^{(2)}\right)}{2\pi\nu_{(1)}^2 \left(\nu_{(1)}^{(2)}\right)^2} + \frac{a_1 a_2 d}{2\pi\nu_{(1)}^2 \nu_{(1)}^{(2)}} - \frac{a_1 a_2 (d-1)\sin\left(\alpha_{(1)}^{(2)}\right)}{2\pi\nu_{(1)}^2}$$

$$- \frac{a_1 a_2 (d-2)\cos\left(\alpha_{(1)}^{(2)}\right)}{2\pi\nu_{(1)}^2 \nu_{(1)}^{(2)}} - \frac{a_1 a_2 (d-2)\sin\left(\alpha_{(1)}^{(2)}\right)\cos\left(\alpha_{(1)}^{(2)}\right)}{2\pi\nu_{(1)}^2 \left(\nu_{(1)}^{(2)}\right)^2}$$

$$+ \frac{a_1 a_2 (d-1)\sin\left(\alpha_{(1)}^{(2)}\right)\cos^2\left(\alpha_{(1)}^{(2)}\right)}{2\pi\nu_{(1)}^2 \left(\nu_{(1)}^{(2)}\right)^2} + \frac{a_1 a_2 \sin\left(\alpha_{(1)}^{(2)}\right)}{2\pi\nu_{(1)}^2 \left(\nu_{(1)}^{(2)}\right)^2} + \frac{a_1 a_2 (d-1)\sin\left(\beta_{(1)}^{(2)}\right)}{2\pi\nu_{(1)}^3}$$

$$+ \frac{a_1 a_2 \sin\left(\beta_{(1)}^{(1)}\right)}{2\pi\nu_{(1)}^3} - \frac{a_1 a_2 (d-1)\sin\left(\beta_{(1)}^{(2)}\right)\cos^2\left(\beta_{(1)}^{(2)}\right)}{2\pi\left(\mu_{(1)}^{(2)}\right)^2 \nu_{(1)}^3} - \frac{a_1 a_2 \sin\left(\beta_{(1)}^{(1)}\right)\cos^2\left(\beta_{(1)}^{(1)}\right)}{2\pi\left(\mu_{(1)}^{(1)}\right)^2 \nu_{(1)}^3}$$

$$+ \frac{a_1 \sin\left(\beta_{(1)}^{(2)}\right)\cos\left(\beta_{(1)}^{(2)}\right)}{2\pi\left(\mu_{(1)}^{(2)}\right)^2 \nu_{(1)}^2} + \frac{a_2^2 (d-2)}{2\pi\nu_{(1)}^2 \nu_{(1)}^{(2)}} - \frac{a_2^2 (d-1)\cos\left(\alpha_{(1)}^{(2)}\right)}{2\pi\nu_{(1)}^2 \nu_{(1)}^{(2)}} + \frac{a_2^2 (d-2)\sin\left(\alpha_{(1)}^{(2)}\right)}{2\pi\nu_{(1)}^2 \left(\nu_{(1)}^{(2)}\right)^2}$$

$$- \frac{a_2^2 (d-1)\sin\left(\alpha_{(1)}^{(2)}\right)\cos\left(\alpha_{(1)}^{(2)}\right)}{2\pi\nu_{(1)}^2 \left(\nu_{(1)}^{(2)}\right)^2} + \frac{a_2 \sin\left(\beta_{(1)}^{(1)}\right)\cos\left(\beta_{(1)}^{(1)}\right)}{2\pi\left(\mu_{(1)}^{(1)}\right)^2 \nu_{(1)}^2} - \frac{\alpha_{(1)}^{(2)}}{2\pi} + \frac{1}{2},$$

$$T_{2,2}^{\mathsf{t}} = \frac{a_1^2}{2\pi\nu_{(1)}^2 \nu_{(1)}^{(2)}} + \frac{a_1^2 \sin\left(\alpha_{(1)}^{(2)}\right)}{2\pi\nu_{(1)}^2 \left(\nu_{(1)}^{(2)}\right)^2} + \frac{a_1 a_2 (d-2)}{\pi\nu_{(1)}^2 \nu_{(1)}^{(2)}} - \frac{a_1 a_2 (d-1)\cos\left(\alpha_{(1)}^{(2)}\right)}{\pi\nu_{(1)}^2 \nu_{(1)}^{(2)}}$$

$$+ \frac{a_1 a_2 (d-2)\sin\left(\alpha_{(1)}^{(2)}\right)}{\pi\nu_{(1)}^2 \left(\nu_{(1)}^{(2)}\right)^2} - \frac{a_1 a_2 (d-1)\sin\left(\alpha_{(1)}^{(2)}\right)\cos\left(\alpha_{(1)}^{(2)}\right)}{\pi\nu_{(1)}^2 \left(\nu_{(1)}^{(2)}\right)^2}$$

$$- \frac{a_2^2 \left(d^2-2d+1\right)\sin\left(\alpha_{(1)}^{(2)}\right)}{2\pi\nu_{(1)}^2} - \frac{a_2^2 \left(d^2-3d+2\right)\cos\left(\alpha_{(1)}^{(2)}\right)}{\pi\nu_{(1)}^2 \nu_{(1)}^{(2)}} + \frac{a_2^2 \left(d^2-3d+\frac{5}{2}\right)}{\pi\nu_{(1)}^2 \nu_{(1)}^{(2)}}$$

$$+ \frac{a_2^2 \left(d^2-4d+4\right)\sin\left(\alpha_{(1)}^{(2)}\right)}{2\pi\nu_{(1)}^2 \left(\nu_{(1)}^{(2)}\right)^2} - \frac{a_2^2 \left(d^2-3d+2\right)\sin\left(\alpha_{(1)}^{(2)}\right)\cos\left(\alpha_{(1)}^{(2)}\right)}{\pi\nu_{(1)}^2 \left(\nu_{(1)}^{(2)}\right)^2}$$

$$+ \frac{a_2^2 \left(d^2-2d+1\right)\sin\left(\alpha_{(1)}^{(2)}\right)\cos^2\left(\alpha_{(1)}^{(2)}\right)}{2\pi\nu_{(1)}^2 \left(\nu_{(1)}^{(2)}\right)^2} + \frac{a_2^2 (d-1)\sin\left(\beta_{(1)}^{(1)}\right)}{2\pi\nu_{(1)}^3}$$

$$+ \frac{a_2^2 \left(d^2-2d+1\right)\sin\left(\beta_{(1)}^{(2)}\right)}{2\pi\nu_{(1)}^3} - \frac{a_2^2 \left(d^2-2d+1\right)\sin\left(\beta_{(1)}^{(2)}\right)\cos^2\left(\beta_{(1)}^{(2)}\right)}{2\pi\left(\mu_{(1)}^{(2)}\right)^2 \nu_{(1)}^3}$$

$$- \frac{a_2^2 (d-1)\sin\left(\beta_{(1)}^{(1)}\right)\cos^2\left(\beta_{(1)}^{(1)}\right)}{2\pi\left(\mu_{(1)}^{(1)}\right)^2 \nu_{(1)}^3} + \frac{a_2 (d-1)\sin\left(\beta_{(1)}^{(2)}\right)\cos\left(\beta_{(1)}^{(2)}\right)}{\pi\left(\mu_{(1)}^{(2)}\right)^2 \nu_{(1)}^2} - \frac{\alpha_{(1)}^{(2)} (d-2)}{2\pi}$$

$$+ \frac{d}{2} + \frac{(d-1)\sin\left(\alpha_{(1)}^{(2)}\right)}{2\pi} - \frac{1}{2} - \frac{(d-1)\sin\left(\beta_{(1)}^{(2)}\right)}{2\pi\nu_{(1)}} - \frac{\sin\left(\beta_{(1)}^{(1)}\right)}{2\pi\nu_{(1)}} - \frac{\sin\left(\beta_{(1)}^{(2)}\right)}{2\pi\left(\mu_{(1)}^{(2)}\right)^2 \nu_{(1)}},$$

$$T_{2,3}^{\mathsf{t}} = -\frac{a_1 \alpha_{(1)}^{(2)}}{\pi} + a_1 - \frac{a_2 \alpha_{(1)}^{(2)} (d-2)}{\pi} + \frac{a_2 (d-1) \sin\left(\alpha_{(1)}^{(2)}\right)}{\pi}$$
$$+ a_2 (d-1) - \frac{a_2 (d-1) \sin\left(\beta_{(1)}^{(2)}\right)}{2\pi\nu_{(1)}} - \frac{a_2 \sin\left(\beta_{(1)}^{(1)}\right)}{2\pi\nu_{(1)}} + \frac{\beta_{(1)}^{(2)}}{2\pi} - \frac{1}{2},$$

$$T_{3,1}^{\mathsf{t}} = \frac{a_1 (d-1) \sin\left(\alpha_{(1)}^{(2)}\right)}{\pi} + a_1 - \frac{a_1 (d-1) \sin\left(\beta_{(1)}^{(2)}\right)}{2\pi\nu_{(1)}}$$
$$- \frac{a_1 \sin\left(\beta_{(1)}^{(1)}\right)}{2\pi\nu_{(1)}} - \frac{a_2 \alpha_{(1)}^{(2)} (d-1)}{\pi} + a_2 (d-1) + \frac{\beta_{(1)}^{(1)}}{2\pi} - \frac{1}{2},$$

$$T_{3,2}^{\mathsf{t}} = -\frac{a_1 \alpha_{(1)}^{(2)} (d-1)}{\pi} + a_1 (d-1) - \frac{a_2 \alpha_{(1)}^{(2)} \left(d^2 - 3d + 2\right)}{\pi}$$
$$+ \frac{a_2 \left(d^2 - 2d + 1\right) \sin\left(\alpha_{(1)}^{(2)}\right)}{\pi} + a_2 \left(d^2 - 2d + 1\right) - \frac{a_2 (d-1) \sin\left(\beta_{(1)}^{(1)}\right)}{2\pi\nu_{(1)}}$$
$$- \frac{a_2 \left(d^2 - 2d + 1\right) \sin\left(\beta_{(1)}^{(2)}\right)}{2\pi\nu_{(1)}} + \frac{\beta_{(1)}^{(2)} (d-1)}{2\pi} - \frac{d}{2} + \frac{1}{2},$$

$$T_{3,3}^{\mathsf{t}} = -\frac{\alpha_{(1)}^{(2)} \nu_{(1)}^2 (d-1) \cos\left(\alpha_{(1)}^{(2)}\right)}{2\pi} + \frac{\nu_{(1)}^2 (d-1) \sin\left(\alpha_{(1)}^{(2)}\right)}{2\pi} + \frac{\nu_{(1)}^2 (d-1) \cos\left(\alpha_{(1)}^{(2)}\right)}{2} + \frac{\nu_{(1)}^2}{2}.$$

# F   Power series and Hessian spectrum by representation

We present the (fractional) power series of the minima described in Theorem 1 to $O(d^{-5/2})$-order, along with the respective eigenvalues arranged by their representation.

## F.1   Theorem 1, case 2a

$$a_1 = 1 + \frac{8}{\pi d^2} + O\left(d^{\frac{-5}{2}}\right),$$
$$a_2 = -\frac{4}{\pi d^2} + O\left(d^{\frac{-5}{2}}\right),$$
$$a_3 = \frac{2}{d} + \frac{-\frac{8}{\pi} - 2}{d^2} + O\left(d^{\frac{-5}{2}}\right),$$
$$a_4 = \frac{4}{\pi d} + \frac{32}{\pi^3 d^{\frac{3}{2}}} + \frac{8\left(-7\pi^3 - 8\pi^2 + 64\right)}{\pi^5 d^2} + O\left(d^{\frac{-5}{2}}\right),$$
$$a_5 = -1 + \frac{\frac{8}{\pi^2} + 2 + \frac{8}{\pi}}{d} + -\frac{64\left(12 - \pi\right)}{3\pi^4 d^{\frac{3}{2}} \left(-2 + \pi\right)} +$$
$$- \frac{2\left(-128\pi^3 - 40\pi^4 - 224\pi^2 - 512\pi + 2560 + \pi^7 + 10\pi^6 + 52\pi^5\right)}{\pi^6 d^2 \left(-2 + \pi\right)} + O\left(d^{\frac{-5}{2}}\right).$$

## F.2   Eigenvalues

| | | | | | | | | | |
|---|---|---|---|---|---|---|---|---|---|
| ɼ-Representation | $\frac{(d-2)(d-3)}{2}$ | $\frac{-2+\pi}{4\pi}$ | | | | | | | |
| ɧ-Representation | $\frac{(d-1)(d-4)}{2}$ | $\frac{2+\pi}{4\pi}$ | | | | | | | |
| Standard Representation | $d-2$ | $0$ | $\frac{-2+\pi}{4\pi}$ | $\frac{-2+\pi}{2\pi}$ | $\frac{1}{4}$ | $\frac{2+\pi}{4\pi}$ | | $\frac{d}{4} + \frac{1}{2}$ | |
| Trivial Representation | $1$ | $0$ | $0$ | $\frac{-2+\pi}{2\pi}$ | $\frac{1}{4}$ | $\frac{d}{4} + \frac{-4+\pi+\pi^2}{-8\pi+2\pi^2}$ | $\frac{d}{4} + \frac{1}{2}$ | $\frac{d}{\pi} + \frac{-10\pi+8+\pi^2}{2\pi(-4+\pi)}$ |

## F.3 Theorem 1, case 2b

$$a_1 = -1 + \frac{2}{d} + -\frac{4\left(-16\pi^2 + (-2 + \pi)\left(-6\pi^3 + 16\pi + 8\pi^2 + \pi^4\right) + 32\pi\right)}{\pi^3 d^2 \left(-2 + \pi\right)^2} + O\left(d^{\frac{-5}{2}}\right),$$

$$a_2 = \frac{2}{d} + \frac{-2 + \frac{8}{\pi}}{d^2} + O\left(d^{\frac{-5}{2}}\right),$$

$$a_3 = -\frac{4\left(-32\pi + (-2 + \pi)\left(-10\pi^2 - 8\pi + 3\pi^3\right) + 16\pi^2\right)}{\pi^3 d^2 \left(-2 + \pi\right)^2} + O\left(d^{\frac{-5}{2}}\right),$$

$$a_4 = \frac{2 - \frac{4}{\pi}}{d} + \frac{32\left(1 - \pi\right)}{\pi^3 d^{\frac{3}{2}}} + \frac{-\frac{136}{\pi^2} - \frac{128}{\pi^3} - 2 - \frac{512}{\pi^5} + \frac{768}{\pi^4} + \frac{52}{\pi}}{d^2} + O\left(d^{\frac{-5}{2}}\right),$$

$$a_5 = 1 + \frac{8\left(-1 + \pi\right)}{\pi^2 d} +$$

$$- \frac{2\left(-90\pi^3 - 792\pi + \pi\sqrt{-160\pi^3 - 12\pi^5 - 192\pi + 64 + \pi^6 + 240\pi^2 + 60\pi^4 + 384 + 11\pi^4 + 468\pi^2}\right)}{3\pi^4 d^{\frac{3}{2}} \left(-2 + \pi\right)}$$

$$+ \frac{h_4}{d^2} + O\left(d^{\frac{-5}{2}}\right).$$

## F.4 Eigenvalues

| | | | | | | | | |
|---|---|---|---|---|---|---|---|---|
| ɼ-Representation | $\frac{(d-2)(d-3)}{2}$ | $\frac{-2+\pi}{4\pi}$ | | | | | | |
| ŋ-Representation | $\frac{(d-1)(d-4)}{2}$ | $\frac{2+\pi}{4\pi}$ | | | | | | |
| Standard Representation | $d-2$ | $0$ | $\frac{-2+\pi}{4\pi}$ | $\frac{-2+\pi}{4\pi}$ | $\frac{1}{4}$ | $\frac{2+\pi}{4\pi}$ | $\frac{d}{4} + \frac{1}{2}$ | |
| Trivial Representation | $1$ | $0$ | $0$ | $\frac{-2+\pi}{2\pi}$ | $\frac{1}{4}$ | $\frac{d}{4} + \frac{-4+\pi+\pi^2}{-8\pi+2\pi^2}$ | $\frac{d}{4} + \frac{1}{2}$ | $\frac{d}{\pi} + \frac{-10\pi+8+\pi^2}{2\pi(-4+\pi)}$ |

## F.5 Theorem 1, case 1b

$$a_1 = -1 + \frac{2}{d} + -\frac{4\left(-16\pi^2 + (-2 + \pi)\left(-6\pi^3 + 16\pi + 8\pi^2 + \pi^4\right) + 32\pi\right)}{\pi^3 d^2 \left(-2 + \pi\right)^2} + O\left(d^{\frac{-5}{2}}\right),$$

$$a_2 = \frac{2}{d} + \frac{-2 + \frac{8}{\pi}}{d^2} + O\left(d^{\frac{-5}{2}}\right).$$

## F.6 Eigenvalues

| | | | | | |
|---|---|---|---|---|---|
| ɼ-Representation | $\frac{(d-1)(d-2)}{2}$ | $\frac{-2+\pi}{4\pi}$ | | | |
| ŋ-Representation | $\frac{d(d-3)}{2}$ | $\frac{2+\pi}{4\pi}$ | | | |
| Standard Representation | $d-1$ | $0$ | $\frac{-2+\pi}{2\pi}$ | $\frac{1}{4}$ | $\frac{d}{4} + \frac{1}{2}$ |
| Trivial Representation | $1$ | $0$ | $\frac{d}{4} + \frac{-4+\pi+\pi^2}{2\pi(-4+\pi)}$ | $\frac{d}{\pi} + \frac{-10\pi+8+\pi^2}{2\pi(-4+\pi)}$ | |

## F.7 Theorem 1, case 3a

$$a_1 = 1 + \frac{16}{\pi d^2} + O\left(d^{\frac{-5}{2}}\right),$$

$$a_2 = -\frac{8}{\pi d^2} + O\left(d^{\frac{-5}{2}}\right),$$

$$a_3 = \frac{2}{d} + \frac{2\left(-8\pi^2 - \pi^3 - 16 + 4\pi\right)}{\pi^2 d^2 \left(2 + \pi\right)} + O\left(d^{\frac{-5}{2}}\right),$$

$$a_4 = \frac{4}{\pi d} + \frac{32}{\pi^3 d^{\frac{3}{2}}} + \frac{4\left(-24\pi^4 - 160\pi^2 - \pi^5 + 256 + 192\pi + 28\pi^3\right)}{\pi^5 d^2 \left(2 + \pi\right)} + O\left(d^{\frac{-5}{2}}\right),$$

$$a_5 = -1 + \frac{\frac{8}{\pi^2} + 2 + \frac{8}{\pi}}{d} + -\frac{64\left(12 - \pi\right)}{3\pi^4 d^{\frac{3}{2}} \left(-2 + \pi\right)}$$

$$+ \frac{2\left(-112\pi^5 - 3008\pi^2 - 240\pi^4 + 5120 + 2560\pi + \pi^8 + 8\sqrt{2}\pi^6 + 4\sqrt{2}\pi^7 + 672\pi^3 + 12\pi^7 + 104\pi^6\right)}{\pi^6 d^2 \left(4 - \pi^2\right)}$$

$$+ O\left(d^{\frac{-5}{2}}\right),$$

$$a_6 = \frac{2\left(-12\pi + 16 + \pi^3 + 4\pi^2\right)}{\pi^2 d \left(2 + \pi\right)} + \frac{16\left(-24\pi^2 + 64 + 24\pi + \pi^4 + 4\pi^3\right)}{\pi^4 d^{\frac{3}{2}} \left(4 + \pi^2 + 4\pi\right)}$$

$$+ \frac{2\left(\begin{smallmatrix}-184\pi^7 - 26\pi^8 - 3968\pi^3 - 8192\pi^2 - 16\sqrt{2}\pi^7 - 4\sqrt{2}\pi^8 - 112\pi^5 - \pi^9 \\ -16\sqrt{2}\pi^6 + 20480 + 24576\pi + 864\pi^4 + 112\pi^6\end{smallmatrix}\right)}{\pi^6 d^2 \left(8 + \pi^3 + 12\pi + 6\pi^2\right)} + O\left(d^{\frac{-5}{2}}\right).$$

## F.8 Eigenvalues

| | | | | | | | |
|---|---|---|---|---|---|---|---|
| $\mathfrak{r}$-Representation | $\frac{(d-3)(d-4)}{2}$ | $\frac{-2+\pi}{4\pi}$ | | | | | |
| $\mathfrak{y}$-Representation | $\frac{(d-2)(d-5)}{2}$ | $\frac{2+\pi}{4\pi}$ | | | | | |
| Standard Representation $\mathfrak{s}_{d-2}$ | $d-3$ | $0$ | $\frac{-2+\pi}{4\pi}$ | $\frac{-2+\pi}{2\pi}$ | $\frac{1}{4}$ | $\frac{2+\pi}{4\pi}$ | $\frac{d}{4} + \frac{1}{2}$ |
| Standard Representation $\mathfrak{s}_2$ | $1$ | $0$ | $\frac{-2+\pi}{4\pi}$ | $\frac{-2+\pi}{2\pi}$ | $\frac{1}{4}$ | $\frac{d}{4} + \frac{1}{2}$ | |
| Trivial Representation | $1$ | $0$ | $0$ | $\frac{-2+\pi}{2\pi}$ | $\frac{1}{4}$ | $\frac{2+\pi}{4\pi}$ | $\frac{d}{4} + \frac{-4+\pi+\pi^2}{-8\pi+2\pi^2}$ |
| | | | | $\frac{d}{4} + \frac{1}{2}$ | $\frac{d}{\pi} + \frac{-10\pi+8+\pi^2}{2\pi(-4+\pi)}$ | | |
| Tensor Representation $\mathfrak{s}_{d-2} \otimes \mathfrak{s}_2$ | $d-3$ | $\frac{-2+\pi}{4\pi}$ | $\frac{2+\pi}{4\pi}$ | | | | |

## F.9 Identity

$a_1 = 1, a_2 = 0.$

## F.10 Eigenvalues

| | | | | | |
|---|---|---|---|---|---|
| $\mathfrak{r}$-Representation | $\frac{(d-1)(d-2)}{2}$ | $\frac{-2+\pi}{4\pi}$ | | | |
| $\mathfrak{y}$-Representation | $\frac{d(d-3)}{2}$ | $\frac{2+\pi}{4\pi}$ | | | |
| Standard Representation | $d-1$ | $0$ | $\frac{-2+\pi}{2\pi}$ | $\frac{1}{4}$ | $\frac{d}{4} + \frac{1}{2}$ |
| Trivial Representation | $1$ | $0$ | $\frac{d}{4} + \frac{-4+\pi+\pi^2}{2\pi(-4+\pi)}$ | $\frac{d}{\pi} + \frac{-10\pi+8+\pi^2}{2\pi(-4+\pi)}$ | |

## F.11  Theorem 1, case 3b

$$a_1 = -1 + \frac{2}{d} + \frac{-4 + \frac{24}{\pi}}{d^2} + O\left(d^{\frac{-5}{2}}\right),$$

$$a_2 = \frac{2}{d} + \frac{-2 + \frac{12}{\pi}}{d^2} + O\left(d^{\frac{-5}{2}}\right),$$

$$a_3 = \frac{8\left(-4\pi - \pi^2 + 8\right)}{\pi^2 d^2 \left(2 + \pi\right)} + O\left(d^{\frac{-5}{2}}\right),$$

$$a_4 = \frac{2 - \frac{4}{\pi}}{d} + \frac{32\left(1 - \pi\right)}{\pi^3 d^{\frac{3}{2}}} + \frac{2\left(-344\pi^3 - \pi^6 - 512 + 4\pi^4 + 384\pi + 512\pi^2 + 26\pi^5\right)}{\pi^5 d^2 \left(2 + \pi\right)} + O\left(d^{\frac{-5}{2}}\right),$$

$$a_5 = 1 + \frac{8\left(-1 + \pi\right)}{\pi^2 d} + -\frac{8\left(-24\pi^3 - 200\pi + 96 + 3\pi^4 + 120\pi^2\right)}{3\pi^4 d^{\frac{3}{2}} \left(-2 + \pi\right)}$$

$$+ \frac{2\left(-224\pi^6 - 6816\pi^3 - 496\pi^5 - 1088\pi^2 - \pi^8 - 5120 + 9728\pi + 52\pi^7 + 4112\pi^4\right)}{\pi^6 d^2 \left(4 - \pi^2\right)}$$

$$+ O\left(d^{\frac{-5}{2}}\right),$$

$$a_6 = \frac{4\left(-\pi^2 - 8 + 6\pi\right)}{\pi^2 d \left(2 + \pi\right)} + \frac{16\left(-40\pi^2 - 40\pi - \pi^4 + 64 + 20\pi^3\right)}{\pi^4 d^{\frac{3}{2}} \left(4 + \pi^2 + 4\pi\right)}$$

$$+ \frac{4\left(-4816\pi^4 - 116\pi^7 - 4544\pi^3 - 10240 + 4096\pi + 5\pi^8 + 14848\pi^2 + 176\pi^6 + 1632\pi^5\right)}{\pi^6 d^2 \left(8 + \pi^3 + 12\pi + 6\pi^2\right)}$$

$$+ O\left(d^{\frac{-5}{2}}\right),$$

## F.12  Eigenvalues

| | | | | | | | |
|---|---|---|---|---|---|---|---|
| $\mathfrak{r}$-Representation | $\frac{(d-3)(d-4)}{2}$ | $\frac{-2+\pi}{4\pi}$ | | | | | |
| $\mathfrak{y}$-Representation | $\frac{(d-2)(d-5)}{2}$ | $\frac{2+\pi}{4\pi}$ | | | | | |
| Standard Representation $\mathfrak{s}_{d-2}$ | $d-3$ | $0$ | $\frac{-2+\pi}{4\pi}$ | $\frac{-2+\pi}{2\pi}$ | $\frac{1}{4}$ | $\frac{2+\pi}{4\pi}$ | $\frac{d}{4} + \frac{1}{2}$ |
| Standard Representation $\mathfrak{s}_2$ | $1$ | $0$ | $\frac{-2+\pi}{4\pi}$ | $\frac{-2+\pi}{2\pi}$ | $\frac{1}{4}$ | $\frac{d}{4} + \frac{1}{2}$ | |
| Trivial Representation | $1$ | $0$ | $0$ | $\frac{-2+\pi}{2\pi}$ | $\frac{1}{4}$ | $\frac{2+\pi}{4\pi}$ | $\frac{d}{4} + \frac{-4+\pi+\pi^2}{-8\pi+2\pi^2}$ |
| | | | $\frac{d}{4} + \frac{1}{2}$ | $\frac{d}{\pi} + \frac{-10\pi+8+\pi^2}{2\pi(-4+\pi)}$ | | | |
| Tensor Representation $\mathfrak{s}_{d-2} \otimes \mathfrak{s}_2$ | $d-3$ | $\frac{-2+\pi}{4\pi}$ | $\frac{2+\pi}{4\pi}$ | | | | |

## F.13 Theorem 1, case 4

$$a_1 = 1 + \frac{24}{\pi d^2} + O\left(d^{\frac{-5}{2}}\right),$$

$$a_2 = -\frac{12}{\pi d^2} + O\left(d^{\frac{-5}{2}}\right),$$

$$a_3 = \frac{2}{d} + \frac{2\left(-10\pi^2 - 32 - \pi^3 + 16\pi\right)}{\pi^2 d^2 \left(2 + \pi\right)} + O\left(d^{\frac{-5}{2}}\right),$$

$$a_4 = \frac{4}{\pi d} + \frac{32}{\pi^3 d^{\frac{3}{2}}} + \frac{8\left(-17\pi^4 - 144\pi^2 - \pi^5 + 128 + 128\pi + 50\pi^3\right)}{\pi^5 d^2 \left(2 + \pi\right)} + O\left(d^{\frac{-5}{2}}\right),$$

$$a_5 = -1 + \frac{\frac{8}{\pi^2} + 2 + \frac{8}{\pi}}{d} + -\frac{64\left(12 - \pi\right)}{3\pi^4 d^{\frac{3}{2}} \left(-2 + \pi\right)}$$

$$+ \frac{2\left(-288\pi^5 - 5056\pi^2 - 272\pi^4 + 5120 + \pi^8 + 3584\pi + 8\sqrt{3}\pi^6 + 4\sqrt{3}\pi^7 + 16\pi^7 + 1824\pi^3 + 144\pi^6\right)}{\pi^6 d^2 \left(4 - \pi^2\right)}$$

$$+ O\left(d^{\frac{-5}{2}}\right),$$

$$a_6 = \frac{2\left(-12\pi + 16 + \pi^3 + 4\pi^2\right)}{\pi^2 d \left(2 + \pi\right)} + \frac{16\left(-24\pi^2 + 64 + 24\pi + \pi^4 + 4\pi^3\right)}{\pi^4 d^{\frac{3}{2}} \left(4 + \pi^2 + 4\pi\right)}$$

$$+ \frac{2\left(\begin{smallmatrix}-1920\pi^5 - 164\pi^7 - 28\pi^8 - 4992\pi^3 - 13824\pi^2 - 16\sqrt{3}\pi^7 - 4\sqrt{3}\pi^8 - \pi^9 \\ -16\sqrt{3}\pi^6 + 20480 + 28672\pi + 312\pi^6 + 4960\pi^4\end{smallmatrix}\right)}{\pi^6 d^2 \left(8 + \pi^3 + 12\pi + 6\pi^2\right)} + O\left(d^{\frac{-5}{2}}\right).$$

## F.14 Eigenvalues

| | | | | | | | | |
|---|---|---|---|---|---|---|---|---|
| $\mathfrak{x}_{d-3}$-Representation | $\frac{(d-5)(d-4)}{2}$ | $\frac{-2+\pi}{4\pi}$ | | | | | | |
| $\mathfrak{x}_3$-Representation | $1$ | $\frac{-2+\pi}{4\pi}$ | | | | | | |
| $\mathfrak{y}$-Representation | $\frac{(d-3)(d-6)}{2}$ | $\frac{2+\pi}{4\pi}$ | | | | | | |
| Standard Representation $\mathfrak{s}_{d-3}$ | $d-4$ | $0$ | $\frac{-2+\pi}{4\pi}$ | $\frac{-2+\pi}{2\pi}$ | $\frac{1}{4}$ | $\frac{2+\pi}{4\pi}$ | $\frac{d}{4} + \frac{1}{2}$ | |
| Standard Representation $\mathfrak{s}_3$ | $2$ | $0$ | $\frac{-2+\pi}{4\pi}$ | $\frac{-2+\pi}{2\pi}$ | $\frac{1}{4}$ | $\frac{d}{4} + \frac{1}{2}$ | | |
| Trivial Representation | $1$ | $0$ | $0$ | $\frac{-2+\pi}{2\pi}$ | $\frac{1}{4}$ | $\frac{2+\pi}{4\pi}$ | $\frac{d}{4} + \frac{-4+\pi+\pi^2}{-8\pi+2\pi^2}$ | |
| | | $\frac{d}{4} + \frac{1}{2}$ | | $\frac{d}{\pi} + \frac{-10\pi+8+\pi^2}{2\pi(-4+\pi)}$ | | | | |
| Tensor Representation $\mathfrak{s}_{d-3} \otimes \mathfrak{s}_3$ | $2d-8$ | $\frac{-2+\pi}{4\pi}$ | $\frac{2+\pi}{4\pi}$ | | | | | |