# OpenReview forum: "Analytic Study of Families of Spurious Minima in Two-Layer ReLU Neural Networks: A Tale of Symmetry II"
_NeurIPS.cc/2021/Conference — NeurIPS 2021 Poster_

### Official Review · Reviewer_NQUU · 2021-07-11

**Rating:** 8
**Confidence:** 3

**Summary:**

The paper presents theory for spurious minima in the classic two-layer ReLU network with squared error loss under the student-teacher framework. The paper establishes analytic expressions for families of minima, including global and local minimizers, according to different isotropy. Then, the paper presents analytic expressions for the Hessian spectrum, i.e. the eigenvalues and their multiplicity. The results support prior findings on skewed spectrum of Hessians in this setting, and further indicate that spurious and global minima can hardly be distinguished by the spectrums.

**Limitations And Societal Impact:**

Yes.

**Main Review:**

Theoretical understanding of spurious minima in ReLU networks is very significant as it informs why and when training these networks can be successful. This is also a hard area because of large amount of parameters, non-linearity, non-convexity, and stochasticity. The analytic results in this paper is original and significant for the community to understand what may occur in a specific, shallow setting. The theoretical tools used in this paper, as well as the insights and conclusions, are very meaningful although the optimization problem is largely simplified. The paper is well written and easy to read with heavy math being well organized.

I enjoyed reading the paper and learned a lot. The symmetry breaking framework and results are elegant. The theortical findings (on spectrum of the Hessian) are consistent with prior empirical findings and informative. The similarity between spurious and global minima in the sense of Hessian spectrum is especially interesting. I think this paper provides an important piece of understanding to the community of ReLU neural network theory.

Below are some (mostly minor) questions in the order of line numbers.

1, Line 2: because the teacher network outputs real values, it is not accurate to call them "labels". Instead, it is better to call them "target values" or "target outputs".

2, Regarding Theorem 1, are those families of minima the only ones? If not, is there any characterization for the rest families? Are the rest families analytic / intractable / very few?

3, Line 183-184: "this is obtained by ... by zeroes" is a little confusing. Which of $W$ or $V$ (or both) do you pad with zeroes? If you pad $V$, then $d$ becomes even larger. If you pad $W$, since the weights are learned instead of assigned, how do you deal with this? This sentence is important for the readers to understand why the $d=k$ results can be extended to the $d>k$ case, so I suggest the authors to explain more here.

4, In Theorem 2, by saying "normalized to one", do you mean setting each $\alpha_i$ to be $\pm1$ by rescaling between each $\alpha_i$ and $w_i$ *after* the optimization process?

5, A general question on the problem formulation: does your theory naturally extend to the empirical average version of eq (2), that is, $\frac1n\sum_{j=1}^n Loss(x_j)$ instead of $\mathbb{E}_x Loss(x)$, where $x_j\sim\mathcal{N}(0,I_d)$?


**Time Spent Reviewing:**

4 hours.

---

> ### Author Response · Authors · 2021-08-10
> **Response to Reviewer NQUU**
>
> > Line 2: because the teacher network outputs real values, it is not accurate to call them "labels". Instead, it is better to call them "target values" or "target outputs".
>
> **The term “labels”.** Thanks for the suggestion. The term ”labels” seems to be used in the context of regression where outputs are real numbers, as well as other relevant works which concern the student-teacher framework. See e.g. section 3.2.2 in the book “Machine Learning: Foundations and Algorithms” by Ben-david and Shalev-Shwartz. We will add a remark on this choice of terminology in the paper.
>
> > Regarding Theorem 1, are those families of minima the only ones? If not, is there any characterization for the rest families? Are the rest families analytic / intractable / very few?
>
> **Identifying all minima**. Please see general comment 3 above.
>
> > Line 183-184: "this is obtained by ... by zeroes" is a little confusing. Which of $W$ or $V$ (or both) do you pad with zeroes? If you pad $V$, then $d$ becomes even larger. If you pad $W$, since the weights are learned instead of assigned, how do you deal with this? This sentence is important for the readers to understand why the $d=k$ results can be extended to the $d>k$ case, so I suggest the authors to explain more here.
>
> **Extending solutions from k=d to k<d.** Please see general comment 4 above.
>
> > In Theorem 2, by saying "normalized to one", do you mean setting each $\alpha_i$ to be $\pm1$ by rescaling between each $\alpha_i$ and $w_i$ after the optimization process?
>
> **Normalization of the 2nd layer.** The normalization of $\alpha_i$ to $\pm1$ is only used as a means of simplifying the presentation of the expressions of the minima; any choice of $\lambda_i$ in theorem 1 will yield a valid local minimum. The expressions provided in theorem 1 describe families of local minima independently of the optimization process. We will make this point clearer in the paper.
>
> > A general question on the problem formulation: does your theory naturally extend to the empirical average version of eq (2), that is, $\frac{1}{n} \sum_{j=1}^n {Loss}(x_i)$  instead of
> $\mathbb{E}_x {Loss}(x)$, where $x_j\sim \mathcal{N}(0,I_d)$?
>
> **Symmetry of the training loss.** That is a very interesting question: current theoretical bounds for the number of samples required for the training loss to uniformly converge (UC) to the population loss are worst-case and are known to miserably fail when used in the parameter regimes encountered in practice (cf., [ https://arxiv.org/abs/1703.11008 ]). We hope that the symmetry-based approach might provide a different perspective (not based on UC) as to why SGD succeeds in these highly non-convex problems nonetheless. Indeed, preliminary numerical experiments we conducted confirm that minima of the training loss are also (approximately) highly symmetric. Thus, potentially, rather than worst-case notions of sample complexity measured over the space of all possible weight matrices, a better quantification might be obtained by focusing on a restricted set of highly symmetric matrices.

---

### Official Review · Reviewer_nCPe · 2021-07-16

**Rating:** 6
**Confidence:** 3

**Summary:**

This paper studies the property of spurious local minima in two-layer ReLU neural networks. In particular, it provides analytic expressions as well as the corresponding Hessians of some families of local minima, and by analyzing the Hessian spectrum it provides justification for some empirical observation.

**Limitations And Societal Impact:**

The authors addressed the limitations. No direct potential negative societal impact of their work seems to exist.

**Main Review:**

The direct analysis of the loss and the Hessian spectrum of local minima is an interesting direction to pursue. The calculated Hessian spectrum of the spurious local minima matches the empirical observation that most eigenvalues concentrate near zero, which may be a good justification for practice. The symmetry of the families of spectrum of local minima also seems interesting, and the mechanism behind this is definitely worth further exploration.

Nevertheless, I do have some questions/concerns about this paper.

1. If I didn't miss something, the rigorous proof of the theorems seems to be missing. It is possible that the proof applies from other works, but if that is the case please explicitly remark on this. Currently I am not fully sure about the correctness of the proof.

2. Theorem 1 discusses some families of the local minima. Are these all the local minima contained in the network? If not, how to analyze other local minima, and will they possess the same property as claimed in this paper?

3. I'm wondering what the "break" of the symmetry of spurious local minima implies. Will it be observed in practice?

4. What if the activation function is not ReLU? Will the analysis be totally different?

With the questions above, I think the paper needs to address some of the issues before being accepted (at least adding a complete proof). It definitely has the potential of being a good paper, but perhaps not the current version.

Update after author response:

After reading the author's response, I'm convinced of the rigorousness of the proof. Therefore I'm happy to raise my score to 6. I still recommend the authors to include the formal details from [41] in this paper (possibly in the appendix) to make it more self-complete.

**Time Spent Reviewing:**

5 hours

---

> ### Author Response · Authors · 2021-08-10
> **Response to Reviewer nCPe**
>
>
> > If I didn't miss something, the rigorous proof of the theorems seems to be missing. It is possible that the proof applies from other works, but if that is the case please explicitly remark on this. Currently I am not fully sure about the correctness of the proof.
>
> **Correctness and completeness.** We have devoted a substantial amount of work to double-check all aspects of the symmetry-based analysis, including the development of a symbolic program to cross-verify all the quantitative results presented in the paper. Please let us know of any specific concerns you have regarding the technical derivation of our results. Theorem 1 uses a general symmetry-based approach developed in [41]. In the paper, we provide a high-level description of the mathematical techniques used (Section 4), and refer to [41] for the rigorous mathematical details. Concerning the proof of Theorem 2, sections D, E and F in the appendix provide a full account of the derivation method used to compute the Hessian spectrum, including a detailed introductory-level exposition of relevant notions from the representation theory of groups (section D).
>
> > Theorem 1 discusses some families of the local minima. Are these all the local minima contained in the network? If not, how to analyze other local minima, and will they possess the same property as claimed in this paper?
>
> **Identifying all minima.** Please see general comment 3 above.
>
> > I'm wondering what the "break" of the symmetry of spurious local minima implies. Will it be observed in practice?
>
> **Symmetry breaking in practice.** Absolutely. You can implement the SGD algorithm for Problem 2 on your personal computer and see how it sometimes gets trapped in local minima with an intriguing symmetry (which we exploit in the paper). Please also see general comment 2 above.
>
>
> > What if the activation function is not ReLU? Will the analysis be totally different?
>
> **Symmetry breaking for non-ReLU activation.** Please see general comment 5 above.

---

> > ### Comment · Reviewer_nCPe · 2021-08-31
> > **Thank you for the response**
> >
> > After reading the response, I'm convinced of the rigorousness of the proof. Therefore I'm happy to raise my score to 6. I still recommend the authors to include the formal details from [41] in this paper (possibly in the appendix) to make it more self-complete.

---

### Official Review · Reviewer_ceVj · 2021-07-19

**Rating:** 6
**Confidence:** 3

**Summary:**

Under a student-teacher setup with unit Gaussian input, this submission studies the Hessian of a finite-width two-layer ReLU network at critical points of the (population) MSE loss, and characterizes the spectrum and loss at different minima. The starting observation is a symmetry-breaking property based on which a power series description of the minima can be derived, which then leads to an analytic description of the Hessian (up to a dimension-dependent residual). The analysis reveals interesting non-asymptotic properties of the landscape of two-layer network.

**Main Review:**

## Strength

The non-convex landscape of neural network is an important subject of study in deep learning theory.
To my knowledge, most prior works studying the spectrum of Hessian (or the Fisher / NTK) considered the model at random initialization, for which exact description is possible in certain asymptotic limits (e.g., proportional scaling of the network width and input size). In contrast, this submission derived an analytic description of the Hessian for finite-width models at critical points (up to $d^{-1/2}$ terms). This setting leads to a number of interesting findings, including
(i) the similar spectral properties of the Hessian at the global and local minima.
(ii) skewed spectrum of the Hessian.
(iii) the different decay rates of loss at different local minima.

## Weakness

1. The theoretical setup is a bit idealized, for the following reasons.
- The analysis only deals with the population loss (with unit Gaussian data); hence it is unclear if these findings can be translated to the more common setting of empirical risk minimization.
- The assumptions on the target network in Theorem 1 and 2 (i.e., identity first-layer and all-one second-layer) are quite restrictive.
- The current study does not cover the $k>d$ case, which is interesting due to the presence of overparameterization.
- The analysis does not take optimization into account. As the authors noted, gradient descent starting from standard initialization favors certain minima over others, so a more practically relevant object to study would be the Hessian of a trained neural network.

2. The writing and readability of the manuscript can be improved.
- The main theorem is lengthy and not easy to parse. It would be nice to add some more interpretation of the result (especially the structure of $W(d)$), or defer the complete statement to the appendix.
- What is the setting of Figure 1 (right)? Is it still the case that $k=d$, and both are growing? If yes, then this is probably not the typical notion of overparameterization (which corresponds to fixing $d$ and increasing $k$); if not, then this setting does not align with Theorem 1.

**Additional Comments and Questions**

1. It is mentioned that the same analysis can be applied to other choices of target network. Can the authors comment on the difference and similarity we should expect under a more general setting?

2. The finding that minima with almost identical Hessian spectra can have distinct loss is interesting. Is there an intuitive explanation of why these different loss solutions all have similar Hessian spectrum?

3. Some relevant papers:
- In the overparameterized case, it has been shown in [Akiyama and Suzuki 2021] that gradient descent on the empirical loss can identify the parameters of a target ReLU network with high probability.
Akiyama and Suzuki, ICML 2021. On Learnability via Gradient Method for Two-Layer ReLU Neural Networks in Teacher-Student Setting.
- The skewed spectrum of the Fisher matrix (related to the Hessian) at initialization has been theoretically investigated in [Karakida et al. 2019].
Karakida et al., AISTATS 2019. Universal statistics of fisher information in deep neural networks: Mean field approach.


**Post-rebuttal Update**
I have read the author's response and the other reviews. The authors addressed some of my concerns; hence I decided to keep my original score of borderline accept.
One followup comment is that I still do not fully understand what the authors meant by "spurious minima may turn into
saddles, by using over-parameterization." The normal notion of overparameterization corresponds to increasing the width $k$ under fixed input dimensionality $d$. Where exactly is this setting investigated in the main text?

**Time Spent Reviewing:**

2-3 hours.

---

> ### Author Response · Authors · 2021-08-10
> **Response to Reviewer ceVj**
>
> > The analysis only deals with the population loss (with unit Gaussian data); hence it is unclear if these findings can be translated to the more common setting of empirical risk minimization.
>
> **Population vs. Training loss.** Following a recent series of works [1, 2, 3, 4, 5, 6, 7, 8, 9] and Hardt et al., 2016; Hardt & Ma, 2016, we focus on the population loss. Various properties of the training loss can be deduced by concentration of measure arguments, e.g., uniform convergence bounds for gradients (or higher-order derivatives) using generalized vector-valued Rademacher complexity (e.g., Foster et al. 2018, Mei et al. 2017).
>
> > The assumptions on the target network in Theorem 1 and 2 (i.e., identity first-layer and all-one second-layer) are quite restrictive.
>
> **Assumptions in theorems 1 and 2 (+ Q1 in the review)**. For the second layer any choice of positive weights is valid due to the use of the slack variables $\lambda_i$ introduced before the statement of Theorem 1. For the weight matrix of the first layer, the principle of symmetry breaking applies to other choices of weight matrices (see appendix section A.2 and _'Scope of applicability of the symmetry-based approach'_ below), which implies that the symmetry-based approach developed in the paper applies to a much wider class of target matrices (the scope of which is currently under study). We shall clarify this in the paper.
>
> > The current study does not cover the $k>d$ case, which is interesting due to the presence of overparameterization.
>
> **The $k>d$ case.** Please see general comment 1 above.
>
> > The analysis does not take optimization into account. As the authors noted, gradient descent starting from standard initialization favors certain minima over others, so a more practically relevant object to study would be the Hessian of a trained neural network.
>
> **The Hessian of a trained neural network.** Theorem 1 provides power series representation of local minima of Problem 2, the minima of which correspond to _trained_ neural networks. Theorem 2 then provides an analytic characterization of the spectrum at these trained neural networks to $O(d^{-1/2})$-terms, conforming with your suggestion.
>
> > The main theorem is lengthy and not easy to parse. It would be nice to add some more interpretation of the result (especially the structure of $W(d)$), or defer the complete statement to the appendix.
>
> **Interpretation of Theorem 1.** Please see general comment 2 above.
>
> > What is the setting of Figure 1 (right)? Is it still the case that $k=d$, and both are growing? If yes, then this is probably not the typical notion of overparameterization (which corresponds to fixing  and increasing ); if not, then this setting does not align with Theorem 1.
>
> **Figure 1 (right).** Here it is shown how saddles turn into local minima when $k=d$ increases from 3 (has negative eigenvalues) to 20 (all eigenvalues are positive). Preliminary numerical and analytical results indicate that this process can be reversed; that is, local minima transform into saddles if $d$ is fixed and $k$ is _continuously_ increased.
>
> > It is mentioned that the same analysis can be applied to other choices of target network. Can the authors comment on the difference and similarity we should expect under a more general setting?
>
> **Scope of applicability of the symmetry-based approach.** The enabler of the symmetry-based analysis presented in the paper is the principle of symmetry breaking. Fortunately, this principle applies to a wide class of fundamental non-convex problems, including other choices of target matrices, distributions (see appendix section A.2 and [42]) and non-ReLU activations which are related to tensor decomposition problems. Please see general comment 5 above.
>
>
> > The finding that minima with almost identical Hessian spectra can have distinct loss is interesting. Is there an intuitive explanation of why these different loss solutions all have similar Hessian spectrum?
>
> **Identical Hessian spectra.** At the moment we do not have an intuitive explanation as to why different minima with a completely different behavior share an Hessian spectrum identical to $O(d^{-1/2})$-terms. This is currently one of our main research objectives; that is, looking for a principle which might govern the limiting models of the ReLU networks studied in the paper.
>
> > Some relevant papers.
>
> We appreciate the references:
> * [Akiyama and Suzuki 2021]: Here the optimization algorithm under consideration is different from the SGD or GD methods used in practice.
> * [Karakida et al. 2019]: The line of work which studies properties of NN with random weight matrices is covered in the discussion starting at line 241 and regards papers [57, 58, 59].
>
> We will add both suggested citations.

---

### Official Review · Reviewer_7HVf · 2021-07-27

**Rating:** 6
**Confidence:** 2

**Summary:**

The paper theoretically study a one hidden layer neural network with ReLU activation trained using Gaussian data, and analytically derive the values of the different classes of local minima of this objective, and the spectrum of the Hessian at these local minima. The results show that as the degree of over-parameterization is increased, the local minima transform into saddle points in the loss landscape, and further find the exact fractional dimension at which this transition occurs.

Note: I am not familiar with the mathematical tools used in this paper. So my comments pertain to the utility of the findings and claims made in the paper.

**Limitations And Societal Impact:**

see main review for limitations.
Societal impact: N/A

**Main Review:**

Introduction and section 1 describe the problem setup. In section 2, the various class of local minima are derived analytically and the difference in the decay of their Hessian spectrum is discussed. Section 3 reveals the spectrum of Hessian of the families of local minima discovered in theorem 1, and shows that the bulk of eigenvalues are small positive values while a small subset of eigenvalues grow linearly with d (the dimensionality of the hidden layer of the target network). In section 4, the authors develop a symmetry-based framework as a way to analytically derive the power series representation of minima and the Hessian spectrum.

strengths:
- the paper studies an important problem in deep learning optimization.
- the paper provides a currently less mainstream approach to study 2 layer ReLU network with the goal of deriving the form of local minima and spectrum of the Hessian at these minima.

weakness:
- the theoretical results are not properly motivated and their implications are discussed. More on this below.
- some of the claims in the paper are misleading or not true. See below.

Detailed review:

- It is not explicitly mentioned anywhere but the dimensionality d of the input vector x is assumed to be the same as the dimensionality d of beta (number of neurons in the hidden layer) in the target network (Eq 2). What is the reasoning behind this and what are the implications of this assumption on the outcome of the analysis?

- It would be nice to establish a connection between dimensionality d and the number of classes C in a classification task (at least intuitively). This would be particularly helpful because it has been shown in [1] that the eigenvalues of the Hessian tend to have C large eigenvalues while the others are nearly zeros. So the dependence on d in the submitted paper should clearly have some connection with C.

- the main statement of section 1: "Spurious minima break the symmetry of global minima" and figure 2, are not very clear. Perhaps a more intuitive explanation of this statement and describing the physical significance of this claim would help.

- On line 117-120, the authors cite [48,49] while claiming that (specifically) local minima lie in fixed low-dimensional subspaces of the parameter space. However, the cited papers do not mention the minima found were local. In fact fig. 1 in [48] suggests the minimum is global.

- section 2 derives the analytical form of the various classes of local minima and the difference in the decay of their Hessian spectrum is discussed. While this is a strong result, it would be informative to also discuss how these differences can impact gradient descent based optimization (e.g. in terms of the likelihood of discovering these minima) or other physical significance of this result.

- One of the claims in the paper (e.g. line 323-329) is around the exposition of how the proposed theory explains why local minima are not major limitation in over-parameterized deep networks since over-parameterization turns local minima into saddle points. However, this claim is not novel and has been known at least since [2] if not before.

[1] Sagun, Levent, et al. "Empirical analysis of the hessian of over-parametrized neural networks." arXiv preprint arXiv:1706.04454 (2017).

[2] Dauphin, Yann, et al. "Identifying and attacking the saddle point problem in high-dimensional non-convex optimization." arXiv preprint arXiv:1406.2572 (2014).

Because of the above concerns, I am giving a score of 6 to the paper. If addressed, I am willing to increase the score.

**Time Spent Reviewing:**

15 hours

---

> ### Author Response · Authors · 2021-08-10
> **Response to Reviewer 7HVf**
>
> > It is not explicitly mentioned anywhere but the dimensionality d of the input vector x is assumed to be the same as the dimensionality d of beta (number of neurons in the hidden layer) in the target network (Eq 2). What is the reasoning behind this and what are the implications of this assumption on the outcome of the analysis?
>
> **Dimensionality vs. number of hidden neurons.** Please see general comment 1 above.
>
>
> > It would be nice to establish a connection between dimensionality d and the number of classes C in a classification task (at least intuitively). This would be particularly helpful because it has been shown in [1] that the eigenvalues of the Hessian tend to have C large eigenvalues while the others are nearly zeros. So the dependence on d in the submitted paper should clearly have some connection with C.
>
> **Number of classes.** Sagun, Bottou and LeCun 2016 [32] is a great work which has been inspiring for us (see also [33] and the discussions starting at lines 72 and 233) . When we managed to obtain an analytic characterization of the Hessian spectrum, we were excited to realize that it completely conforms with what was (only) empirically observed before. Indeed, in [33] the authors show that _"the spectrum of the Hessian is composed of two parts: (1) the bulk centered near zero, (2) and outliers away from the bulk.”_. The symmetry based approach rigorously establishes this empirical observation. In our settings the magnitude of the dominating eigenvalues grows linearly with d and their cardinality, counting multiplicity, is d+1 (out of a total of d^2 eigenvalues. See Theorem 2). This can indeed indicate an effective number of d+1 classes in the regression problem studied in the paper. We will add a brief discussion of this point in the body of the paper.
>
>
>
> > the main statement of section 1: "Spurious minima break the symmetry of global minima" and figure 2, are not very clear. Perhaps a more intuitive explanation of this statement and describing the physical significance of this claim would help.
>
> **The principle of symmetry breaking.** Please see general comment 2 above.
>
>
>
> > On line 117-120, the authors cite [48,49] while claiming that (specifically) local minima lie in fixed low-dimensional subspaces of the parameter space. However, the cited papers do not mention the minima found were local. In fact fig. 1 in [48] suggests the minimum is global.
>
> **Hidden low-dimensional structure.** Papers [48, 49] study settings in DL where a hidden low-dimensional structure is evident and is suggested as a means of addressing current theoretical gaps. In our paper, the low dimensional fixed point spaces forced by symmetry appear to form yet another instance of a general principle of this hidden low-dimensional structure. In the symmetric case, these subspaces typically contain local minima as well as instances of the global minima. It was not our intention to imply that the minima in Fig. 1 of [48] were not global minima.  We will make sure that this point is clear in the revision. We emphasise that in our setting the fixed point spaces always contain at least one global minimum (in the symmetric case there may be many critical points, all symmetrically related).
>
>
>
>
> > section 2 derives the analytical form of the various classes of local minima and the difference in the decay of their Hessian spectrum is discussed. While this is a strong result, it would be informative to also discuss how these differences can impact gradient descent based optimization (e.g. in terms of the likelihood of discovering these minima) or other physical significance of this result.
>
> **Gradient-based optimization.** In terms of dynamical accessibility, the various classes of minima studied in the paper behave rather differently. For example, one insight obtained through the symmetry-based framework is that if Xavier initialization is used then, although different minima have Hessian spectra identical to d^{-½} terms, SGD seems to favor the class of minima for which the objective value decays as Θ(1/d). Of course, a better understanding of the mechanism which drives SGD towards adequate minima (i.e., the _"likelihood of discovering these minima"_) is required, and we hope that the symmetry-based approach will shed more light on this mechanism. This is a part of a more general effort currently taken by the ML community to understand various beneficial biases of SGD (e.g., https://arxiv.org/abs/1710.10345 and follow-ups). Please also see the discussion in lines 243-252 regarding the flat minima conjecture which addresses possible implications of local curvature for the dynamics of gradient-based methods.
>
>
>
> > One of the claims in the paper (e.g. line 323-329) is around the exposition of how the proposed theory explains why local minima are not major limitation in over-parameterized deep networks since over-parameterization turns local minima into saddle points. However, this claim is not novel and has been known at least since [2] if not before.
>
> **Creation of saddles.** [2] is an interesting paper which has been very influential since its publication. Our work differs in three fundamental aspects: A. [2] only covers numerical results, rather than the precise analytic information of the Hessian spectrum derived using the symmetry-based approach. B. Our own work further addresses the phenomenon in which local minima dissolve into saddle points under over-parameterization--a phenomenon which we believe to be crucial to the success of DL. C. In [2], the theory follows from random Gaussian fields (see section 3 in [2]) rather than concrete instances of ReLU networks of finite width which we address in this work. We shall add this clarification to the body of the paper.

---

### Author Response · Authors · 2021-08-10
**General Comments**

We thank the reviewers for their time, feedback and most helpful critical comments.

1. **Dimensionality $d$ vs. number of hidden neurons $k$.** The power series representation and the derived Hessian spectrum for different families of minima stated in the paper refer to the regime where $k \le d$, the primary focus of this work (see page 2, line 65). The case where $k = d$ is explicitly listed in the hypotheses of Theorem 1. The same analysis applies, mutatis mutandis, when $k < d$ as we describe in the follow-up discussion of Theorem 1 (lines 175-181, page 6). In particular, each family of spurious minima we obtain for $k = d$, naturally determines a family of spurious minima for all $k < d$ (see comment 4 below). Our work is part of a general program to understand the creation and the annihilation of spurious minima in the highly nonconvex landscapes associated with ReLU networks. The overparametrized case $k>d$ uses techniques described in our article, but also depends on ideas from equivariant bifurcation theory which are not needed for the case $k\le d$. We will make the reference to the different regimes of $d$ and $k$ clearer in the introduction and also add brief details on the power series extension.

2. **The principle of symmetry breaking.** There is a natural action of $S_k \times S_d$ on the space of $k \times d$ matrices: $S_k$ permutes rows, $S_d$ permutes columns. This group action allows one to measure the symmetry of a given weight matrix $W$ as the subgroup of $S_k \times S_d$ fixing $W$ (see the definition of “isotropy group”, line 99. Equation (3)). We found that, empirically, the isotropy group of local minima are sub-groups of the isotropy group of global minima. Thus, figuratively speaking, “Spurious minima break the symmetry of global minima”. This empirical principle forms the basis for our analytic approach (see lines 104-110 for a more general discussion of this principle). The principle of symmetry breaking further allows one to predict the structure of local minima to a large extent. Indeed, if the isotropy of a given weight matrix is known to be a large subgroup of the isotropy of the target matrix then its entries must exhibit a rather restrictive form. Examples are given in Figure 2. Please let us know if you find this explanation clearer, and we will adapt the wording used in the paper.

3. **Identifying all minima.** Other families of minima for Problem 2 exist. We have observed that all minima obey the principle of symmetry breaking (see comment 2 above), and are therefore amenable to the same symmetry-based analysis shown in the paper. The powerful language of symmetry is what makes possible the, otherwise seemingly hopeless, task of identifying and classifying all minima in the highly nonconvex optimization landscape associated with problem 2.

4. **Extending solutions from $k=d$ to** $k<d$.   Starting with a $k \times k$ matrix $W$ defining a non-degenerate local minimum, suppose $d > k$ and define a $k \times d$ matrix $W^\star$ by adding $d-k$ zeros to the end of each row of $W$. The matrix $V$ is similarly extended to a $k \times d$ matrix. It is easy to see that for the new optimization problem on $M(k,d)$, $W^\star$ is a critical point, the Hessian at $W^\star$ has all eigenvalues strictly positive ([9], Theorem 30, Appendix E) and so defines a non-degenerate local minimum. The basic idea for the extension appears in [9].

5. **Symmetry breaking for non-ReLU activation.** The symmetry-based technique also applies to non-ReLU activations, such as polynomial activations (some of which lead to the fundamental class of tensor decomposition problems). Non-ReLU activations are studied in detail in paper [43] ([ https://arxiv.org/abs/2103.06234 ]).

---

### Decision · Program_Chairs · 2021-09-27

**Decision:**

Accept (Poster)

**Comment:**

This paper gives a detailed characterization of spurious local minima for 2-layer ReLU networks, which was a model that received substantial theoretical interest. The paper was able to do this using a powerful technique of symmetry-breaking, and the results explained some of the observations in previous works. Overall the paper is a very solid contribution to the understanding of optimization landscape of neural networks and the symmetry-breaking technique might be useful in more general settings.